# AgentGym: Evaluating and Evolving Large Language Model-based Agents across Diverse Environments

## Abstract

Large language models (LLMs), with their generalized capabilities, are considered as a promising foundation to build generally-capable agents that can handle multi-turn decision-making tasks across various interactive environments. Previous attempts typically gather expert-provided trajectories and have LLM-based agents imitate these trajectories step-by-step. However, this supervised fine-tuning approach depends heavily on human supervision, limiting scalability and restricting the agent's exploration and learning in the environments. In this paper, we take the first step towards developing generally-capable LLM-based agents that can explore and evolve themselves across diverse environments. To achieve this, we identify a trinity of ingredients: 1) diverse interactive environments for agent exploration, 2) a trajectory set to equip agents with basic capabilities and prior knowledge, and 3) an effective and scalable approach for agent improvement across environments. We propose AgentGym, a new interactive framework featuring various real-world scenarios and environments for broad, unified, real-time, and concurrent agent exploration. AgentGym also includes a database with expanded instructions, high-quality trajectories, and a benchmark suite. Next, we investigate the potential of agent self-evolution across various environments with a derived exploration-learning method named AgentEvol. Experimental results show that the evolved agents can achieve results comparable to SOTA models. We will release the code, dataset, benchmark, and checkpoints.

## 1 Introduction

Developing agents capable of performing a wide spectrum of tasks across various environments at human-level has been a long-standing goal for AI community (Wooldridge & Jennings, 1995; Silver et al., 2017; 2018; Reed et al., 2022; Xi et al., 2023). Large language models (LLMs) are considered a promising foundation for constructing such generalist agents due to their generalized abilities (OpenAI, 2023; Anthropic, 2024; Anil et al., 2023), and many efforts have been made in this realm to train generally-capable LLM-based agents (Xi et al., 2023; Wang et al., 2024b).

Previous work involves recruiting human experts to interact with various environments to collect high-quality multi-turn trajectories, which are then used to train agents to imitate these trajectories step by step through behavioral cloning (BC) [1]. This method, while effective, requires skilled annotators and significant financial resources, making it hard to scale (Yang et al., 2024). Moreover, such a paradigm may encounter bottlenecks in performance and adaptability due to insufficient exploration of the environment (Aksitov et al., 2023). Another line of research allows LLM-based agents to improve themselves based on environmental feedback (i.e., self-improvement), reducing reliance on human supervision while enriching exploration of the environment (Zhou et al., 2024; Tao et al., 2024; Song et al., 2024). Yet, they typically train agents in isolated environments, and the resulted specialist agents are limited to narrow tasks.

In this paper, we take the initial step to explore the potential of self-evolution in generally-capable LLM-based agents across various environments. We expect to let the agents first perform imitation

---

[1] In LLM field, the term behavioral cloning can be understood as supervised fine-tuning (SFT).

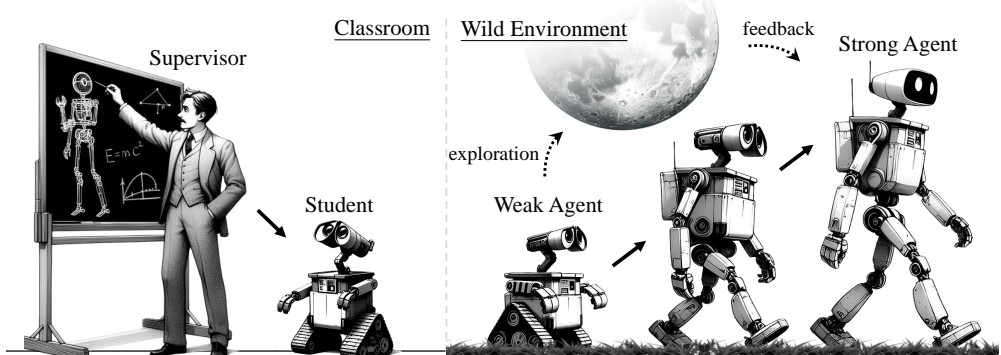

Figure 1: An illustration of self-evolution for generally-capable LLM-based agents in our paper. The agent first imitates human behavior and then performs exploration and learning across various environments and tasks to evolve itself.

and then interact with the environment to evolve themselves just like the way humans learn and develop (Standish, 2003; Taylor et al., 2016; Fan et al., 2022; Team et al., 2024), as shown in Figure 1. To achieve this research goal, we identify three key pillars. First, diverse environments and tasks that allow the agents to explore dynamically and comprehensively, rather than being confined to an isolated world (Standish, 2003; Langdon, 2005; Taylor et al., 2016; Fan et al., 2022). Second, a trajectory set to train a base agent with preliminary abilities and prior knowledge. This facilitates further exploration as in diverse, complex environments, it would be extremely inefficient for an agent to learn everything from scratch through trial and error (Fan et al., 2022; Song et al., 2024). Third, an effective and scalable method for the agents to perform self-improvement across diverse environments and tasks. This involves how the agent interacts with the environment and how it utilizes the feedback to improve itself (Yang et al., 2024; Aksitov et al., 2023).

Considering the three pillars, we present AGENTGYM (see Figure 2), a new framework designed to comprehensively evaluate and develop generally-capable LLM-based agents. Our main contributions are:

**1. An interactive framework that includes diverse scenarios and environments for LLM-based agents.** AGENTGYM offers convenient APIs through HTTP services, standardizing task specifications, environment settings, and the observation/action spaces for agents. Within this framework, we have implemented a unified interface for multi-round interactions and real-time feedback across different environments to support holistic evaluation, trajectory collection, and agent exploration. For comprehensiveness, AGENTGYM identifies 7 real-world scenarios critical for evaluating and developing agents, i.e., web navigating, text games, house-holding tasks, digital games, embodied tasks, tool-using, and programming. Specifically, it encompasses 14 environments and 89 tasks, with high flexibility for further expansion. The diversity of AGENTGYM not only challenges the agents' multitasking abilities but also enhances their adaptability and practicality in real-world applications, laying the foundation for more robust and versatile LLM-based agents.

**2. Expanded instructions, benchmark suite, and high-quality trajectories across environments.** We collect instructions from various environments and tasks, expanding them through rule-based strategies and AI-based techniques such as self-instruct (Wang et al., 2023b) and instruction evolution (Xu et al., 2023). Subsequently, we leverage several principles to construct a benchmark suite named AGENTEVAL to comprehensively evaluate LLM-based agents. Next, we use a gather-and-filter pipeline to get a trajectory set named AGENTTRAJ. The set is used to train a base agent with basic capabilities and prior knowledge, bootstrapping further agent exploration and learning. For a fair comparison, we also collect a larger trajectory set AGENTTRAJ-L with the same pipeline to train an agent that serves as the performance upper bound achievable through SFT/BC. Note that AGENTTRAJ-L is an extension of AGENTTRAJ and we will release it as well.

**3. Investigation of self-evolution for LLM-based agents across various environments with AGENTEVOL method.** Starting from the base agent, we explore its potential of self-improvement across various environments with a proposed method named AGENTEVOL, which is derived based on the classical *RL as Inference* framework (Dayan & Hinton, 1997). We then implement it in an

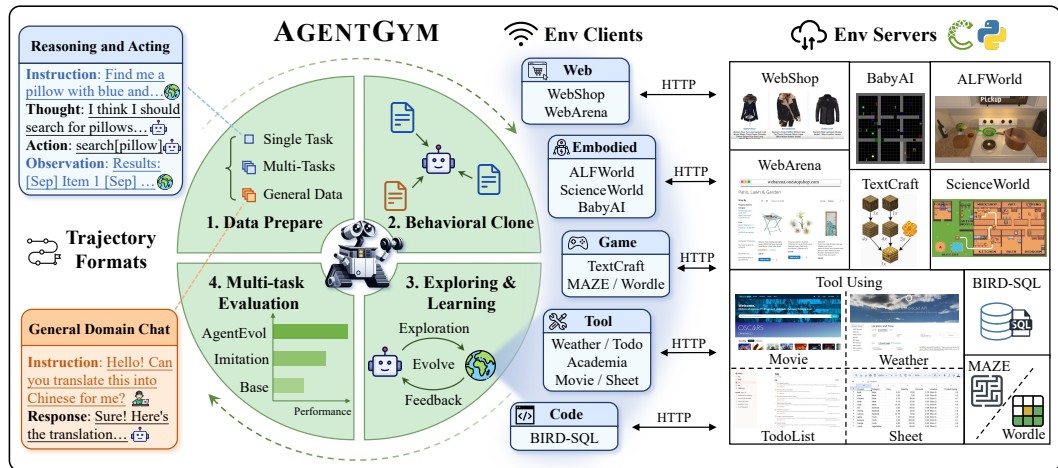

Figure 2: Overview of the AGENTGYM framework. It covers fourteen environments spanning diverse scenarios. It adopts a decoupled client-server architecture for unified and concurrent agent-environment interaction. AGENTGYM also includes expanded instructions, a comprehensive benchmark suite AGENTEVAL, and the high quality trajectory set AGENTTRAJ (-L). We also derive and implement an exploration-learning method named AGENTEVOL to explore the agent's self-evolution across various environments.

exploration-learning paradigm in practice. We focus on whether agents can evolve themselves when facing previously unseen tasks and instructions, where they need to perform exploration and learning. Experimental results demonstrate that the self-evolution process is highly effective, with agents achieving performance on par with or even surpassing SOTA models.

In summary, we propose AGENTGYM (See Figure 2 for an overview), a comprehensive framework that includes an interactive platform covering diverse scenarios, a benchmark suite AGENTEVAL, and two trajectory sets AGENTTRAJ and AGENTTRAJ-L. We also derive and implement an exploration-learning method AGENTEVOL to explore self-evolution in LLM-based agents across various environments. We will release the whole suite, implementations, and checkpoints. We hope AGENTGYM will help the community to develop better generalist LLM-based agents.

## 2  AGENTGYM: FRAMEWORK ARCHITECTURE, INSTRUCTION SET, BENCHMARK SUITE, AND TRAJECTORY SET

AGENTGYM is a framework built for the community to facilitate the evaluation, training, and evolution of generally-capable LLM-based agents. It features diverse interactive environments and tasks with ReAct format (Yao et al., 2023). It supports real-time feedback and concurrency, and is easily scalable and extendable. The comparison between AGENT-GYM and other LLM-based agent frameworks is demonstrated in Table 1.

Table 1: Comparison of AGENTGYM with other agent frameworks covers several aspects: the number of environments, presence of an interactive platform and its usage, availability of trajectory sets, support for evolution, and the evolution mode.

| Frameworks | Env. | Inter. Plat. | Traj. | Evol. |
|---|---|---|---|---|
| AgentBench (Liu et al., 2023a) | 8 | Eval | No | No |
| AgentBoard Ma et al. (2024) | 12 | Eval | No | No |
| AgentOhana (Zhang et al., 2024) | 10 | No | Yes | No |
| Pangu-Agent Christianos et al. (2023) | 6 | No | Yes | Single-Env |
| AGENTGYM (Ours) | 14 | Eval & Train | Yes | Multi-Env |

### 2.1  DIVERSE TARGETED ENVIRONMENTS AND TASKS FOR LLM-BASED AGENTS.

To ensure the comprehensiveness of the framework, we identify 7 real-world scenarios essential for evaluating and developing agents' capabilities, including web navigating, text games, house-holding tasks, digital games, embodied tasks, tool-using, and programming. These scenarios are represented by 14 environments and 89 tasks in our framework, as shown in Table 2. We further elaborate on the motivation for selecting these diverse environments from two perspectives.

Table 2: Statistics of AGENTGYM, including scenarios, count of task types, evaluation metric, instruction set size, evaluation set size, trajectory set size (AGENTTRAJ and AGENTTRAJ-L), and the average interactive rounds of each environment in AGENTTRAJ-L.

| Env. | Scenario | Task Num. | Eval. Metric | Inst. Size | Eval. Size | Traj. Size | Traj-L Size | Rounds |
|---|---|---|---|---|---|---|---|---|
| WebArena (WA, Zhou et al. 2023a) | Web Navigating | 3 | Success rate | 812 | 20 | 0 | 0 | − |
| WebShop (WS, Yao et al. 2022) | Web Navigating | 1 | Success rate | 6910 | 200 | 1000 | 3930 | 5.1 |
| MAZE (MZ, Abdulhai et al. 2023b) | Text Game | 1 | Success rate | 240 | 25 | 100 | 215 | 4.3 |
| Wordle (WD, Abdulhai et al. 2023b) | Text Game | 1 | Success rate | 980 | 25 | 500 | 955 | 4.3 |
| ALFWorld (ALF, Shridhar et al. 2021) | House-holding | 6 | Success rate | 3827 | 200 | 500 | 2420 | 13.3 |
| SciWorld (Sci, Wang et al. 2022) | Embodied Tasks | 30 | Reward | 2320 | 200 | 1000 | 2120 | 19.9 |
| BabyAI (Baby, Chevalier-Boisvert et al. 2019) | Embodied Tasks | 40 | Reward | 900 | 90 | 400 | 810 | 5.7 |
| TextCraft (TC, Prasad et al. 2023) | Digital Game | 1 | Success rate | 544 | 100 | 300 | 374 | 8.0 |
| Tool-Weather (WT, Ma et al. 2024) | Tool Use | 1 | Success rate | 331 | 20 | 160 | 311 | 5.5 |
| Tool-Movie (MV, Ma et al. 2024) | Tool Use | 1 | Success rate | 235 | 20 | 100 | 215 | 4.0 |
| Tool-Academia (AM, Ma et al. 2024) | Tool Use | 1 | Success rate | 20 | 20 | 0 | 0 | − |
| Tool-Sheet (ST, Ma et al. 2024) | Tool Use | 1 | Reward | 20 | 20 | 0 | 0 | − |
| Tool-TODOList (TL, Ma et al. 2024) | Tool Use | 1 | Success rate | 155 | 20 | 70 | 135 | 5.6 |
| BIRD (BD, Zheng et al. 2023a) | Programming | 1 | Success rate | 3200 | 200 | 2000 | 3000 | 1.0 |
| Total | − | 89 | − | 20494 | 1160 | 6130 | 14485 | − |

**Starting from the definition of an LLM-based agent.** An LLM-based agent refers to an agent with a decision-making core based on a large language model, extending its input and action (Wooldridge & Jennings, 1995; Xi et al., 2023; Wang et al., 2024b). Consequently, the capabilities required for an LLM-based agent include:

- **Input Side.** An effective agent must demonstrate the ability to process diverse textual observations, such as plain text, HTML, code, etc. Thus, the environments and tasks in AGENTGYM are designed to encompass a wide range of input types. For instance, we include textual environments (ALF, TC), web-based environments (WS, WA), and coding environments (BD).

- **Decision-making Side.** At the core of an LLM-based agent lies its reasoning, planning and environmental understanding capabilities. Therefore, the tasks within AGENTGYM are crafted to be challenging and rigorous. For example, in the WS task, when interacting with a web page containing many products, an LLM-based agent needs to guide its next actions through reasoning and information extraction. Similarly, in the Sci task, an agent needs to read manuals, find raw materials, and synthesize items when facing observations from different rooms. These tasks demand not only basic common knowledge but also advanced planning and situational understanding capabilities.

- **Action Side.** An LLM-based agent's outputs can take various forms, such as plain text, code, API calls and embodied actions. As a result, the environments must be equipped to accommodate and process these different types of actions. For example, WT and MV environments offer a variety of tool-calling APIs for the agent; the DB environment requires the generation of SQL code; Baby and ALF involve embodied actions, while WD focuses on producing plain text output.

**Starting from the expectation for an LLM-based agent.** An LLM-based agent is expected to tackle complex, long-term, multi-turn sequential decision-making tasks by dynamically interacting with the environment (Yao et al., 2023; Sumers et al., 2024; Zhou et al., 2024). In this context, the environments and tasks included in AGENTGYM are designed to embody these characteristics. Specifically, the tasks require multiple rounds of interaction with the environment, and the context length is usually longer than those of LLM's Q&A or reasoning tasks. To accomplish these tasks, the agent needs to process not only instructions but also inputs derived from historical information and current observations. From this perspective, AGENTGYM is an framework specifically designed for LLM-based agents.

## 2.2 FRAMEWORK ARCHITECTURE AND DATABASE CONSTRUCTION.

We employ a decoupled architecture in AGENTGYM to facilitate standardized and scalable agent-environment interactions. It supports functions like creating environments, retrieving observations and performing actions via HTTP-based communication. See Appendix C for more details.

Regarding database construction, we first gather $20,494$ instructions using rule-based generation and self-instruction techniques. Then, we construct a benchmark suite with a size of $1,160$ named

AGENTEVAL to holistically evaluate the capabilities of LLM-based agents. As for the trajectory set, we collect and filter $6,130$ high-quality trajectories from 11 environments with various strategies, including but not limited to crowdsourcing and annotations by SOTA models (e.g., GPT-4-Turbo). This set, named AGENTTRAJ, is used to train a base agent with preliminary abilities and prior knowledge. For a fair comparison, we also perform the same annotation pipeline on all instructions to get AGENTTRAJ-L, which represents the performance upper bound of BC. More collection and annotating details are presented in Appendix B and D.

## 3 EVOLVING OF LLM-BASED AGENTS ACROSS VARIOUS ENVIRONMENTS

In this section, we explore the potential of agents to evolve themselves across multiple environments. We begin by training a base agent using BC to equip it with basic interaction capabilities. Then, based on the *RL as Inference* theory (Dayan & Hinton, 1997), we derive and implement an exploration-learning method, AGENTEVOL, for multi-turn interaction scenarios, enabling agents to improve themselves across various environments. The method is summarized in Algorithm 1.

### 3.1 PRELIMINARIES

We define the collection of environments as $\mathcal{E}$. For a specific $e \in \mathcal{E}$, we formalize the agent task in the environment as a partially observable Markov decision process (POMDP) $(\mathcal{U}, \mathcal{S}, \mathcal{A}, \mathcal{O}, \mathcal{T}, r)_e$ with instruction space $\mathcal{U}$, state space $\mathcal{S}$, action space $\mathcal{A}$, observation space $\mathcal{O}$, deterministic state transition function $\mathcal{T} : \mathcal{S} \times \mathcal{A} \to \mathcal{S}$, and reward function $r : \mathcal{S} \times \mathcal{A} \to \mathbb{R}$.

Given a task instruction $u$ in environment $e$, the LLM-based agent parameterized by $\theta$ generates an action $a_1 \sim \pi_\theta(\cdot|e, u)$ based on its policy $\pi_\theta$. Then, the state space is transitioned to $s_1 \in \mathcal{S}$, and the agent receives feedback $o_1 \in \mathcal{O}$. Subsequently, the agent interacts with the environment until the task ends or exceeds the maximum number of steps. We adopt ReAct (Yao et al., 2023) to model the outputs of agent, where the LLM-based agent generates a reasoning thought before outputting an action. Thus, at time step $t$, given the history and current feedback, the agent generates the thought $h_{t+1} \sim \pi_\theta(\cdot|e, u, h_1, a_1, o_1, ..., h_t, a_t, o_t)$ first and the subsequent action $a_{t+1} \sim \pi_\theta(\cdot|e, u, h_1, a_1, o_1, ..., h_t, a_t, o_t, h_{t+1})$. Hence, the trajectory can be represented as:

$$\tau = (h_1, a_1, o_1, ..., o_{T-1}, h_T, a_T) \sim \pi_\theta(\tau|e, u), \tag{1}$$

$$\pi_\theta(\tau|e, u) = \prod_{t=1}^{T} \pi_\theta(h_t, a_t|e, u, c_{t-1}) = \prod_{t=1}^{T} \pi_\theta(a_t|e, u, c_{t-1}, h_t) \cdot \pi_\theta(h_t|e, u, c_{t-1}), \tag{2}$$

where $T$ is the number of interaction rounds, and $c_{t-1} = (h_1, a_1, o_1, ..., h_{t-1}, a_{t-1}, o_{t-1})$ represents the interactive history up to $t-1$. The final reward $r(e, u, \tau) \in [0, 1]$ is computed after the interaction ends or the maximum interactive round number is met.

### 3.2 BEHAVIORAL CLONING WITH COLLECTED TRAJECTORIES

In diverse and complex environments, learning everything from scratch through trial and error is inefficient for LLM-based agents due to the vast action space of language space and the varying prior knowledge across different environments (Fan et al., 2022; Song et al., 2024). Hence, we employ the behavioral cloning (BC) method to train a base agent using AGENTTRAJ, providing it with basic interaction capability and prior knowledge. Specifically, BC fine-tunes LLM-based agents by having them mimic the expert trajectories step-by-step. In practice, we expect the agent to generate both appropriate thought $h$ and action $a$. So we maximize the following objective:

$$\mathcal{J}_{BC}(\theta) = \mathbb{E}_{(e,u,\tau) \sim \mathcal{D}_s}\left[ \log \pi_\theta(\tau|e, u) \right]$$

$$= \mathbb{E}_{(e,u,\tau) \sim \mathcal{D}_s} \sum_{t=1}^{T} \left[ \log \pi_\theta(a_t|e, u, c_{t-1}, h_t) + \log \pi_\theta(h_t|e, u, c_{t-1}) \right]. \tag{3}$$

Note that we include a general domain dataset $\mathcal{D}_{general}$ as in Zeng et al. (2023) to maintain the agent's ability in language understanding and generation. And the resulting agent $\pi_{\theta_{base}}$ serves as a starting point for later exploration and learning across diverse environments.

### 3.3 EVOLUTION THROUGH EXPLORATION AND LEARNING

To achieve self-evolution, agents need to explore environments, receive feedback, and improve themselves based on the feedback, as humans do (Standish, 2003; Taylor et al., 2016; Fan et al., 2022; Team et al., 2024). More importantly, the agents may face previously unseen tasks and instructions during exploration. Standard reinforcement learning (RL) algorithms, such as Proximal Policy Optimization (PPO) (Schulman et al., 2017), are worth considering due to their excellent performance in alignment of LLMs (Bai et al., 2022). However, in the setting of multi-turn decision-making for LLM-based agents, PPO faces significant challenges, e.g., large action space, long interaction chains and sparse reward signals, leading to high computational complexity and training instability (Huang et al., 2022; Zheng et al., 2023b; Xi et al., 2024).

Hence, we draw inspiration from the well-established connection between RL and probabilistic inference, i.e., *RL as Inference* (Dayan & Hinton, 1997; Levine & Koltun, 2013; Neumann, 2011; Rawlik et al., 2013; Abdolmaleki et al., 2018; Singh et al., 2023), and derive the exploration-learning method called AGENTEVOL for evolution in multi-turn decision-making scenarios, which involves agents alternating between exploration and learning across various environments.

**Theoretical foundation: learning from the estimated optimal policy using the classical *RL as Inference* framework.** We view RL as an inference problem within a specific probabilistic model (Dayan & Hinton, 1997). Differing from traditional RL formulations that focus on identifying a trajectory that maximizes the expected reward, inference-based approaches start with an optimal distribution over trajectories. Following previous work (Toussaint, 2009; Levine & Koltun, 2013; Abdolmaleki et al., 2018), we initially define $P(O = 1)$ to represent the event of "obtained optimal policy by maximum expected rewards" or "achieving success in the RL task", which can be calculated by integrating the optimal policy probability at each sampling point. Given the policy agent $\pi_\theta$, the optimal policy can be obtained by maximizing:

$$\log P_{\pi_\theta}(O = 1) = \log \int \pi_\theta(\tau) p(O = 1|\tau) d\tau. \tag{4}$$

However, the above optimization process is difficult to proceed directly due to the fact that LLM-based agents require token-wise feedback to perform gradient updates. In this paper, we alternatively construct the variational lower bound of Eq.4 by introducing an estimation function $q$ on the optimal policy. With Jensen's inequality, we soon have:

$$\log \int \pi_\theta(\tau) p(O = 1|\tau) d\tau = \log \mathbb{E}_{q(\tau)} \left[ \frac{\pi_\theta(\tau)}{q(\tau)} p(O = 1|\tau) \right] \geq \mathbb{E}_q \left[ \log \frac{\pi_\theta(\tau)}{q(\tau)} p(O = 1|\tau) \right]$$
$$= \mathbb{E}_q[\log p(O = 1|\tau)] - \mathrm{KL}[q(\tau)||\pi_\theta(\tau)] = \mathcal{J}(q, \pi_\theta), \tag{5}$$

where $\pi_\theta$ is the trajectory distribution induced by the agent, and $q(\tau)$ is a variational distribution.

Due to the monotonicity of the logarithmic function, by maximizing the lower bound $\mathcal{J}(q, \pi_\theta)$, we can obtain a policy with an expected return higher than before. Generally, our framework can be divided into two steps of loop iteration. The first part of $\mathcal{J}(q, \pi_\theta)$ can be explained as estimating the optimal policy distribution on the sampled trajectories by maximizing the expected reward over the state space. The second part relates to updating the current agent's parameters $\theta$ towards the optimal policy $q$, thus completing the optimization of one single iteration. In analogy to SGD (Robbins & Monro, 1951), the estimation process introduces noise to the policy optimization due to the presence of unseen decision-making trajectories. This error gradually decreases as the optimization proceeds and converges to zero when the current agent becomes optimal (Dayan & Hinton, 1997).

In AGENTEVOL, we refer to the two steps as **Exploration step** and **Learning step** like Singh et al. (2023). Specifically, with current agent parameters $\theta^m$ and the variational distribution $q^m$, at **exploration step**, the estimation of optimal policy $q$ is updated by maximizing the expected reward: $q^{m+1} = \arg\max_q \mathcal{J}(q, \pi_{\theta^m})$. As $\max_q \mathcal{J}(q, \pi_{\theta^m}) = \min_q[\mathrm{KL}(q(\tau)||p(O = 1|\tau)\pi_{\theta^m}(\tau))]$, we have $q^{m+1} \propto p(O = 1|\tau)\pi_{\theta^m}(\tau)$. This step is equivalent to evaluating the likelihood that the samples generated from the current agent's policy achieve best rewards, and observe the returns of $q$ by empirically estimating on a pre-constructed training set. And at **learning step**, we optimize $\mathcal{J}(q^{m+1}, \pi_\theta)$ by updating $\theta$. This process is similar to learning a new distribution sampled from the optimal policy on the original training data. Since the first term of $\mathcal{J}(q^{m+1}, \pi_\theta)$ relates only to $q$ as

---

**Algorithm 1:** AGENTEVOL

---

**Input:** Initialized policy LLM-based agent $\pi_\theta$, environment set $\mathcal{E}$, trajectory subset $\mathcal{D}_s$, full instruction set $\mathcal{Q}$, reward function $r$.

**Procedure** Behavioral cloning:

    Maximize objective $\mathcal{J}_{BC}(\theta) = \mathbb{E}_{(e,u,\tau)\sim\mathcal{D}_s}\Big[\log \pi_\theta(\tau|e,u)\Big]$ to get $\pi_{\theta_{base}}$;

**Procedure** Evolution :

    $\pi_{\theta^1} \leftarrow \pi_{\theta_{base}}$;

    **for** iteration $m = 1$ to $M$ **do**

       // Perform **Exploration Step**

       $\mathcal{D}_m = \bigcup_{e\in\mathcal{E}} \mathcal{D}_m^e$, where $\mathcal{D}_m^e = \{(e, u^j, \tau^j)\,|u^j \sim \mathcal{Q}_e, \tau^j \sim \pi_{\theta^m}(\tau|e,u^j)\}_{j=1}^{|\mathcal{D}_m^e|}$;

       Compute reward for $\mathcal{D}_m$ with $r$;

       $\mathcal{D}_m \leftarrow \mathcal{D}_m \cup \mathcal{D}_s$;

       // Perform **Learning Step**

       Maximize objective $\mathcal{J}_{Evol}(\theta) = \mathbb{E}_{(e,u,\tau)\sim\mathcal{D}_m}[r(e,u,\tau)\log\pi_\theta(\tau|e,u)]$ to get $\pi_{\theta^{m+1}}$;

    **end**

---

well as $\tau$, the training objective is equivalent to measuring the KL divergence between the estimated policy $q^{m+1}(\cdot)$ and the current policy $\pi_\theta(\cdot)$ over all training samples. We finally derive:

$$\theta^{m+1} := \arg\min_\theta \text{KL}[q^{m+1}(\tau)\|\pi_\theta(\tau)] = \arg\min_\theta \sum_\tau -q^{m+1}(\tau)\log\pi_\theta(\tau). \tag{6}$$

This involves optimizing a weighted negative log-likelihood function based on $q^{m+1}$, which adjusts the agent policy to increase the likelihood of generating higher-reward trajectories, thereby improving the agent's performance.

**Practical implementation for evolving LLM-based agents.** In our LLM-based agent setting, the trajectory is conditioned on the environment $e$ and instruction $u$. Considering our non-negative reward function $r(e,u,\tau)$, we can get $P(O|e,u,\tau) \propto r(e,u,\tau)$ (Singh et al., 2023). Consequently, $q^{m+1}(\tau|e,u) \propto r(e,u,\tau) \cdot \pi_{\theta^m}(\tau|e,u)$. Thus, the policy update in the learning is:

$$\theta^{m+1} := \arg\min_\theta \sum_\tau -(r(e,u,\tau) \cdot \pi_\theta(\tau|e,u))\log\pi_\theta(\tau|e,u)$$
$$= \arg\max_\theta \mathbb{E}_{e\in\mathcal{E},u\sim\mathcal{Q}_e,\tau\sim\pi_{\theta^m}(\tau|e,u)}[r(e,u,\tau)\log\pi_\theta(\tau|e,u)]. \tag{7}$$

This can be viewed as a supervised fine-tuning objective weighted by reward. This approach uses the fixed policy agent from the previous iteration to sample data, thereby separating data collection and policy optimization. This decoupled process can improve the training stability (Zelikman et al., 2022; Singh et al., 2023).

Now we describe the two steps of evolution part in AGENTEVOL in practice:

**Exploration Step.** In the $m$-th exploring iteration, for each environment $e$, we have an instruction set $\mathcal{Q}_e$ which is larger than that used in the BC phase, allowing us to investigate agents evolving to unseen tasks and instructions. The current policy agent interacts with this environment, generating a collection of interaction trajectories $\mathcal{D}_m^e = \{(e, u^j, \tau^j)\,|u^j \sim \mathcal{Q}_e, \tau^j \sim \pi_{\theta^m}(\tau|e,u^j)\}_{j=1}^{|\mathcal{D}_m^e|}$. Subsequently, based on the reward function of the environment, we calculate the reward $r(e,u,\tau)$ for each trajectory. The generated dataset from each environment is then merged, resulting in $\mathcal{D}_m = \bigcup_{e\in\mathcal{E}} \mathcal{D}_m^e$. Note that we also include the original trajectory set in Section 3.2 for the learning step, i.e., $\mathcal{D}_m = \mathcal{D}_m \bigcup \mathcal{D}_s$.

**Learning Step.** In the $m$-th learning iteration, we utilize the dataset $\mathcal{D}_m$ obtained from the exploration step to fine-tune the agent with the objective $\mathcal{J}_{Evol}(\theta) = \mathbb{E}_{(e,u,\tau)\sim\mathcal{D}_m}[r(e,u,\tau)\log\pi_\theta(\tau|e,u)]$ to get $\pi_{\theta^{m+1}}$. We also include the general domain dataset as in the BC phase. We optimize the initial agent $\pi_\theta$ at each iteration, aiming to minimize overfitting and prevent drift from the base agent. In this learning step, the agent is improved, similar to previous work done on LLM reasoning Zelikman et al. (2022); Singh et al. (2023); Aksitov et al. (2023).

By alternating between the two steps, empirical results show that the method facilitates the evolution of an LLM-based agent across both seen and unseen tasks and instructions.

Table 3: Evaluating results on diverse tasks. $BC_{base}$ means the agent trained with AGENTTRAJ, providing a base agent with basic ability and prior knowledge. $BC_{large}$ means the agent that performs BC on AGENTTRAJ-L, representing the performance upper limit of BC in this paper. Our evolution method, AGENTEVOL, outperforms $BC_{large}$ on most tasks and environments through exploration and learning. The best performance of each part is highlighted in **bold**.

| Method | WS | ALF | TC | Sci | Baby | MZ | WD | WT | MV | TL | BD |
|---|---|---|---|---|---|---|---|---|---|---|---|
| **Close-sourced Models & Agents** | | | | | | | | | | | |
| DeepSeek-Chat | 11.00 | 51.00 | 23.00 | **16.80** | 45.67 | 4.00 | 24.00 | 70.00 | 70.00 | 75.00 | 13.50 |
| Claude-3-Haiku | 5.50 | 0.00 | 0.00 | 0.83 | 1.93 | 4.00 | 16.00 | 55.00 | 50.00 | 65.00 | 13.50 |
| Claude-3-Sonnet | 1.50 | 13.00 | 38.00 | 2.78 | **79.25** | 0.00 | 36.00 | 65.00 | 80.00 | 80.00 | **17.00** |
| GPT-3.5-Turbo | 12.50 | 26.00 | 47.00 | 7.64 | 71.36 | 4.00 | 20.00 | 25.00 | 70.00 | 40.00 | 12.50 |
| GPT-4-Turbo | **15.50** | **67.50** | **77.00** | 14.38 | 72.83 | **68.00** | **88.00** | **80.00** | **95.00** | **95.00** | 16.00 |
| **Open-sourced Models & Agents** | | | | | | | | | | | |
| Llama2-Chat-7B | 0.50 | 2.00 | 0.00 | 0.83 | 0.23 | 0.00 | 0.00 | 0.00 | 0.00 | 0.00 | 1.50 |
| Llama2-Chat-13B | 1.00 | 3.50 | 0.00 | 0.83 | 0.10 | 0.00 | 0.00 | 0.00 | 0.00 | 0.00 | 1.50 |
| AgentLM-7B | 36.50 | 71.00 | **4.00** | 1.63 | 0.49 | **12.00** | 4.00 | 0.00 | **5.00** | 15.00 | 5.00 |
| AgentLM-13B | 39.50 | **73.00** | 0.00 | 2.75 | 0.45 | 8.00 | 0.00 | **10.00** | **5.00** | 5.00 | 3.00 |
| AgentLM-70B | **49.50** | 67.00 | **4.00** | **10.68** | **0.66** | 8.00 | 4.00 | 0.00 | 0.00 | **40.00** | **7.50** |
| **Ours** | | | | | | | | | | | |
| $BC_{base}$ | 66.50 | 77.50 | 44.00 | 26.42 | 69.31 | **12.00** | 12.00 | 25.00 | 5.00 | 45.00 | 8.00 |
| $BC_{large}$ | 73.50 | 83.00 | 60.00 | **74.47** | 74.19 | **12.00** | **36.00** | **45.00** | 5.00 | 65.00 | 8.50 |
| **AGENTEVOL** | **76.50** | **88.00** | **64.00** | 38.00 | **82.70** | **12.00** | 12.00 | 25.00 | **60.00** | **70.00** | **9.00** |

## 4 EXPERIMENTS AND DISCUSSION

### 4.1 EXPERIMENTAL SETUP

**Environments and Tasks.** We explore the self-evolution of LLM-based agents with the AGENT-GYM framework. Main experiments cover the following environments: WS, ALF, Sci, Baby, TC, BD, MZ, WD, TL, WT, and MV. Note that instructions used in BC are fewer than those in evolution, to study the agent's ability to generalize when performing exploration.

**Baselines.** We include closed-source models like GPT-3.5-Turbo (Ouyang et al., 2022), GPT-4-Turbo (OpenAI, 2023), Claude 3 (Anthropic, 2024), and DeepSeek-Chat (DeepSeek-AI, 2024). We also include open-source models like Llama-2-Chat (Touvron et al., 2023), and agents trained on expert trajectories, i.e., AgentLM (Zeng et al., 2023). For a fair comparison, we include a baseline that performs BC on AGENTTRAJ-L, serving as the maximum performance achievable through BC.

**Implementation Details.** Experiments are conducted with eight A100-80GB GPUs. Our main backbone model is Llama-2-Chat-7B. Different environment services are deployed on different ports of the same server. We set the iteration number $M$ to 4. Each instruction is sampled once during the evolution process for efficiency. Note that some environments provide dense rewards $r \in [0, 1]$, while others give only binary feedback $r \in \{0, 1\}$. For simplicity and consistency, we follow previous work (Singh et al., 2023) and use binary rewards. We set $r = 0$ for trajectories where $r < 1$, while for those with $r = 1$, we keep it unchanged. See Appendix E for more implementation details. Detailed prompts for each environment are in Appendix G.

### 4.2 MAIN RESULTS

Experiment results in Table 3 demonstrate that: (1) While closed-source models perform well, even SOTA closed-source models like GPT-4-Turbo fail to achieve satisfactory performance on all tasks, highlighting the need for developing more capable agents. (2) Open-source models, represented by Llama2-Chat, perform poorly on all tasks, highlighting the importance of the initialization step of BC. (3) Models trained on agent trajectories, like AgentLM (Zeng et al., 2023), can perform on par with GPT-4-Turbo on many tasks, particularly the 70B version. However, they do not match performance on tasks like TextCraft (Prasad et al., 2023) or SciWorld (Wang et al., 2022), which can be attributed to the lack of training data. (4) The agent trained on AGENTTRAJ-L ,i.e., $BC_{large}$, achieves excellent performance, matching or even surpassing SOTA models, showing that it is a strong baseline. (5) AGENTEVOL, despite having limited trajectories for imitation, surpasses $BC_{large}$ and SOTA models on many tasks like WebShop (Yao et al., 2022), ALFWorld (Shridhar et al., 2021) and BabyAI (Chevalier-Boisvert et al., 2019), validating the superiority and promise of agent evolution.

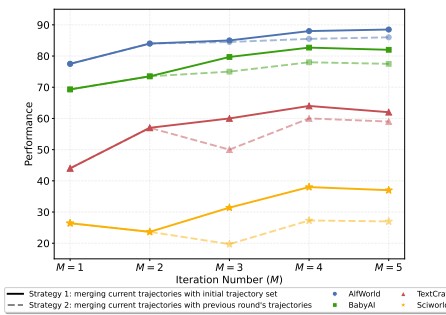 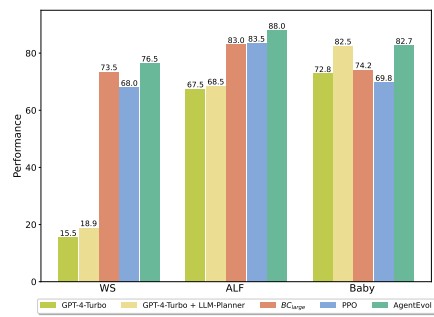

Figure 3: Ablation on data merging strategies and iteration number $M$.

Figure 4: Comparison with other exploration-based methods.

Moreover, we report the number of interactive rounds required by different models to solve the task, in order to demonstrate the efficiency of AGENTEVOL (Appendix F.1). We also conduct experiments to explore evolution in isolated environments (Appendix F.4) and perform case study (Appendix F.6).

### 4.3 DISCUSSION & ANALYSIS

**Ablation on data merging strategies and iteration number $M$.** In our experiments, we merge the trajectories sampled during each iteration with the initial trajectories for training, rather than merging it with the trajectories generated in the previous iteration. Here, we conduct an ablation study to show the impact of this merging strategy and the iteration number $M$. Experimental results in Figure 3 show that merging with the initial data provides more stable improvements, while merging with the trajectories from the previous iteration leads to performance fluctuations, possibly due to overfitting (Yuan et al., 2023b; Singh et al., 2023). Additionally, as $M$ increases, performance improves but gradually converges in later iterations. So we choose $M = 4$ to balance performance and efficiency. More ablation studies can be found in Appendix F.2.

**Comparison with other exploration-based methods.** The AGENTEVOL method explores and learns through interactions with environments. For comparison, we include two additional methods: online RL method PPO (Schulman et al., 2017) and prompt-based method LLM-Planner (Song et al., 2023). PPO employs the same backbone as AGENTEVOL, Llama-2-Chat-7B, whereas LLM-Planner is implemented with GPT-4-Turbo to showcase its effectiveness. We run PPO in the isolated environment, as it tends to experience training instability and show poorer performance when applied across multiple environments. Results in Figure 4 show that (1) LLM-Planner outperforms GPT-4-Turbo but falls short on WS, and overall, it doesn't surpass AGENTEVOL. (2) PPO performs well but lags behind AGENTEVOL by a significant margin. These findings indicate that AGENTEVOL consistently improves performance for smaller open-source models compared to larger SOTA models.

**Evolution with both successful and failed trajectories.** In the learning step, we only utilize the sampled trajectories with high rewards (success) and do not use failed trajectories. Inspired by previous work (Mitra et al., 2024; Hosseini et al., 2024; Song et al., 2024; Yang et al., 2024; Wang et al., 2023a), we explore whether failed trajectories can be included for better evolution. Specifically, we construct pairs of successful and failed trajectories and optimize the agent using the DPO method (Rafailov et al., 2023), which fits models to the pair-wise dataset (Wang et al., 2023a; Mitra et al., 2024; Lai et al., 2024). Results in Table 4 show that using both types of trajectories can still bring about evolutionary effects, but the performance is not as good as our method, indicating that preference optimization in multi-task setting is more challenging compared to single-task (Mitra et al., 2024; Song et al., 2024). In the future, we hope to explore more advanced algorithms and methods to make full use of all trajectories for comprehensive evolution.

**Effectiveness on different models.** To demonstrate the generalizability of our method across different backbone models, we conduct experiments on Llama-2-13B (Touvron et al., 2023) and DeepSeek-Coder-1.3B (Guo et al., 2024). The experimental results in Table 5 show that our AGENTEVOL maintains its evolutionary capabilities across different backbone models, achieving performance that is comparable to or surpasses $BC_{large}$.

Table 4: Experiments on evolution with both successful and failed trajectories.

| Method | WS | ALF | Baby | TC |
|---|---|---|---|---|
| $BC_{base}$ | 66.5 | 77.5 | 69.3 | 44.0 |
| AGENTEVOL | **77.0** | **88.0** | **82.9** | **65.0** |
| DPO with failed traj | 75.0 | 86.5 | 78.3 | 58.0 |

Table 5: Effectiveness of AGENTEVOL on different models.

| Model | Method | WS | ALF | Baby | TC |
|---|---|---|---|---|---|
| DeepSeek-Coder-1.3B | $BC_{base}$ | 54.0 | 33.0 | 68.9 | 31.0 |
| | $BC_{large}$ | 65.0 | **62.5** | 73.8 | 37.0 |
| | AGENTEVOL | **67.5** | 54.5 | **77.3** | **38.0** |
| Llama2-Chat-13B | $BC_{base}$ | 65.5 | 81.5 | 76.6 | 59.0 |
| | $BC_{large}$ | 74.0 | 85.0 | 81.1 | 61.0 |
| | AGENTEVOL | **78.5** | **89.5** | **86.8** | **71.0** |

## 5 RELATED WORK

With the development of LLMs (OpenAI, 2023; Anil et al., 2023), developing agents based on them has become an important research direction (Xi et al., 2023; Wang et al., 2024b). These agents are typically designed to perform multi-turn decision-making tasks (Yao et al., 2023; Aksitov et al., 2023; Chen et al., 2023). To evaluate these agents, researchers have proposed various benchmarks (Yao et al., 2022; Liu et al., 2023a; Ma et al., 2024; Zhou et al., 2023b). Our benchmark AGENTE-VAL covers more diverse scenarios and environments, providing a more comprehensive evaluation.

Closed-source LLMs, equipped with prompting methods like ReAct (Yao et al., 2023) and PlanAct (Liu et al., 2023b), can achieve great performance in agent tasks, while agents based on open-source methods perform poorly on these tasks (Liu et al., 2023a; Christianos et al., 2023). To address this challenge, a series of work collects expert trajectories from diverse environments and tasks and trains LLM-based agents through behavioral cloning (Zeng et al., 2023; Chen et al., 2023; 2024; Zhang et al., 2024). However, obtaining these expert trajectories is often costly and they lack sufficient exploration of the environment (Yang et al., 2024; Aksitov et al., 2023).

Another line of work trains LLM-based agents based on environmental feedback, referred to as interactive learning methods (Zhou et al., 2024; Christianos et al., 2023; Song et al., 2024; Abdulhai et al., 2023a). Specifically, they involve training LLMs or agents through exploration and learning. As a representative method, RL has succeeded in LLM alignment (Askell et al., 2021; Bai et al., 2022; Ouyang et al., 2022; Zheng et al., 2023b; Wang et al., 2024a), and has been introduced to reasoning and agent tasks, achieving excellent results (Xi et al., 2024; Luong et al., 2024; Zhou et al., 2024; Christianos et al., 2023). However, in our multi-environment scenarios, reward consistency and training stability can become problematic (Zhou et al., 2024; Song et al., 2024; Cao et al., 2024). Another line of work uses self-evolution/self-improvement, where the model explores the environment to obtain high-reward trajectories and fine-tunes itself based on these trajectories, achieving promising performance in reasoning, coding, and web tasks (Gülçehre et al., 2023; Singh et al., 2023; Zelikman et al., 2022; Yuan et al., 2023a; Aksitov et al., 2023; Yang et al., 2024; Song et al., 2024; Tao et al., 2024; Tian et al., 2024; Lai et al., 2024). However, like RL-based methods, these works only explore within a single environment. With AGENTGYM, our work explores agent evolution using AGENTEVOL method, conducting exploration across multiple environments.

## 6 CONCLUSION

In this work, we present a new framework named AGENTGYM that encompasses 14 interactive environments and 89 tasks, covering 7 key scenarios for agent evaluation and development. It also includes expanded instructions, a comprehensive benchmark named AGENTEVAL, and trajectory sets called AGENTTRAJ and AGENTTRAJ-L. Additionally, we derive and implement an exploration-learning method AGENTEVOL to investigate the self-evolution of LLM-based agents across multiple environments. Empirical results demonstrate the effectiveness of the framework and the method. We also perform sufficient ablation and analysis to investigate how the method works. We hope our work can help the AI community develop more advanced generalist LLM-based agents.

### ETHICS STATEMENT

AGENTGYM and AGENTEVOL facilitate the self-evolution of generally-capable agents, and our focus is on the self-evolution of decision-making capabilities, but it is crucial to consider safety and ethical issues during usage. Agents must not be allowed to violate human values. Therefore, it is

essential to strengthen supervision and regulation when eliciting agents' self-evolution capabilities. In the future, we hope to improve the framework's functionality to align agents with human values.

## REPRODUCIBILITY STATEMENT

We upload anonymized versions of our data and code in a Zip file with a Readme file. We have listed our implementation details in Section 4.1 and Appendix E. We will release the code, dataset, benchmark, and checkpoints.

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

## A  LIMITATIONS

This paper proposes a new framework named AGENTGYM. It includes an interactive platform with diverse environments and tasks, an agent benchmark AGENTEVAL, and two collections of expert trajectories AGENTTRAJ and AGENTTRAJ-L. Additionally, we introduce a novel algorithm, AGENTEVOL, to explore the evolutionary capabilities of generally-capable LLM-based agents. Despite the contributions and the fact that our method performs well, our work still has some limitations. **Firstly**, for computational efficiency, we do not perform multiple samplings in each iteration. However, in the analysis in Section 4.3, we find that more sampling leads to better results, although the improvement is not significant. In the future, we hope to increase the number of samples $K$ to a larger value when sufficient computational resources are available, to explore the upper limits of our method. **Secondly**, although we validate the effectiveness of our method on three different models (Llama2-Chat-7B, Llama-2-Chat-13B, and DeepSeek-Coder-1.3B), we hope to verify it on stronger and larger base models in the future to explore the potential for building more generally-capable agents.

## B  DETAILS OF ENVIRONMENTS IN AGENTGYM

**WebShop (WS) (Yao et al., 2022).**  WebShop is an interactive web environment for web shopping. The agents are given instructions, and need to buy a product that matches the specifications. The agents can click a button on the webpage or search for something by the search engine. WebShop contains 12k instructions and provides over one million real products from amazon.com. We select 6910 instructions. For AGENTTRAJ, we collect 1000 trajectories with SOTA models (700) and human annotations (300). For AGENTTRAJ-L, we collect 3930 trajectories with SOTA models (3430) and human annotations (500). We take the success rate as the evaluation metric and set the maximum round to 10.[2]

**WebArena (WA) (Zhou et al., 2023a).**  WebArena is a realistic and reproducible web environment. It contains four categories: E-commerce platforms, social forum platforms, collaborative development platforms, and content management systems. It supports 12 different web browsing actions. The observation space consists of a web page URL, the opened tabs, and the web page content. Completing tasks in this highly realistic environment requires the agent to possess strong memory, high-level planning, common sense, and reasoning abilities. The reward from the environment is consistent with the original paper. We filter 20 evaluating test instances from the original dataset for three main sub-tasks: Information-seeking, Site Navigation, and Content & configuration operation. We take the success rate as the evaluation metric and set the maximum round to 25.[3]

**MAZE (MZ) (Abdulhai et al., 2023b).**  MAZE is a word game. Agents, acting as players, can know their own position, the goal position, and the directions where there are walls around them. Agents decide to move one square in one of four directions (up, down, left, or right) each time, receiving a reward of -1 for every move until they reach the goal position. We use GPT-4-Turbo to add thoughts to the trajectories sampled by LMRL-Gym and create our dataset. For AGENTTRAJ, we include 100 trajectories. For AGENTTRAJ-L, we include 215 trajectories. We take the success rate as the evaluation metric and set the maximum round to 15.[4]

**Wordle (WD) (Abdulhai et al., 2023b).**  Wordle is a word-guessing game that tests agents' ability to reason at the level of individual letters. Agents guess the target word from a given vocabulary containing some five-letter words. After each guess, agents are told whether each letter in the guessed word is in the target word and whether its position is correct and receive a reward of -1 for each step until they guess the target word or run out of attempts. We take the success rate as the evaluation metric and set the maximum round to 8. We also use GPT-4-Turbo to add thoughts to the trajectories sampled by LMRL-Gym. For AGENTTRAJ, we include 500 trajectories. For AGENTTRAJ-L, we include 955 trajectories.

---

[2]https://github.com/princeton-nlp/WebShop/blob/master/LICENSE.md

[3]https://github.com/web-arena-x/webarena/blob/main/LICENSE

[4]https://github.com/abdulhaim/LMRL-Gym/blob/main/LICENSE

**ALFWorld (ALF) (Shridhar et al., 2021).** ALFWorld is a household environment based on TextWorld, where agents need to explore rooms and use common sense reasoning to execute tasks. The action space of ALFWorld includes picking up and placing items, observing surroundings, using furniture, and more. The environment provides feedback on the execution of actions based on predefined logic. We take the success rate as the evaluation metric and set the maximum round to 30. ALFWorld has six types of tasks. We get 3827 instructions from the original work. For AGENTTRAJ, we collect 500 trajectories with SOTA models(400) and human annotations (100). For AGENTTRAJ-L, we collect 2420 trajectories with SOTA models(1920) and human annotations (500). [5]

**SciWorld (Sci) (Wang et al., 2022).** ScienceWorld is a benchmark for testing agents' scientific reasoning abilities in a new interactive text environment at the standard elementary science curriculum level. ScienceWorld includes 30 types of tasks, such as using measurement instruments and conducting mechanics experiments. Its action space is task-related, with the environment simulator providing the effects of actions. Because the ScienceWorld repository provides golden paths and existing models cannot achieve high performance, we use GPT-4-Turbo to generate thoughts for golden paths of 22 types of interactions that are not too long. For AGENTTRAJ, we include 1000 trajectories. For AGENTTRAJ-L, we include 2120 trajectories. We take reward as the evaluation metric and set the maximum round to 30.[6]

**BabyAI (Baby) (Chevalier-Boisvert et al., 2019).** The BabyAI platform is an interactive grid world simulator with 40 instruction-following tasks where the agent is asked to interact with objects. The agent has a limited 7x7 sight of view and can only operate objects in front. The original implementation of BabyAI presents observations in the form of images and low-level actions like "move forward" and "turn left". The implementation from AgentBoard converts graphic observations into textual instructions and expands the action space with high-level actions like "pickup green key 1" and"go through blue locked door 2". The agent receives a non-zero reward discounted by the number of steps when reaching the goal, and 0 otherwise. For AGENTTRAJ, we annotate 400 trajectories of 18 out of all 40 tasks with SOTA models. For AGENTTRAJ-L, we annotate 810 trajectories with SOTA models. We take reward as the evaluation metric and set the maximum round to 20.[7]

**TextCraft (TC) (Prasad et al., 2023).** Similar to WordCraft, TextCraft is a text-only environment for crafting Minecraft items. This environment constructs a crafting tree based on Minecraft's crafting recipes, comprising 544 nodes, each representing a target item. In TextCraft, each task provides a specific target item alongside a list of crafting commands generated by the tree. These tasks are structured compositionally, incorporating crafting recipes of varying complexity ranging from 1 to 4 steps. The environment supports three valid actions: craft <item> using <ingredients>, get <item>, and inventory. Each round, the environment checks the agent's actions and returns the execution state. Apart from craftable items and their ingredients, all other items are obtainable from the environment. Agents can get a reward of 1 only upon successfully crafting the target item. We select 100 tasks for the test set and use the remaining tasks for training. For AGENTTRAJ, we annotate 300 trajectories with SOTA models (254) and human annotation (46), with every action in the trajectories verified by the environment. For AGENTTRAJ-L, we annotate 374 trajectories with SOTA models (299) and human annotation (75). We take the success rate as the evaluation metric and set the maximum round to 20.[8]

**Weather (WT) (Ma et al., 2024).** The Weather Environment allows LLM agents to utilize a weather tool to access data on temperature, precipitation, and air quality for various locations and time periods. It includes 18 different actions that agents can use to achieve weather-related objectives. This environment leverages Python code to integrate the Open-Meteo API and implement the necessary functions. If the agent's final answer matches the reference answer, it receives a reward of 1; otherwise, it receives a reward of 0. We expand the original dataset of 20 queries to a total of 331

---

[5]https://github.com/alfworld/alfworld/blob/master/LICENSE

[6]https://github.com/allenai/ScienceWorld/blob/main/LICENSE

[7]https://github.com/mila-iqia/babyai/blob/master/LICENSE

[8]https://github.com/archiki/ADaPT/blob/main/LICENSE

queries by using GPT-3.5-Turbo and GPT-4-Turbo for augmentation using self-instruct and instruction evolution. Finally, we select 20 questions as the evaluating set, leaving the remaining questions as the training set. For AGENTTRAJ, we annotate 160 trajectories with SOTA models (140) and human annotators (20). We also refine the annotations with human review to ensure accuracy. For AGENTTRAJ-L, we annotate 311 trajectories with SOTA models (230) and human annotators (81). We take the success rate as the evaluation metric and set the maximum round to 10.[9]

**Movie (MV) (Ma et al., 2024).**  The Movie Environment grants LLM agents to utilize the movie tool for accessing cinematic data, including film details, personnel, and production companies. It offers 16 distinct actions that agents can use to achieve various movie-related objectives. This tool integrates the API and data from The Movie Database, implementing the necessary functions to establish its capabilities. If the agent's final answer matches the reference answer, it receives a reward of 1; otherwise, it receives a reward of 0. To enhance the dataset, we expand the original 20 questions to 235 by using GPT-3.5-Turbo and GPT-4-Turbo for query augmentation. GPT-4-Turbo is employed to annotate 100 trajectories in AGENTTRAJ, and the annotations are further corrected through human annotations to ensure accuracy. We also use GPT-4-Turbo to annotate 215 trajectories for AGENTTRAJ-L. We select 20 questions for the evaluating set, with the remaining questions designated as the training set. We take the success rate as the evaluation metric and set the maximum round to 12.

**Academia (AM) (Ma et al., 2024).**  The Academia Environment equips LLM agents with the academic tools to query information related to computer science research, including academic papers and author details. It offers 7 different actions for achieving various academic research objectives. During its development, it utilizes data from the Citation Network Dataset, crafts the necessary functions, and subsequently constructs the Academia tool. If the agent's final answer matches the reference answer, it receives a reward of 1; otherwise, it receives a reward of 0. The original 20 questions are used as the evaluating set. We take the success rate as the evaluation metric and set the maximum round to 12.

**TODOList (TL) (Ma et al., 2024).**  The TodoEnvironment enables LLM agents to query and amend personal agenda data through the todo tool, offering 11 different actions. This tool is implemented based on the TodoList API. If the agent's final answer or operations matches the reference ones, it receives a reward of 1; otherwise, it receives a reward of 0. To enhance the dataset, we expand the original 20 questions to 155 using GPT-3.5-Turbo and GPT-4-Turbo for data augmentation. For AGENTTRAJ, we annotate 70 trajectories with GPT-4-Turbo. For AGENTTRAJ-L, we annotate the queries to get 135 trajectories with GPT-4-Turbo (96) and human annotators (39). The annotations are further refined by human review to ensure accuracy. Finally, we select 20 questions for the evaluating set, with the remaining questions designated as the training set. We take the success rate as the evaluation metric and set the maximum round to 15.

**Sheet (ST) (Ma et al., 2024).**  The Sheet Environment allows LLM agents to use the sheet tool to access and modify spreadsheet data, providing 20 different actions for operating on an Excel sheet. This tool is built upon the Google Sheets API. The reward returned by the environment is based on the similarity between the table manipulated by the agent and the reference table, with a value range of $[0, 1]$. The original 20 questions are used as the evaluating set. We take reward as the evaluation metric and set the maximum round to 15.

**BIRD (BD) (Zheng et al., 2023a).**  Code ability is a crucial aspect of capability for LLM-based agents. In this environment, we focus on database management ability. We wrap the BIRD-SQL dataset and provide a unified API for agents to interact with. BIRD-SQL is a bench for large-scale database-grounded text-to-SQL evaluation. It requires the agent to query a database using a SELECT statement to get the correct answer. It contains 9428 unique problems with a golden answer for training. We select 3200 of them as the instruction set. For AGENTTRAJ, we employ GPT-4-Turbo to add thoughts for 2000 of the training set problems. For AGENTTRAJ-L, we employ

---

[9]https://github.com/hkust-nlp/AgentBoard. The codebase is licensed under an Apache-2.0 License and the dataset is licensed under a GNU General Public License, version 2.

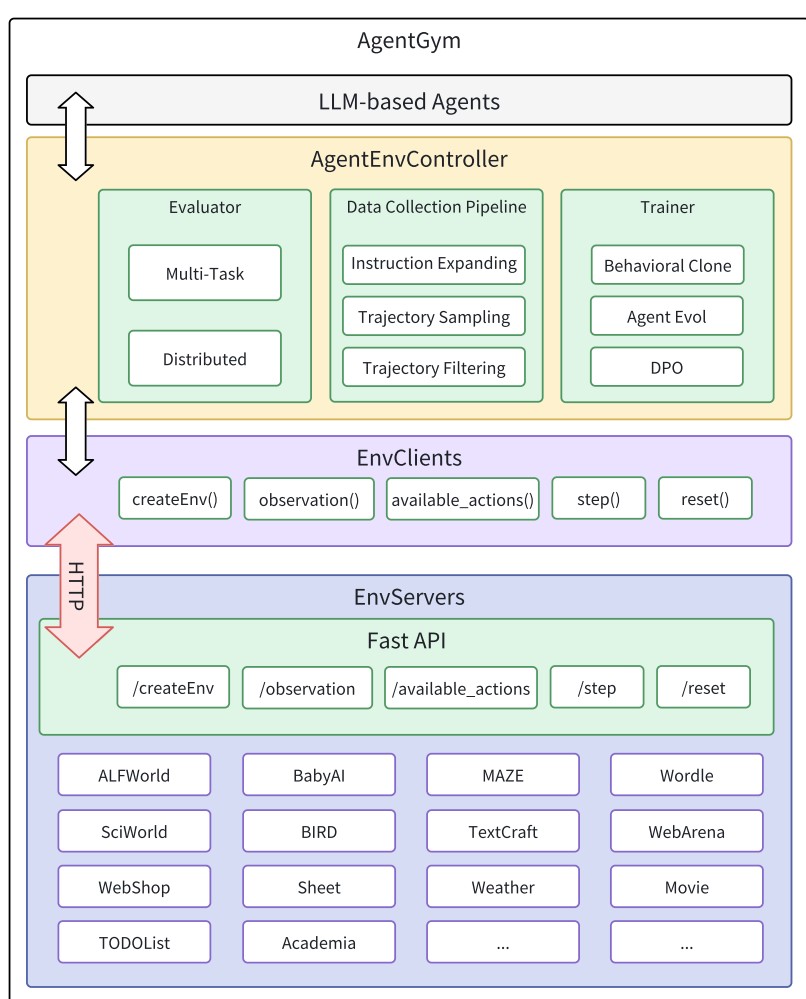

Figure 5: An illustration of the architecture of AGENTGYM.

GPT-4-Turbo to add thoughts for 3000 of the training set problems. We take success rate as the evaluation metric and the maximum round is 1 as BD is a single-round programming task.[10]

## C   FRAMEWORK ARCHITECTURE OF AGENTGYM

We adopt a decoupled architecture to construct the agent-environment interaction framework, as illustrated in Figure 5 in Appendix C. Specifically, recognizing the distinct dependencies present in different environments, AGENTGYM deploys separate services for each environment in a user-friendly manner to prevent conflicts. These environments are implemented to offer standardized and parallelizable functions, such as /createEnv to create an environment, /observation to obtain the current observation from the environment, /available_actions to retrieve the currently available actions, /step to perform an action, and /reset to reset the environment. Clients communicate with the servers via HTTP protocol. At the core of this architecture is the controller, which facilitates interactions between agents and environmental services, providing a unified and encapsulated interface for agents to invoke environmental functions or operations. Additionally, we implement user-friendly components such as the evaluator, trainer, and data collection pipeline to support further development of the community. We have implemented 14 types of environments

---

[10]https://github.com/AlibabaResearch/DAMO-ConvAI/tree/main/bird. The bench is under a CC BY-NC 4.0 License.

and 89 tasks, and developers can easily develop new environments and add them to AGENTGYM by encapsulating the aforementioned interfaces.

## D   DATABASE CONSTRUCTION OF AGENTTRAJ, AGENTTRAJ-L AND AGENTEVAL

**Instruction collection.**   We have gathered $20,494$ instructions across the aforementioned environments using appropriate strategies. *(i)* For environments whose original datasets contain sufficient instructions, we use their original instruction sets or subsets (WA, ALF, Sci, AM, ST, BD). *(ii)* For certain environments, we generate instructions using rule-based automated pipelines (TC, WS, MZ, WD, Baby). For example, in TC, we first construct rule trees for forging different items and generate instructions of varying difficulty (levels 1-4) based on these rules. For WS, we generate instructions based on available products by fixing the random seed. For MZ, we randomly select starting points in the maze and construct instructions accordingly. For WD, we fix the seed and generate words for guessing. For Baby, we pass a fixed seed to the generator provided by the environment to generate instructions. *(iii)* In environments where instructions are relatively scarce and difficult to construct through rules, we use self-instruct (Wang et al., 2023b) and instruction evolution (Xu et al., 2023) methods to generate instructions. These methods provide an LLM (GPT-4-Turbo) with available actions and instruction examples, and query it to generate diverse and challenging instructions that might be needed in real-world scenarios (WT, MV, TL). Note that we manually verify the instructions generated by these AI-based techniques to ensure that they can be successfully completed.

**Benchmark construction.**   We then construct a benchmark suite with a size of $1160$ named AGENTEVAL to holistically evaluate the ability of LLM-based agents in diverse scenarios. Specifically, *(i)* for environments that have different task categories or varying difficulty levels, we either uniformly sample test examples from different subsets or use them all (ALF, Sci, Baby, TC); *(ii)* for remained environments with existing test sets, we use the original test sets or randomly sample from them (WA, AM, ST, BD); *(iii)* for others, we randomly sampled from the collected or augmented instructions (WS, MZ, WD, WT, MV, TL).

**Trajectory collecting and filtering.**   In AGENTGYM, the server provides instructions including task description, environment setup, and problem to the agent. Next, as described in Section 3.1, the agent interacts with the environment in ReAct-Style until the task is completed. We collect 6130 trajectories from 11 environments with different strategies. *(i)* For environments with human annotated trajectories or where the correct action sequences can be obtained using a rule-based solver, we use GPT-4-Turbo to add thought step by step for each action, thus forming outputs in the ReAct-Style (MZ, WD, Sci, BD). *(ii)* For environments where only instructions are provided and the correct trajectories are neither available nor can be derived through rules, we annotate the correct trajectories with SOTA models (e.g., GPT-4-Turbo) and crowdsourcing. Then, we rigorously filter the trajectories based on rewards and correctness to ensure their quality (WS, ALF, Baby, TC, WT, MV, TL). This set, named AGENTTRAJ, is used to train a base agent with preliminary abilities and prior knowledge. For a fair comparison, we perform annotation and filtering on all instructions using the same pipeline and get AGENTTRAJ-L to represent the performance upper bound of BC.

## E   MORE IMPLEMENTATION DETAILS

**Behavior Cloning.**   We train the model for 3 epochs with a learning rate of $1 \times 10^{-5}$. The batch size is set to 2, and gradient accumulation is performed over 2 steps. We do not employ weight decay or learning rate warmup.

**AGENTEVOL.**   First, we train a base agent on the AGENTTRAJ set, running BC for 3 epochs with a learning rate of $1 \times 10^{-5}$. Then, we perform the self-evolution phase. In Learning Step, we run 1 epoch per iteration. In Exploration Step, we set the temperature to 0.7 to sample trajectories across environments. We perform a total of $M$=4 iterations. All other parameters remain the same as in Behavior Cloning. All experiments are conducted on eight A100-80GB GPUs.

**LLM-Planner.**   LLM-Planner is a prompt-based baseline. In our experiments, we enhance the initial System Prompt with exploration-based guidance. We directly prompt the agent to generate

a high-level plan in the first turn, mapping the instruction into subgoals, and let it interact with the environment in a ReAct-style during subsequent turns.

**PPO.** We use full parameter fine-tuning instead of LoRA tuning. We load three models: the actor model, the reference model, and the critic model. We do not use a reward model, as our environment automatically assigns rewards to the agent based on interaction results. We follow the implementation from the TRL library [11], where the actor and critic models share the same backbone. On top of this, we add a trainable value head as the output for the critic model. The learning rate is set to $5 \times 10^{-7}$, with a batch size of 1 and gradient accumulation steps of 2. We do not use weight decay or learning rate warmup. We adhere to OpenAI's implementation of the PPO algorithm Ouyang et al. (2022), where `KL_coef` = 0.01, `gamma` = 1.0, `lambda` = 0.95, and `ppo_epoch` = 2 . For all environments, we first perform supervised fine-tuning for 1∼2 epochs as a warm-up, followed by PPO training for 5∼10 epochs.To alleviate memory constraints, we employ gradient checkpointing and flash-attention 2 techniques.

**DPO.** We also use full parameter fine-tuning. During the data sampling phase, we perform two rounds of sampling on the base model to construct DPO training data pairs. Responses with a reward exceeding the expert threshold are labeled as "chosen responses". Responses with a reward gap greater than that of the "chosen responses" are labeled as "rejected responses", and these form the data pairs. In our experiments, the expert threshold is set to 0.9, and the reward gap is set to 0.1. We train for 3 epochs with a learning rate of $5 \times 10^{-7}$, a batch size of 2, gradient accumulation steps of 4, weight decay of 0.1, and a warmup ratio of 0.1. Additionally, we include a BC objective to stabilize the training procedure, following previous work (Lai et al., 2024). Both the DPO and BC objectives are assigned equal weights.

**Evaluation.** We set `do_sample = False` during evaluation. When evaluating models that have not been fine-tuned on expert trajectories, we use a few-shot approach; when evaluating models that have been trained on expert trajectories, we use a zero-shot approach.

We upload anonymized versions of our data and code in a Zip file.

# F MORE EXPERIMENTS

## F.1 INTERACTIVE ROUNDS IN MAIN EXPERIMENTS

Interactive rounds reflect the efficiency of an agent in solving tasks. Table 6 shows the interactive rounds of each model/agent across tasks. We also present the evaluation performance in Table 6 for better and clearer illustration. We find that agents trained with AGENTTRAJ-L and AGENTEVOL both demonstrate high efficiency, indicating that they can complete tasks in a small number of rounds. Additionally, we observe a trend: agents that require fewer interactive rounds to complete the same task generally perform better. This may be because underperforming agents often struggle to find the optimal path to achieve the final goal or exceed the maximum number of rounds. For example, in ALFWorld and BabyAI, AGENTEVOL achieves the best performance as well as the fewest interactive rounds.

## F.2 MORE ABLATION STUDIES

**Ablation on sample number $K$.** In the exploration step, we perform sampling on each instruction once per iteration. Here, we conduct ablation on sample number $K$ with four tasks. The results in Table 7 show no significant performance increases with higher $K$. So we select $K = 1$ for computational efficiency.

**Ablation on exploration scope.** In our experiment, we first train a base agent using $\mathcal{D}_s$ and then let it explore a wider range of instructions and tasks. We conduct an ablation study on four tasks to see how well the agent evolves with limited instructions as in the BC phase. Table 7 shows that even in a limited scope, the base agent's performance improves, which may be attributed to more

---

[11]https://github.com/huggingface/trl/blob/main/examples/scripts/ppo/ppo.py

Table 6: Evaluating performance and interactive rounds on diverse tasks. The first row of each method indicates performance, while the second row of each method shows the number of interaction rounds between the model/agent and the environment.

| Method | WS | ALF | TC | Sci | Baby | MZ | WD | WT | MV | TL | BD |
|---|---|---|---|---|---|---|---|---|---|---|---|
| **Close-sourced Models & Agents** | | | | | | | | | | | |
| DeepSeek-Chat | 11.00 | 51.00 | 23.00 | **16.80** | 45.67 | 4.00 | 24.00 | 70.00 | 70.00 | 75.00 | 13.50 |
| | 6.9 | 20.4 | 15.1 | 20.7 | 11.7 | 14.5 | 5.2 | 6.1 | 5.9 | 4.4 | 1.0 |
| Claude-3-Haiku | 5.50 | 0.00 | 0.00 | 0.83 | 1.93 | 4.00 | 16.00 | 55.00 | 50.00 | 65.00 | 13.50 |
| | 8.0 | 30.0 | 20.0 | 29.8 | 19.9 | 14.4 | 5.7 | 7.3 | 6.0 | 4.0 | 1.0 |
| Claude-3-Sonnet | 1.50 | 13.00 | 38.00 | 2.78 | **79.25** | 0.00 | 36.00 | 65.00 | 80.00 | 80.00 | **17.00** |
| | 9.5 | 27.9 | 14.6 | 28.7 | 6.6 | 15.0 | 5.2 | 6.9 | 5.1 | 4.5 | 1.0 |
| GPT-3.5-Turbo | 12.50 | 26.00 | 47.00 | 7.64 | 71.36 | 4.00 | 20.00 | 25.00 | 70.00 | 40.00 | 12.50 |
| | 4.9 | 25.2 | 13.1 | 16.5 | 8.4 | 14.4 | 5.3 | 6.6 | 4.6 | 4.6 | 1.0 |
| GPT-4-Turbo | **15.50** | 67.50 | 77.00 | 14.38 | 72.93 | 68.00 | 88.00 | 80.00 | 95.00 | 95.00 | 16.00 |
| | 8.2 | 18.3 | 9.9 | 18.1 | 9.1 | 9.0 | 4.0 | 6.0 | 4.5 | 4.0 | 1.0 |
| **Open-sourced Models & Agents** | | | | | | | | | | | |
| Llama2-Chat-7B | 0.50 | 2.00 | 0.00 | 0.83 | 0.23 | 0.00 | 0.00 | 0.00 | 0.00 | 0.00 | 1.50 |
| | 6.4 | 22.6 | 14.5 | 27.5 | 9.5 | 15.0 | 6.0 | 9.9 | 12.0 | 15.0 | 1.0 |
| Llama2-Chat-13B | 1.00 | 3.50 | 0.00 | 0.83 | 0.10 | 0.00 | 0.00 | 0.00 | 0.00 | 0.00 | 1.50 |
| | 8.1 | 19.6 | 16.5 | 21.3 | 10.9 | 13.4 | 6.0 | 10.0 | 12.0 | 15.0 | 1.0 |
| AgentLM-7B | 36.50 | 71.00 | **4.00** | 1.63 | 0.49 | **12.00** | 4.00 | 0.00 | **5.00** | 15.00 | 5.00 |
| | 4.7 | 17.7 | 19.4 | 28.5 | 7.5 | 13.9 | 2.0 | 8.3 | 11.7 | 10.6 | 1.0 |
| AgentLM-13B | 39.50 | **73.00** | 0.00 | 2.75 | 0.45 | 8.00 | 0.00 | **10.00** | **5.00** | 5.00 | 3.00 |
| | 4.8 | 17.8 | 19.4 | 28.5 | 7.6 | 13.9 | 6.0 | 6.6 | 10.7 | 8.4 | 1.0 |
| AgentLM-70B | **49.50** | 67.00 | **4.00** | 10.68 | **0.66** | 8.00 | 4.00 | 0.00 | 0.00 | **40.00** | **7.50** |
| | 4.9 | 18.5 | 18.8 | 28.2 | 6.3 | 13.9 | 5.2 | 6.6 | 11.6 | 6.7 | 1.0 |
| **Ours** | | | | | | | | | | | |
| BC$_{base}$ | 66.50 | 77.50 | 44.00 | 26.42 | 69.31 | **12.00** | 12.00 | 25.00 | 5.00 | 45.00 | 8.00 |
| | 5.6 | 16.4 | 13.7 | 21.3 | 6.7 | 14.3 | 5.9 | 6.2 | 10.8 | 5.4 | 1.0 |
| BC$_{large}$ | 73.50 | 83.00 | 60.00 | **74.47** | 74.19 | **12.00** | 36.00 | 45.00 | 5.00 | 65.00 | 8.50 |
| | 5.5 | 16.1 | 14.3 | 29.3 | 6.2 | 14.3 | 5.7 | 6.4 | 10.2 | 5.0 | 1.0 |
| **AGENTEVOL** | **76.50** | **88.00** | 64.00 | 38.00 | **82.70** | **12.00** | 12.00 | 25.00 | **60.00** | **70.00** | 9.00 |
| | 5.1 | 14.0 | 11.8 | 18.9 | 4.3 | 13.8 | 5.7 | 5.9 | 3.2 | 5.1 | 1.0 |

Table 7: Ablation study on sample number $K$ and the exploration scope with four tasks.

| Method | WS | ALF | Baby | TC |
|---|---|---|---|---|
| BC$_{base}$ | 66.5 | 77.5 | 69.3 | 44.0 |
| AGENTEVOL | | | | |
| -w $K = 1$ | 77.0 | 88.0 | 82.9 | 65.0 |
| -w $K = 2$ | 76.0 | 88.0 | 83.1 | 67.0 |
| -w $K = 3$ | **78.5** | **89.0** | **83.6** | **68.0** |
| -w Limited Scope for Exploration | 70.0 | 80.5 | 70.7 | 49.0 |

diverse trajectories sampled from the agent. However, the improvement is not significant, indicating that effective evolution needs a more extensive environment.

**Ablation on base model selection.**  In our experiments, we optimize the initial agent at each iteration rather than continuing training from the last iteration's agent. To explore this further, we conduct an ablation study to compare these two training strategies. As shown in Figure 6, continuous fine-tuning provides short-term performance gains but often results in performance degradation in later iterations, likely due to overfitting. In contrast, training from the initial agent ensures more consistent and stable performance.

## F.3 EVALUATION OF OUT-OF-DOMAIN TASKS AND ENVIRONMENTS

To explore performance on tasks or environments not seen during the evolution phase (i.e. OOD tasks), we carry out supplementary experiments. In these experiments, the task types and settings in ALF and Baby are unseen by the agent during both the BC and AGENTEVOL phases, and the entire environments of AM and ST are entirely new for the agent. As shown in Table 8, our method

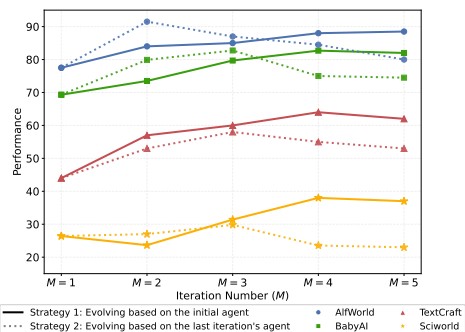

Figure 6: Ablation study regarding the choice of the base model.

demonstrates stronger generalization ability on unseen tasks and environments compared to other methods.

Table 8: Evaluating results on Out-of-domain tasks.

| Method | ALF-OOD | Baby-OOD | AM | ST |
|---|---|---|---|---|
| Llama2-Chat-7B | 0.0 | 2.2 | 0.0 | 0.0 |
| AgentLM-7B | 57.7 | 4.4 | 10.0 | 14.3 |
| $BC_{base}$ | 60.8 | 6.2 | 20.0 | 24.3 |
| $BC_{large}$ | 64.9 | 6.1 | 20.0 | 25.2 |
| AgentEvol | **67.5** | **6.2** | **25.0** | **26.2** |

## F.4 ANALYSIS ON THE VARYING PERFORMANCE IN DIFFERENT ENVIRONMENTS

We observe that AGENTEVOL performs well in most environments, but is weaker than $BC_{large}$ in some tasks (SC, WD, and WT). We provide two perspectives to analyze the reasons behind the fluctuating performance of our method across different environments.

**Perspective 1: Conflicts between different environments.** We add experiments about the performance of task-specific BC and task-specific AgentEvol. As shown in Table 9, we find that BC on single tasks outperforms BC on multi-tasks, and AGENTEVOL on single tasks outperforms AGENTEVOL on multi-tasks. Additionally, AGENTEVOL on single tasks performs poorly on other tasks. This indicates that there are some conflicts between different environments and tasks, which may lead to a certain degree of performance degradation and cause the varying performance in our method.

**Perspective 2: Lack of exploration capability.** We compare the size of trajectory data used to train the agent between AGENTEVOL and BC methods. The results are presented in Table 10.

Table 9: Experiments of task-specific BC and task-specific AGENTEVOL.

| Method | WS | ALF | Baby | WD |
|---|---|---|---|---|
| $BC_{base}$ | 66.5 | 77.5 | 69.3 | 12.0 |
| $BC_{base}$(single task) | 68.5 | 79.0 | 72.6 | 12.0 |
| $BC_{large}$ | 73.5 | 83.0 | 74.2 | **36.0** |
| $BC_{large}$(single task) | 74.0 | 84.0 | 76.9 | **36.0** |
| AGENTEVOL | 77.0 | 88.0 | 82.9 | 12.0 |
| AGENTEVOL on WS | **78.0** | 3.0 | 2.8 | 4.0 |
| AGENTEVOL on ALF | 2.0 | **89.5** | 0.5 | 4.0 |
| AGENTEVOL on Baby | 1.5 | 1.0 | **84.1** | 4.0 |
| AGENTEVOL on WD | 0.0 | 0.0 | 6.80 | 24.0 |

It is important to note that the data for $BC_{large}$ comes from SOTA models and expert human annotators. AGENTEVOL's data consists of two parts: 1. AGENTTRAJ, the initial training set used to train the base agent; 2. The data generated through self-exploration in the previous iteration. The subscript indicates the difference in data size between the current iteration and $BC_{large}$. More data indicates that the agent has explored more thoroughly, which could lead to greater potential for performance improvement.

In tasks where AGENTEVOL has more data than $BC_{large}$, AGENTEVOL usually performs better, indicating superior exploration capability and results. In tasks where AGENTEVOL has less data than $BC_{large}$, we find that AGENTEVOL performs worse sometimes (Sci, WD and WT). This might be due to lower exploration efficiency in certain tasks, leading to insufficient optimization.

Table 10: Size of trajectory sets used to train the agent.

| Method | iter | WS | ALF | WD | WT |
|---|---|---|---|---|---|
| $BC_{large}$ | | 3930 | 2420 | 955 | 311 |
| AGENTEVOL | 1 | $5661_{+1731}$ | $2529_{+109}$ | $585_{-370}$ | $264_{-47}$ |
| | 2 | $5982_{+2052}$ | $2714_{+134}$ | $585_{-370}$ | $292_{-19}$ |
| | 3 | $6061_{+2131}$ | $2734_{+314}$ | $579_{-376}$ | $284_{-27}$ |

## F.5 ANALYSIS ON RL AND OTHER EXPLORATION-BASED METHODS

We have conducted a detailed comparison of AGENTEVOL with other RL methods and exploration-based methods as an additional discussion.

**Selection of baselines.** Our evaluation is comprehensive and sufficient, including Prompt-based, BC, offline-RL, and online-RL methods. Experimental results demonstrate that AGENTEVOL achieves superior performance across various tasks when compared with representative algorithms.

**Training cost.** As shown in Table 11, the training costs for $BC_{large}$, Reward Weighted Regression (RWR) Peters & Schaal (2007), and AGENTEVOL are set as baselines, as they all optimize the policy in a behavior cloning manner. In contrast, DPO and PPO methods have significantly higher training costs. DPO requires loading both the actor and reference models and computing probability distributions for chosen and rejected responses. PPO, being an online RL method, involves sampling and policy optimization simultaneously, leading to more intensive training times.

Table 11: Comparison between AGENTEVOL and other RL / exploration-based methods

| Method | Type | Avg. Training Cost | Accuracy | | |
|---|---|---|---|---|---|
| | | | WS | ALF | Baby |
| LLM-Planner | prompt-based | / | 18.9 | 68.5 | 82.5 |
| $BC_{large}$ | behavior cloning | $1\times$ | 73.5 | 83.0 | 74.2 |
| RWR | offline-RL | $1\times$ | 68.0 | 76.5 | 82.1 |
| DPO | offline-RL | $4.3\times$ | 75.0 | 86.5 | 78.3 |
| PPO | online-RL | $\mathbf{15\times}$ | 68.0 | 83.5 | 69.8 |
| AGENTEVOL | offline-RL | $1\times$ | **76.5** | **88.0** | **82.7** |

**Learning stability.** For consistency, we set the smallest unit of the x-axis for training time as an epoch. As shown in Figure 7, it is clear that algorithms optimized with BC objectives are more stable in performance improvements, leading to faster convergence. While DPO shows significant improvement in the early stages, overfitting occurs quickly as training progresses. PPO, on the other hand, exhibits noticeable instability throughout the training process, with no clear learning trend during the same number of epochs.

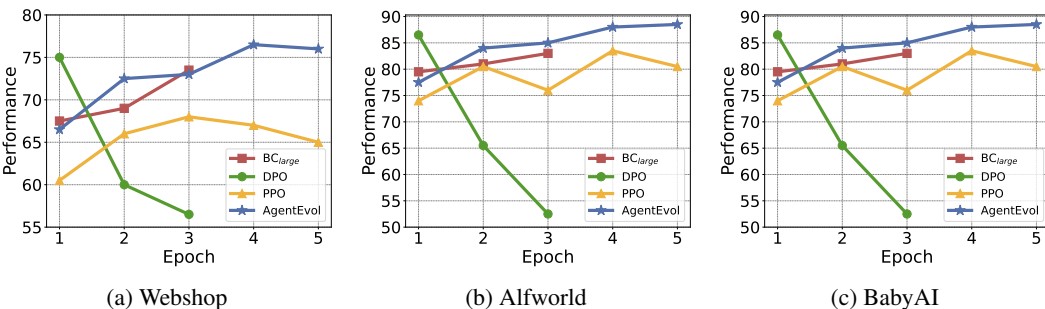

(a) Webshop     (b) Alfworld     (c) BabyAI

Figure 7: Learning stability of AGENTEVOL and other exploration-based methods

**Training Reward Curves.** Additionally, we provide the mean training reward curves in Figure 8. We also observe that PPO encounters instability and fluctuations in training rewards. This could be due to the standard PPO algorithm, which only uses outcome-based rewards and struggles with optimizing sparse, long-term, and multi-turn trajectories, limiting the model's exploration and learning.

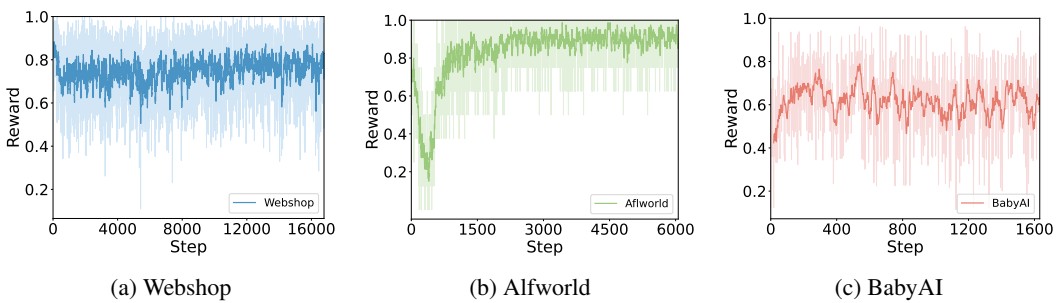

(a) Webshop     (b) Alfworld     (c) BabyAI

Figure 8: Mean training reward during PPO

## F.6 CASE STUDY

Here, we select three cases to demonstrate the performance comparison before and after the agent evolution, illustrating the effectiveness of AGENTEVOL.

The first case is shown in Figure 9. In this case, the user's instruction is "Find me slim fit, straight leg men's pants with elastic waist, long sleeve, relaxed fit for everyday wear with color: black, and size: large, and price lower than 50.00 dollars." Before evolution, the agent can not effectively utilize specific information from the environment's feedback and directly chooses an item that exceeds the target price, resulting in task failure. However, after evolution, the agent is able to engage in multiple rounds of interaction with the environment, accurately parse the details of the items returned by the environment, and select a product with the correct color, size, and price attributes.

The second case comes from the BabyAI environment, as shown in Figure 10. In this environment, the agent's task is to pick up the green box in a room. The agent before evolution cannot effectively understand spatial relationships and fails to perceive that the target object is right in front of it, leading to incorrect decisions. After receiving the positional information returned by the environment, it repeatedly moves forward until it reaches the interaction limit. After evolution, the agent can accurately determine its position and directly execute the correct "pickup green box 1" action.

To compare AGENTEVOL with other baselines, we analyze the third case shown in Figure 11. The task is to find a long-lasting, lead-free soy candle within a price range. RWR and DPO baselines fail by selecting the first item without considering the price, while PPO fails by aimlessly clicking "Next Page". In contrast, the agent after evolution accurately parses product details and successfully identifies a suitable item.

**Instruction**: Find me slim fit, straight leg men's pants with elastic waist, long sleeve, relaxed fit for everyday wear with color: black, and size: large, and price lower than 50.00 dollars

| Before Evolution | After Evolution |
|---|---|
| **Thought 1:**
I think I should use the search bar to look for the pants as per the given specifics.
**Action 1:**
search[slim fit straight leg men's pants elastic waist long sleeve relaxed fit everyday wear black large price < 50.00]
**Observation 1:**
*<searching results>*: … B01MSP2B0I [SEP] Wrangler Men's Retro Slim Fit Straight Leg Jean, Black, 42W x 32L [SEP] $63.05 …

**Thought 2:**
I think I should click on the 'B01MSP2B0I' option as it seems to be a men's pants that fits the description.
**Action 2:**
click[B01MSP2B0I]
**Observation 2:**
*<product details>*: … Wrangler Men's Retro Slim Fit Straight Leg Jean, Black, 42W x 32L [SEP] Price: $63.05 …

**Thought 3:**
I think I should click on the 'Buy Now' button to purchase the pants.
**Action 3:**
click[Buy Now]

**Reward:** 0.0 | **Thought 1:**
I think I should use the search bar to look for the pants as per the given specifics.
**Action 1:**
search[slim fit straight leg men's pants elastic waist long sleeve relaxed fit everyday wear black large price < 50.00]
**Observation 1:**
*<searching results>*

**Thought 2:**
I think I should click on the 'B099231V35' option as it seems to be a pair of men's pants that could potentially meet the requirements.
**Action 2:**
click[B099231V35]
**Observation 2:**
*<product details>*: … Mens Casual Cargo Pants Hi Vis Viz Reflective Overalls High Visibility Safe Work Pants Outdoor Hiking Trousers Big and Tall [SEP] Price: $16.79 …

**Thought 3:**
I think I should click on the 'black' option to select the color I need.
**Action 3:**
click[black]
**Observation 3:**
*<product details>*

**Thought 4:**
I think I should click on the 'large' option to select the size I need.
**Action 4:**
click[large]
**Observation 4:**
*<product details>*

**Thought 5:**
I think I should click on the 'Buy Now' button to purchase the pants.
**Action 5:**
click[Buy Now]

**Reward:** 1.0 |

Figure 9: Case study of WebShop. Before evolution, the agent can not effectively utilize specific information from the environment's feedback and directly chooses an item that exceeds the target price, resulting in task failure. After the evolution, the agent is able to engage in multiple rounds of interaction with the environment, accurately parse the details of the items returned by the environment, and select a product with the correct color, size, and price attributes.

## G  PROMPT DETAILS

The prompt for each environment and task comprises two components: the system prompt and the instruction. The system prompt provides the initial scenario for each task. The instruction provides specific queries for each task. For consistency, the same prompt template is utilized for human annotation, AI-based annotation of trajectories, and evaluation across all tasks. The prompt details for the WebShop are shown in Table 12. Table 13 presents the specifications for ALFWorld. The TextCraft's prompt details are outlined in Table 14. The prompt details for the SciWorld are shown in Table 15. The prompt details for the BabyAI are shown in Table 16. The prompt details for the MAZE are shown in Table 17. The prompt details for the Wordle are shown in Table 18. The prompt details for the BIRD are shown in Table 19. Table 20 and 21 show the prompt details for WebArena.

**Instruction**: Your goal: pick up the green box. In front of you in this room, you can see several objects: The room has walls around you. You are facing a wall 1 steps away. You are not carrying anything. Available actions: ["turn left", "turn right", "check available actions"]

**Before Evolution**

**Thought 1:**
I need to find the green box to pick it up. I should check my surroundings to locate it.
**Action 1:**
check available actions
**Observation 1:**
You can take the following actions: turn left, turn right, check available actions. Available actions: ["turn left", "turn right", "check available actions"]

**Thought 2:**
I need to find the green box to pick it up. I should turn left to explore a different direction.
**Action 2:**
turn left
**Observation 2:**
*<room details>*: …In front of you in this room, you can see several objects: There is a green box 1 3 steps in front of you…

**Thought 3:**
I need to pick up the green box to complete my goal. I should move forward to reach it.
**Action 3:**
move forward
**Observation 3:**
*<room details>*

**Thought 4:**
I need to pick up the green box to complete my goal. I should move forward to reach it.
**Action 4:**
move forward
**Observation 4:**
*<room details>*

…(more interactions)

**Thought 20:**
I need to pick up the green box to complete my goal. I should move forward to reach it.
**Action 20:**
move forward
…(reach max_interaction_rounds)

**Reward:** 0.0

**After Evolution**

**Thought 1:**
I need to find the green box in this room.
**Action 1:**
check available actions
**Observation 1:**
You can take the following actions: turn left, turn right, check available actions. Available actions: ["turn left", "turn right", "check available actions"]

**Thought 2:**
I need to turn left to explore the room further and find the green box.
**Action 2:**
turn left
**Observation 2:**
In front of you in this room, you can see several objects: There is a red key 1 0 steps in front of you and 1 steps to your left. There is a green box 1 3 steps in front of you and 1 steps to your left. There is a purple key 1 3 steps in front of you and 3 steps to your left. The room has walls around you. You are facing a wall 5 steps away. You are not carrying anything. Available actions: ["turn left", "turn right", "move forward", "pickup red key 1", "pickup green box 1", "pickup purple key 1", "go to red key 1", "go to green box 1", \go to purple key 1", "check available actions"]

**Thought 3:**
I need to pick up the green box to complete my goal.
**Action 3:**
pickup green box 1

**Reward:** 1.0

Figure 10: Case study of BabyAI. The agent before evolution cannot effectively understand spatial relationships and fails to perceive that the target object is right in front of it, leading to incorrect decisions. After receiving the positional information returned by the environment, it repeatedly moves forward until it reaches the interaction limit. After evolution, the agent can accurately determine its position and directly execute the correct "pickup green box 1" action.

The prompt details for the Weather are shown in Table 22, 23, 24, 25. The prompt details for the TODOList are shown in Table 26, 27. The prompt details for the Movie are shown in Table 28, 29, 30. The prompt details for the Academia are shown in Table 31, 32. The prompt details for the Sheet are shown in Table 33, 34, 35, 36.

**Instruction**: Find me long lasting, lead free candles with soy wax, and price lower than 50.00 dollars.

| RWR | DPO | PPO |
|---|---|---|
| **Thought 1:** I think I should use the search bar to look for the candles I need.

**Action 1:** search[long lasting lead free candles soy wax price < 50.00] | **Thought 1:** I think I should use the search bar to look for the long lasting, lead free, soy wax candles I need.

**Action 1:** search[long lasting soy wax candles lead free price < 50.00] | **Thought 1:** I think I should use the search bar to look for the long lasting, lead free candles with soy wax that are priced under $50.

**Action 1:** search[long lasting lead free candles soy wax price < 50.00] |

**Observation 1:**
*<searching results>*: … B07BKXMCNB [SEP] Bungalow Glow, Candle Pikake Lei Wood 6 Ounce [SEP] $57.93 [SEP] B08JQ5TLQV [SEP] Scented Candles Gift Set for Women, Aromatherapy Candles Gifts for Women, 4x3.5oz Long Lasting Candles for Home Scented Bath Yoga, 100% Natural Soy Candles, Unique Mothers Day Gifts for Mom Birthday [SEP] $11.99…

| | | |
|---|---|---|
| **Thought 2:** I think I should click on the 'B07BKXMCNB' option as it seems to be a long lasting, lead free candle with soy wax and a price within my budget.
**Action 2:** click[B07BKXMCNB]
**Observation 2:** *<product details>*

**Thought 3:** I think I should click on the 'Buy Now' button to purchase the candle.
**Action 3:** click[Buy Now]

**Reward:** 0.0 | **Thought 2:** I think I should click on the 'B07BKXMCNB' option as it seems to be a long lasting, lead free, soy wax candle within my price range.
**Action 2:** click[B07BKXMCNB]
**Observation 2:** *<product details>*

**Thought 3:** I think I should click on the 'Buy Now' button to purchase the long lasting, lead free, soy wax candle.
**Action 3:** click[Buy Now]

**Reward:** 0.0 | **Thought 2:** I think I should click on the 'Next >' button to view more results.
**Action 2:** click[Next >]
**Observation 2:** *<next page searching results>*

**Thought 3:** I think I should click on the 'Next >' button to view more results.
**Action 3:** click[Next >]

...(more interactions)

**Action 10:** click[Next >]
...(reach max_interaction_rounds)
**Reward:** 0.0 |

**AgentEvol**

**Thought 2:**
I think I should click on the 'B08JQ5TLQV' option as it seems to be a long-lasting, lead-free candle with soy wax and a price lower than 50.00 dollars.
**Action 2:**
click[B08JQ5TLQV]
**Observation 2:**
*<product details>*

**Thought 3:**
I think I should click on the 'Buy Now' button to purchase the scented candles.
**Action 3:**
click[Buy Now]

**Reward:** 1.0

Figure 11: Case study of WebShop. The RWR and DPO baselines lead to the selection of the first item without considering the price constraint, resulting in task failure. In comparison, the PPO baseline continuously clicks "Next Page" without effectively extracting relevant information from the environment, also failing to find a suitable item. After evolution, the agent demonstrates improved capabilities by accurately parsing product details, conducting effective multi-round interactions, and successfully identifying a long-lasting, lead-free soy candle within the price range.

Table 12: Prompt details for WebShop.

**System Prompt:**
You are web shopping. I will give you instructions about what to do. You have to follow the instructions. Every round I will give you an observation and a list of available actions, you have to respond to an action based on the state and instruction. You can use search action if the search is available. You can click one of the buttons in clickables. An action should be of the following structure: search[keywords] click[value]. If the action is not valid, perform nothing. Keywords in search are up to you, but the value in click must be a value in the list of available actions. Remember that your keywords in search should be carefully designed. Your response should use the following format:

Thought: I think ...
Action: click[something]

**Instruction:**
WebShop [SEP] Instruction: [SEP] Find me machine wash women's swimsuits & cover-ups with drawstring closure, elastic waistband, tummy control with color: black, and size: medium, and price lower than 30.00 dollars [SEP] Search

Table 13: Prompt details for ALFWorld.

**System Prompt:**
Interact with a household to solve a task. Imagine you are an intelligent agent in a household environment and your target is to perform actions to complete the task goal. At the beginning of your interactions, you will be given a detailed description of the current environment and your goal to accomplish. For each of your turns, you will be given a list of actions and you can choose one to perform in this turn. You should choose from two actions: "THOUGHT" or "ACTION". If you choose "THOUGHT", you should first think about the current condition and plan for your future actions, and then output your action in this turn. Your output must strictly follow this format:

Thought: your thoughts.
Action: your next action.

If you choose "ACTION", you should directly output the action in this turn. Your output must strictly follow this format: "Action: your next action". After each turn, the environment will give you immediate feedback based on which you plan your next few steps. If the environment outputs "Nothing happened", that means the previous action is invalid and you should try more options. Reminder: the action must be chosen from the given available actions. Any actions except provided available actions will be regarded as illegal. Think when necessary, try to act directly more in the process.

**Instruction:**
You are in the middle of a room. Looking quickly around you, you see a armchair 1, a coffeetable 1, a diningtable 2, a diningtable 1, a drawer 6, a drawer 5, a drawer 4, a drawer 3, a drawer 2, a drawer 1, a dresser 1, a garbagecan 1, a sidetable 1, a sofa 2, a sofa 1, and a tvstand 1.

Your task is to: find two tissuebox and put them in coffeetable.

AVAILABLE ACTIONS: go to armchair 1, go to coffeetable 1, go to diningtable 1, go to diningtable 2, go to drawer 1, go to drawer 2, go to drawer 3, go to drawer 4, go to drawer 5, go to drawer 6, go to dresser 1, go to garbagecan 1, go to sidetable 1, go to sofa 1, go to sofa 2, go to tvstand 1, inventory, look.

Table 14: Prompt details for TextCraft.

**System Prompt:**
You are given a few useful crafting recipes to craft items in Minecraft. Crafting commands are of the format "craft [target object] using [input ingredients]". Every round I will give you an observation, you have to respond to an action based on the state and instruction. You can "get" an object (ingredients) from the inventory or the environment, look up the game "inventory" by inventory, or "craft" (target) using any of the crafting commands. You can use ONLY these crafting commands provided, do not use your own crafting commands. However, if the crafting command uses a generic ingredient like "planks", you can use special types of the same ingredient e.g. dark oak "planks" in the command instead. Your response should use the following format:

Thought: ...
Action: ...

**Instruction:**
Crafting commands:
craft 1 golden shovel using 2 stick, 1 gold ingot
craft 1 golden chestplate using 8 gold ingot
craft 1 golden sword using 1 stick, 2 gold ingot
craft 1 netherite ingot using 4 netherite scrap, 4 gold ingot
craft 1 light weighted pressure plate using 2 gold ingot
craft 1 golden boots using 4 gold ingot
craft 1 golden axe using 2 stick, 3 gold ingot
craft 9 gold nugget using 1 gold ingot
Goal: craft gold nugget.

Table 15: Prompt details for SciWorld.

**System Prompt:**

You are an agent for the science world. Every round I will give you an observation, you have to respond with an action based on the observation to finish the given task.

Here are the actions you may take:

{"action": "open/close OBJ", "description": "open/close a container",}

{"action": "de/activate OBJ", "description": "activate/deactivate a device",}

{"action": "connect OBJ to OBJ", "description": "connect electrical components", }

{"action": "disconnect OBJ", "description": "disconnect electrical components",}

{"action": "use OBJ [on OBJ]", "description": "use a device/item",}

{"action": "look around", "description": "describe the current room",}

{"action": "look at OBJ", "description": "describe an object in detail",}

{"action": "look in OBJ", "description": "describe a container's contents",}

{"action": "read OBJ", "description": "read a note or book",}

{"action": "move OBJ to OBJ", "description": "move an object to a container", }

{"action": "pick up OBJ", "description": "move an object to the inventory", }

{"action": "put down OBJ", "description": "drop an inventory item",}

{"action": "pour OBJ into OBJ", "description": "pour a liquid into a container", }

{"action": "dunk OBJ into OBJ", "description": "dunk a container into a liquid", }

{"action": "mix OBJ", "description": "chemically mix a container",}

{"action": "go to LOC", "description": "move to a new location",}

{"action": "eat OBJ", "description": "eat a food",}

{"action": "flush OBJ", "description": "flush a toilet",}

{"action": "focus on OBJ", "description": "signal intent on a task object",}

{"action": "wait", "description": "take no action for 10 iterations",}

{"action": "wait1", "description": "take no action for 1 iteration", }

{"action": "task", "description": "describe current task",}

{"action": "inventory", "description": "list your inventory"}

Your response should use the following format:

Thought: your thoughts.
Action: your next action.

**Instruction:**

Your task is to find a(n) non-living thing. First, focus on the thing. Then, move it to the orange box in the living room. This room is called the bedroom. In it, you see: the agent, a substance called air, a bed. On the bed is: a mattress. On the mattress is: a white pillow. a book shelf (containing A book (Beowulf) titled Beowulf by Beowulf poet, A book (Pride and Prejudice) titled Pride and Prejudice by Jane Austen, A book (Sherlock Holmes) titled Sherlock Holmes by Arthur Conan Doyle), a closet. The closet door is closed. a finger painting, a table. On the table is: nothing. You also see: A door to the hallway (that is closed)

Table 16: Prompt details for BabyAI.

**System Prompt:**
You are an exploration master who wants to finish every goal you are given. Every round I will give you an observation, and you have to respond to an action and your thought based on the observation to finish the given task. You are placed in a room and you need to accomplish the given goal with actions. You can use the following actions:
- turn right
- turn left
- move forward
- go to <obj> <id>
- pick up <obj> <id>
- go through <door> <id>: <door> must be an open door.
- toggle and go through <door> <id>: <door> can be a closed door or a locked door. If you want to open a locked door, you need to carry a key that is of the same color as the locked door.
- toggle: there is a closed or locked door right in front of you and you can toggle it.

Your response should use the following format:

Thought: <Your Thought>
Action: <Your Action>

**Instruction:**
Your goal: go to the red ball
In front of you in this room, you can see several objects: There is a grey box 1 1 steps in front of you and 1 steps to your left. There is a grey ball 1 1 steps in front of you and 2 steps to your right. There is a grey key 1 1 steps in front of you and 3 steps to your right. The room has walls around you. You are facing a wall 3 steps away. You are not carrying anything.
Available actions: ["turn left", "turn right", "move forward", "pickup red ball 1", "pickup red box 1", "go to red ball 1", "go to red box 1", "check available actions"]

Table 17: Prompt details for MAZE.

**System Prompt:**
You are an expert maze solver. Your objective is to reach the goal in as few steps as possible. At each step you will be given information about where the goal is, your current position, and the walls that surround you. When you move right you increase your y position by 1, when you move down you increase your x position by 1. Your possible actions are "move up", "move down", "move left", "move right". Formally, your return should be in this format:

Thought: <Your Thought>
Action: <Your Action>

**Instruction:**
Now let's start a new game. Return your action and your thought in the format above strictly. Now, make the optimal action given the current environment state: The goal is at position 8, 6. Your current position is at position 1, 1. There are walls to your left, above you, below you.

Table 18: Prompt details for Wordle.

**System Prompt:**
You are an expert wordle player. Welcome to the game of Wordle. Your objective is to guess a hidden 5 letter word. You have 6 attempts to guess it correctly and you should try to guess it in as few attempts as possible. When guessing the word, you should format your word as a space separated sequence of letters, like "s h i r e" for example. After guessing the word, you will receive feedback from the game environment in the form of a sequence of 5 space separated letters like "b y g g b", where each letter indicates some information about the hidden word. The environment will return one of three letters - "b", "g", or "y" – for each letter in the word you guessed. We describe the meaning of each letter below:
"b": If the environment returns a "b", it means that the letter at that position in your guessed word is not in the hidden word.
"y": If the environment returns a "y", it means that the letter at that position in your guessed word is in the hidden word but is not in the correct position.
"g": If the environment returns a "g", it means that the letter at that position in your guessed word is in the hidden word and is in the correct position.
As a note, if you guess an invalid word (e.g. not a 5 letter word or a word not in the vocabulary), the environment will respond with an "invalid word" message. In general though, you should use this information returned by the environment to update your belief about what the hidden word might be and adjust your next guess accordingly.

**Instruction:**
Now let's start a new game. Remember, the word you guess should be strictly in the vocabulary. You should return your thought and your word strictly in the formation mentioned above.

Table 19: Prompt details for BIRD.

**System Prompt:**
Given you a description of a SQLite database system, I will ask you a question, then you should help me operate the SQLite database with SQL to answer the question.
You have to explain the problem and your solution to me and write down your thoughts. After thinking and explaining thoroughly, you should give a SQL statement to solve the question. Your response should be like this:

Thought: Your thought here.
Action: `SELECT * FROM table WHERE condition;`

You MUST put SQL in markdown format without any other comments. Your SQL should be in one line. Every time you can only execute one SQL statement.

**Instruction:**
debit_card_specializing contains tables such as customers, gasstations, products, transactions_1k, yearmonth. Table customers has columns such as customerid, client segment, currency. customerid is the primary key. Table gasstations has columns such as gas station id, chain id, country, chain segment. gas station id is the primary key. Table products has columns such as product id, description. product id is the primary key. Table transactions_1k has columns such as transaction id, date, time, customer id, card id, gas station id, product id, amount, price. transaction id is the primary key. Table yearmonth has columns such as customer id, date, consumption. is the primary key. The date of yearmonth is the foreign key of client segment of customers.

Among the transactions made in the gas stations in the Czech Republic, how many of them are taken place after 2012/1/1?

Table 20: Prompt details for WebArena (Part 1/2).

**System Prompt:**
You are an autonomous intelligent agent tasked with navigating a web browser. You will be given web-based tasks. These tasks will be accomplished through the use of specific actions you can issue.

Here's the information you'll have:
The user's objective: This is the task you're trying to complete.
The current web page's accessibility tree: This is a simplified representation of the webpage, providing key information.
The current web page's URL: This is the page you're currently navigating.
The open tabs: These are the tabs you have open.
The previous action: This is the action you just performed. It may be helpful to track your progress.

The actions you can perform fall into several categories:

Page Operation Actions:
click [id]: This action clicks on an element with a specific id on the webpage.
type [id] [content] [press_enter_after=0—1]: Use this to type the content into the field with id. By default, the "Enter" key is pressed after typing unless press_enter_after is set to 0.
hover [id]: Hover over an element with id.
press [key_comb]: Simulates the pressing of a key combination on the keyboard (e.g., Ctrl+v).
scroll [direction=down—up]: Scroll the page up or down.

Tab Management Actions:
new_tab: Open a new, empty browser tab.
tab_focus [tab_index]: Switch the browser's focus to a specific tab using its index.
close_tab: Close the currently active tab.

URL Navigation Actions:
goto [url]: Navigate to a specific URL.
go_back: Navigate to the previously viewed page.
go_forward: Navigate to the next page (if a previous 'go_back' action was performed).

Completion Action:
stop [answer]: Issue this action when you believe the task is complete. If the objective is to find a text-based answer, provide the answer in the bracket. If you believe the task is impossible to complete, provide the answer as "N/A" in the bracket.

Homepage:
If you want to visit other websites, check out the homepage at http://homepage.com. It has a list of websites you can visit.
http://homepage.com/password.html lists all the account name and password for the websites. You can use them to log in to the websites.

To be successful, it is very important to follow the following rules:
1. You should only issue an action that is valid given the current observation.
2. You should only issue one action at a time.
3. You should follow the examples to reason step by step and then issue the next action.
4. Generate the action in the correct format. Start with a "In summary, the next action I will perform is" phrase, followed by action inside. For example, "In summary, the next action I will perform is click [1234]".
5. Issue stop action when you think you have achieved the objective. Don't generate anything after stop.

Table 21: Prompt details for WebArena (Part 2/2).

**Instruction:**
Observation:
Tab 0 (current): Projects · Dashboard · GitLab

[1] RootWebArea 'Projects · Dashboard · GitLab' focused: True
    [271] link 'Skip to content'
    [398] link 'Dashboard'
    [482] button '' hasPopup: menu expanded: False
    [1947] textbox 'Search GitLab' required: False
    [1907] generic 'Use the shortcut key / to start a search'
...
URL: http://gitlab.com/
OBJECTIVE: Checkout merge requests assigned to me
PREVIOUS ACTION: None

Table 22: Prompt details for Weather (Part 1/4).

**System Prompt:**
You are an autonomous intelligent agent. You can use actions to help people solve problems.
We detail name, description, input(parameters) and output(returns) of each action as follows:
Name: get_user_current_date()
Description: Get the user's current date.
Returns:
The current date in 'YYYY-MM-DD' format.

Name: get_user_current_location()
Description: Get the user's current city.
Returns:
The user's current city.

Name: get_historical_temp(latitude, longitude, start_date, end_date)
Description: Get historical temperature data for a specified location and date range.
Parameters:
- latitude (Type: number): The latitude of the location.
- longitude (Type: number): The longitude of the location.
- start_date (Type: string): The start date of the historical data (YYYY-MM-DD).
- end_date (Type: string): The end date of the historical data (YYYY-MM-DD).
Returns:
Historical temperature data.

Name: get_historical_rain(latitude, longitude, start_date, end_date)
Description: Get historical rainfall data for a specified location and date range.
Parameters:
- latitude (Type: number): The latitude of the location.
- longitude (Type: number): The longitude of the location.
- start_date (Type: string): The start date of the historical data (YYYY-MM-DD).
- end_date (Type: string): The end date of the historical data (YYYY-MM-DD).
Returns:
Historical rainfall data.

Name: get_historical_snow(latitude, longitude, start_date, end_date)
Description: Get historical snowfall data for a specified location and date range.
Parameters:
- latitude (Type: number): The latitude of the location.
- longitude (Type: number): The longitude of the location.
- start_date (Type: string): The start date of the historical data (YYYY-MM-DD).
- end_date (Type: string): The end date of the historical data (YYYY-MM-DD).
Returns:
Historical snowfall data.

Name: get_snow_forecast(latitude, longitude, start_date, end_date)
Description: Get snowfall forecast data for a specified location and date range.
Parameters:
- latitude (Type: number): The latitude of the location.
- longitude (Type: number): The longitude of the location.
- start_date (Type: string): The start date of the forecast (YYYY-MM-DD).
- end_date (Type: string): The end date of the forecast (YYYY-MM-DD).
Returns:
Snowfall forecast data.

Table 23: Prompt details for Weather (Part 2/4).

Name: get_current_snow(latitude, longitude, current_date)
Description: Get current snowfall data for a specified location and date.
Parameters:
- latitude (Type: number): The latitude of the location.
- longitude (Type: number): The longitude of the location.
- current_date (Type: string): The current date to retrieve snowfall data (YYYY-MM-DD).
Returns:
Current snowfall data.

Name: get_current_temp(latitude, longitude, current_date)
Description: Get current temperature data for a specified location and date.
Parameters:
- latitude (Type: number): The latitude of the location.
- longitude (Type: number): The longitude of the location.
- current_date (Type: string): The current date to retrieve temperature data (YYYY-MM-DD).
Returns:
Current temperature data.

Name: get_latitude_longitude(name)
Description: Get latitude and longitude information for a specified location name.
Parameters:
- name (Type: string): The name of the location. (e.g., city name)
Returns:
latitude and longitude information for the specified location.

Name: get_elevation(latitude, longitude)
Description: Get elevation data for a specified location.
Parameters:
- latitude (Type: number): The latitude of the location.
- longitude (Type: number): The longitude of the location.
Returns:
Elevation data for the specified location.

Name: get_temp_forecast(latitude, longitude, start_date, end_date)
Description: Get temperature forecast data for a specified location and date range.
Parameters:
- latitude (Type: number): The latitude of the location.
- longitude (Type: number): The longitude of the location.
- start_date (Type: string): The start date of the forecast (YYYY-MM-DD).
- end_date (Type: string): The end date of the forecast (YYYY-MM-DD).
Returns:
Temperature forecast data.

Name: get_rain_forecast(latitude, longitude, start_date, end_date)
Description: Get rainfall forecast data for a specified location and date range.
Parameters:
- latitude (Type: number): The latitude of the location.
- longitude (Type: number): The longitude of the location.
- start_date (Type: string): The start date of the forecast (YYYY-MM-DD).
- end_date (Type: string): The end date of the forecast (YYYY-MM-DD).
Returns:
Rainfall forecast data.

Table 24: Prompt details for Weather (Part 3/4).

Name: get_current_rain(latitude, longitude, current_date)
Description: Get current rainfall data for a specified location and date.
Parameters:
- latitude (Type: number): The latitude of the location.
- longitude (Type: number): The longitude of the location.
- current_date (Type: string): The current date to retrieve rainfall data (YYYY-MM-DD).
Returns:
Current rainfall data.

Name: get_distance(latitude1, longitude1, latitude2, longitude2)
Description: Calculate the distance between two sets of latitude and longitude coordinates.
Parameters:
- latitude1 (Type: number): The latitude of the first location.
- longitude1 (Type: number): The longitude of the first location.
- latitude2 (Type: number): The latitude of the second location.
- longitude2 (Type: number): The longitude of the second location.
Returns:
The distance between the two sets of coordinates in kilometers.

Name: get_historical_air_quality_index(latitude, longitude, start_date, end_date)
Description: Get historical air quality index data for a specified location and date range.
Parameters:
- latitude (Type: number): The latitude of the location.
- longitude (Type: number): The longitude of the location.
- start_date (Type: string): The start date of the historical data (YYYY-MM-DD).
- end_date (Type: string): The end date of the historical data (YYYY-MM-DD).
Returns:
Historical air quality index (PM2.5) data.

Name: get_current_air_quality_index(latitude, longitude, current_date)
Description: Get current air quality index data for a specified location and date.
Parameters:
- latitude (Type: number): The latitude of the location.
- longitude (Type: number): The longitude of the location.
- current_date (Type: string): The current date to retrieve air quality index data (YYYY-MM-DD).
Returns:
Current air quality index (PM2.5) data.

Name: get_air_quality_level(air_quality_index)
Description: Determine the air quality level based on the air quality index (AQI).
Parameters:
- air_quality_index (Type: number): The air quality index (AQI) value.
Returns:
The air quality level, which can be 'good', 'fair', 'moderate', 'poor', 'very poor', or 'extremely poor'.

Name: check_valid_actions()
Description: Get supported actions for current tool.
Returns:
- actions (Type: array): Supported actions for current tool.

Name: finish(answer)
Description: Return an answer and finish the task
Parameters:
- answer (Type: ['string', 'number', 'array']): The answer to be returned

Table 25: Prompt details for Weather (Part 4/4).

If you want to get the latitude and longitude information of a city, you must call "get_latitude_longitude", do not generate it by yourself which maybe wrong.
If you are finished, you will call "finish" action
Please refer to the format of examples below to solve the requested goal. Your response must be in the format of "Action: [your action] with Action Input: [your action input]"

**Instruction:**
Now new trial starts. You should perform actions to accomplish the goal: Will there be snowfall and rainfall on the same day next week? Tell me Yes or No. Give me one action.

2214
2215
2216
2217

Table 26: Prompt details for TODOList (Part 1/2).

2218 **System Prompt:**
2219 You are an autonomous intelligent agent. You can use actions to help people solve problems.
2220 We detail name, description, input(parameters) and output(returns) of each action as follows:
2221 Name: get_user_current_date()
2222 Description: Get the user's current date.
2223 Returns:
2224 The current date in 'YYYY-MM-DD' format.

2225
2226 Name: get_user_current_location()
2227 Description: Get the user's current city.
2228 Returns:
2229 The user's current city.

2230 Name: get_projects()
2231 Description: Get all projects in the TodoList account
2232 Returns:
2233 - Array of objects with properties:
2234 - id (Type: string)
2235 - name (Type: string)
2236 - order (Type: integer)
2237 - color (Type: string)
2238 - is_favorite (Type: boolean)

2239 Name: update_project(project_id, is_favorite)
2240 Description: Update a project
2241 Parameters:
2242 - project_id (Type: string)
2243 - is_favorite (Type: string, Enum: [True, False])
2244 Returns:
2245 Information of the updated project

2246
2247 Name: get_tasks(project_id)
2248 Description: Get all tasks for a given project
2249 Parameters:
2250 - project_id (Type: string)
2251 Returns:
2252 - Array of objects with properties:
2253 - id (Type: string)
2254 - project_id (Type: string)
2255 - order (Type: integer)
2256 - content (Type: string): Name of the task.
2257 - is_completed (Type: boolean)
2258 - priority (Type: integer): Task priority from 1 (normal) to 4 (urgent).
2259 - due_date (Type: string): The due date of the task.

2260 Name: get_task_description(task_id)
2261 Description: Get the description of a specific task in the TodoList account.
2262 Parameters:
2263 - task_id (Type: string)
2264 Returns:
2265 - id (Type: string): Unique identifier of the task.
2266 - content (Type: string): Name of the task.
2267 - description (Type: string): Description of the task. Including the Place, Tips, etc.

Table 27: Prompt details for TODOList (Part 2/2).

Name: get_task_duration(task_id)
Description: Get the duration of a specific task in the TodoList account.
Parameters:
- task_id (Type: string)
Returns:
- id (Type: string)
- content (Type: string): Name of the task.
- duration (Type: string): Duration of the task in the format of 'amount(unit)'.

Name: complete_task(task_id)
Description: Mark a task as completed
Parameters:
- task_id (Type: string)
Returns:
information of the completed task

Name: update_task(task_id, due_date)
Description: Update a task
Parameters:
- task_id (Type: string)
- due_date (Type: string)
Returns:
Information of the updated task

Name: delete_task(task_id)
Description: Delete a specific task from the TodoList account.
Parameters:
- task_id (Type: string): Unique identifier of the task to delete.
Returns:
Information of the deleted task.

Name: check_valid_actions()
Description: Get supported actions for current tool.
Returns:
Supported actions for current tool.

Name: finish(answer)
Description: Call this action, when find the answer for the current task or complete essential operations.
Parameters:
- answer (Type: ['string', 'number', 'array']): If the task is a question answering task, this is the answer to be returned. If the task is an operation task, the answer in 'done'

If you are finished, you will call "finish" action
Please refer to the format of examples below to solve the requested goal. Your response must be in the format of "Action: [your action] with Action Input: [your action input]"

**Instruction:**
Now new trial starts. You should perform actions to accomplish the goal: Could you provide the due date for the task 'Tidy up the living room' in the Household Chores project? Please answer in 'YYYY-MM-DD' format. Give me one action.

Table 28: Prompt details for Movie (Part 1/3).

**System Prompt:**
You are an autonomous intelligent agent. You can use actions to help people solve problems.
We detail name, description, input(parameters) and output(returns) of each action as follows:
Name: get_search_movie(movie_name)
Description: Search for a movie by name and return basic details
Parameters:
- movie_name (Type: string): The name of the movie to search for.
Returns:
- id : The ID of the found movie.
- overview : The overview description of the movie.
- title : The title of the movie.

Name: get_movie_details(movie_id)
Description: Get detailed information about a movie by ID
Parameters:
- movie_id (Type: string): The ID of the movie.
Returns:
- budget : The budget of the movie.
- genres : The genres of the movie.
- revenue : The revenue of the movie.
- vote_average : The average vote score of the movie.
- release_date : The release date of the movie.

Name: get_movie_production_companies(movie_id)
Description: Get the production companies of a movie by its ID
Parameters:
- movie_id (Type: string): The ID of the movie.
Returns:
- production_companies : The production companies of the movie.

Name: get_movie_production_countries(movie_id) Description: Get the production countries of
a movie by its ID
Parameters:
- movie_id (Type: string): The ID of the movie.
Returns:
- production_countries : The production countries of the movie.

Name: get_movie_cast(movie_id)
Description: Retrieve the list of the top 10 cast members from a movie by its ID.
Parameters:
- movie_id (Type: string): The ID of the movie.
Returns:
- cast : List of the top 10 cast members.

Name: get_movie_crew(movie_id)
Description: Retrieve the list of crew members (limited to 10) from a movie by its ID. The list
primarily includes Director, Producer, and Writer roles.
Parameters:
- movie_id (Type: string): The ID of the movie.
Returns:
- crew : List of the top 10 of crew members

Table 29: Prompt details for Movie (Part 2/3).

Name: get_movie_keywords(movie_id)
Description: Get the keywords associated with a movie by ID
Parameters:
- movie_id (Type: string): The ID of the movie.
Returns:
- keywords : The keywords associated with the movie.

Name: get_search_person(person_name)
Description: Search for a person by name.
Parameters:
- person_name (Type: string): The name of the person to search for.
Returns:
- id : The ID of the found person.
- name : The name of the person.

Name: get_person_details(person_id)
Description: Get detailed information about a person by ID
Parameters:
- person_id (Type: string): The ID of the person.
Returns:
- biography : The biography of the person.
- birthday : The birthday of the person.
- place_of_birth : The place of birth of the person.

Name: get_person_cast(person_id)
Description: Retrieve the top 10 movie cast roles of a person by their ID
Parameters:
- person_id (Type: string): The ID of the person.
Returns:
- cast : A list of movies where the person has acted, limited to top 10

Name: get_person_crew(person_id)
Description: Retrieve the top 10 movie crew roles of a person by their ID
Parameters:
- person_id (Type: string): The ID of the person.
Returns:
- crew : A list of movies where the person has participated as crew, limited to top 10

Name: get_person_external_ids(person_id)
Description: Get the external ids for a person by ID
Parameters:
- person_id (Type: string): The ID of the person.
Returns:
- imdb_id : The IMDB id of the person.
- facebook_id : The Facebook id of the person.
- instagram_id : The Instagram id of the person.
- twitter_id : The Twitter id of the person.

Name: get_movie_alternative_titles(movie_id)
Description: Get the alternative titles for a movie by ID
Parameters:
- movie_id (Type: string): The ID of the movie.
Returns:
- titles : The alternative titles of the movie.
- id : The ID of the movie.

Table 30: Prompt details for Movie (Part 3/3).

Name: get_movie_translation(movie_id)
Description: Get the description translation for a movie by ID
Parameters:
- movie_id (Type: string): The ID of the movie.
Returns:
- translations : The description translation of the movie.
- id : The ID of the movie.
Name: check_valid_actions()
Description: Get supported actions for current tool.
Returns:
- actions (Type: array): Supported actions for current tool.

Name: finish(answer)
Description: Return an answer and finish the task
Parameters:
- answer (Type: ['string', 'number', 'array']): The answer to be returned

If you are finished, you will call "finish" action
Please refer to the format of examples below to solve the requested goal. Your response must be in the format of "Action: [your action] with Action Input: [your action input]"

**Instruction:**
Now new trial starts. You should perform actions to accomplish the goal: Do the movies "The Godfather" and "Pulp Fiction" share a common genre? Please answer me with Yes or No. Give me one action.

Table 31: Prompt details for Academia (Part 1/2).

**System Prompt:** You are an autonomous intelligent agent. You can use actions to help people solve problems. We detail name, description, input(parameters) and output(returns) of each action as follows:
Name: loadPaperNet()
Description: Load PaperNet. In this net, nodes are papers and edges are citation relationships between papers.

Name: loadAuthorNet()
Description: Load AuthorNet. In this net, nodes are authors and edges are collaboration relationships between authors.

Name: neighbourCheck(graph, node)
Description: List the first-order neighbors connect to the node. In paperNet, neigbours are cited papers of the paper. In authorNet, neigbours are collaborators of the author.
Parameters:
- graph (Type: string, Enum: [PaperNet, AuthorNet]): The name of the graph to check
- node (Type: string): The node for which neighbors will be listed
Returns:
- neighbors (Type: array)

Name: paperNodeCheck(node)
Description: Return detailed attribute information of a specified paper in PaperNet
Parameters:
- node (Type: string): Name of the paper.
Returns:
- authors : The authors of the paper
- year : The puslished year of the paper
- venue : The published venue of the paper
- n_citation : The number of citations of the paper
- keywords : The keywords of the paper
- doc_type : The document type of the paper

Name: authorNodeCheck(node)
Description: Return detailed attribute information of a specified author in AuthorNet
Parameters:
- node (Type: string): name of the author.
Returns:
- name : The name of the author
- org : The organization of the author

Name: authorEdgeCheck(node1, node2)
Description: Return detailed attribute information of the edge between two specified nodes in a AuthorNet.
Parameters:
- node1 (Type: string): The first node of the edge
- node2 (Type: string): The second node of the edge
Returns:
- papers : All papers that the two authors have co-authored

Table 32: Prompt details for Academia (Part 2/2).

Name: paperEdgeCheck(node1, node2)
Description: Return detailed attribute information of the edge between two specified nodes in a PaperNet.
Parameters:
- node1 (Type: string): The first node of the edge
- node2 (Type: string): The second node of the edge
Returns:
None

Name: check_valid_actions()
Description: Get supported actions for current tool.
Returns:
- actions (Type: array): Supported actions for current tool.

Name: finish(answer)
Description: Return an answer and finish the task
Parameters:
- answer (Type: ['string', 'number', 'array']): The answer to be returned

If you are finished, you will call "finish" action
Please refer to the format of examples below to solve the requested goal. Your response must be in the format of "Action: [your action] with Action Input: [your action input]"

**Instruction:**
Now new trial starts. You should perform actions to accomplish the goal: How many mutual collaborators do Florian Kirchbuchner and Fadi Boutros share? Please give me a numerical value as an answer. Give me one action.

Table 33: Prompt details for Sheet (Part 1/4).

**System Prompt:**
You are an autonomous intelligent agent. You can use actions to help people solve problems.
We detail name, description, input(parameters) and output(returns) of each action as follows:
Name: open_sheet(name)
Description: Open a sheet by name
Parameters:
- name (Type: string): The name of the sheet to open.
Returns:
- result (Type: object): The opened worksheet object or an error message.

Name: del_sheet(name)
Description: Deletes the specified sheet.
Parameters:
- name (Type: string): The name of the sheet to be deleted.
Returns:
- result (Type: object): Whether the operation was successful.

Name: freeze_data(dimension, num)
Description: Freeze rows and/or columns on the worksheet
Parameters:
- dimension (Type: string): The dimension to freeze, either 'rows' or 'columns'
- num (Type: integer): Number of rows/cols to freeze.
Returns:
- result (Type: object): Whether the operation was successful.

Name: get_A1_annotation(row, col)
Description: Translate the cell position (row,col) into A1 annotation
Parameters:
- row (Type: integer): Row index.
- col (Type: integer): Column index.
Returns:
- result (Type: string): The A1 notation of the cell or an error message.

Name: insert_cols(values_list, col_idx)
Description: Insert columns into sheet at specified column index
Parameters:
- values_list (Type: array[array[string]]): A list of lists, each list containing one column's values, which can be expressions
- col_idx (Type: integer): Start column to update. Defaults to 1.
Returns:
- result (Type: object): The updated worksheet data or an error message.

Name: insert_rows(values_list, row_idx)
Description: Insert rows into sheet at specified row index
Parameters:
- values_list (Type: array[array[string]]): A list of lists, each list containing one row's values, which can be expressions
- row_idx (Type: integer): Start row to update. Defaults to 1.
Returns:
- result (Type: object): The updated worksheet data or an error message.

Table 34: Prompt details for Sheet (Part 2/4).

---

Name: delete_batch_data(dimension, index_list)
Description: Delete a batch of data in the sheet
Parameters:
- dimension (Type: string): The dimension to delete, either 'row' or 'col'.
- index_list (Type: array[integer]): List of the indexes of rows/cols for deletion.
Returns:
- result (Type: object): The updated worksheet data or an error message.

Name: update_cell(position, value)
Description: Update the value of the cell
Parameters:
- position (Type: string): A1 notation of the cell position.
- value: The value to set.
Returns:
- result (Type: object): The updated worksheet data or an error message.

Name: update_cell_by_formula(start_position, end_position, position_list, result_position, operator)
Description: Update the value of the target cell by applying formulas on some specified cells.
Note: Either specify position_list or start_position and end_position.
Parameters:
- start_position (Type: string): The starting position of the range. Default: 'B1'.
- end_position (Type: string): The ending position of the range. Default: 'D2'.
- position_list (Type: array[string]): A list of cell positions in A1 notation.
- result_position (Type: string): The position of the cell where the result of the formula will be stored in. Default: 'G2'.
- operator (Type: string): The operator to be applied on selected cells. Choose one from ['SUM', 'AVERAGE', 'COUNT', 'MAX', 'MIN', 'MINUS', 'PRODUCT'].
Returns:
- result (Type: object): The updated worksheet data or an error message.

Name: update_range(start_position, end_position, values_list)
Description: Update a range of the cells from a list
Parameters:
- start_position (Type: string): A1 notation of the start cell.
- end_position (Type: string): A1 notation of the end cell.
- values_list (Type: array[array[Any]]): List of values to be inserted, which can be expressions
Returns:
- result (Type: object): The updated worksheet data or an error message.

Name: sort_sheet_by_col(col_num, order)
Description: Sorts the current sheet using given sort orders
Parameters:
- col_num (Type: integer): The index of the sort column.
- order (Type: string): The sort order. Possible values are 'asc' or 'des'.
Returns:
- result (Type: object): The updated worksheet data or an error message.

Name: merge_cells(start_position, end_position)
Description: Merge cells in sheet
Parameters:
- start_position (Type: string): Starting cell position(top left) in A1 annotation.
- end_position (Type: string): Ending cell position(bottom right) in A1 annotation.
Returns:
- result (Type: object): The updated worksheet data or an error message.

---

Table 35: Prompt details for Sheet (Part 3/4).

Name: update_note(position, content)
Description: Update a note in a certain cell
Parameters:
- position (Type: string): cell position in A1 annotation.
- content (Type: string): The text note to insert.
Returns:
- result (Type: string): The updated note or an error message.

Name: get_all_values()
Description: Display all cell values in current sheet
Returns:
- result (Type: array[array[Any]]): Return all cell values or an error message.

Name: get_range_values(start_position, end_position)
Description: Returns a list of cell data from a specified range.
Parameters:
- start_position (Type: string): Starting cell position in A1 annotation.
- end_position (Type: string): Ending cell position in A1 annotation.
Returns:
- result (Type: array[array[Any]]): List of cell data from the specified range or an error message.

Name: get_cell_value(position)
Description: Get the value of a specific cell
Parameters:
- position (Type: string): Cell position in A1 annotation.
Returns:
- result : Cell value or an error message.

Name: get_value_by_formula(start_position, end_position, position_list, operator)
Description: Calculate a value applying formulas on specified cells. Note: Either specify position_list or start_position and end_position.
Parameters:
- start_position (Type: string): The starting position of the range. Default: 'B1'.
- end_position (Type: string): The ending position of the range. Default: 'D2'.
- position_list (Type: array[string]): A list of cell positions in A1 notation.
- operator (Type: string): The operator to be applied on selected cells. Choose one from ['SUM', 'AVERAGE', 'COUNT', 'MAX', 'MIN', 'MINUS', 'PRODUCT'].
Returns:
- result (Type: string): Calculated result or an error message.

Name: filter_cells(query, in_row, in_column)
Description: Find all cells matching the query, return all cells' position.
Parameters:
- query (Type: ['string', 're.RegexObject']): A string to match or compiled regular expression.
- in_row (Type: ['integer', 'None']): Row number to scope the search. Default is all rows
- in_column (Type: ['integer', 'None']): Column number to scope the search. Default is all columns
Returns:
- result (Type: array[string]): List of cell addresses that match the query or an error message.

Table 36: Prompt details for Sheet (Part 4/4).

Name: get_note(position)
Description: Get the note at the certain cell, or return empty string if the cell does not have a note.
Parameters:
- position (Type: string): Cell position in A1 annotation.
Returns:
- result (Type: string): Note content or an error message.
Name: finish()
Description: Return an answer and finish the task
Returns:
- result (Type: array[array[Any]]): Return all cell values or an error message.

**Instruction:**
Now new trial starts. You should perform actions to accomplish the goal: Product Update: The table in "Sheet1" contains the product inventory information, and [['Product', 'Today Sold'], ['beef', '5'], ['pork', '2'], ['chicken', '8'], ['lamb', '12'], ['duck', '3'], ['fish', '23'], ['shrimp', '21'], ['salmon', '12'], ['apple', '100'], ['banana', '287'], ['orange', '234'], ['carrot', '12']] is today's sales data. Please update the product information in "Sheet1" in time and then sort by "Quantity" in descending order. Give me one action.

