# OpenReview forum: "AgentGym: Evaluating and Evolving Large Language Model-based Agents across Diverse Envronments"
_ICLR.cc/2025/Conference — Submitted to ICLR 2025_

### Official Review · Reviewer_jPpZ · 2024-10-24

**Soundness:** 2
**Presentation:** 2
**Contribution:** 3
**Rating:** 5
**Confidence:** 4

**Summary:**

The authors present AgentGym, a global framework for training LLM agents. AgentGym contains a software infrastructure to train an agent onto a set of varied environments, the environments themselves, a benchmark made of a set of instructions (AgentEval), two datasets of expert curated trajectories of different size (AgentTraj and AgentTraj-L), and finally a method to train agents with a behavioral cloning (BC) part and a fine-tuning part based on exploration and learning (AgentEvol). The paper presents the whole framework over the first 3pages, then the rest is dedicated to AgentEvol with a presentation of the method and a experimental study.

I have slightly increased my score as I appreciate the effort of the authors, but I still believe they should make two good papers out of their work.

**Strengths:**

- building the AgentGym framework is an impressive and useful piece of engineering, that needs to be properly advertized
- the AgentEvol method might be a useful piece of research with interesting capabilities to improve the performance of LLM agents, if properly evaluated with a stronger methodology

**Weaknesses:**

- the AgentGym framework part is mostly presented as an engineering effort and is of low interest in the context of a scientific research paper.
- the AgentEvol method is not devoted enough space and is not evaluated thoroughly enough

**Questions:**

This section does not only contain questions, but also remarks and advices for submitting better papers next time. Note also that I knew this work before becoming a reviewer of this paper, so I cannot make as if I was discovering it.
After reading the paper and given my knowledge of the AgentGym repository, I’m convinced that the authors are making a publication strategy mistake : they should split their paper into two contributions:
- one about the AgentGym framework, presenting all the engineering contributions, the datasets, etc. In their ICLR submission, to avoid disclosing their identity, the authors are obliged to state that the framework will be open-sourced after publication, while it is already available and it already has users. This paper will be useful as the framework looks great, but I don’t believe it should be sent to a top scientific machine learning conference, as it is mostly engineering. Doing so would also save a lot of space to present the scientific contribution (AgentEvol) in more details. This paper would contain a part of the introduction, Section 3, and Appendices B, C and D. In this paper, the authors might give more details on the way they collect instructions when they rely on an AI-based, automated method.
- and one about the AgentEvol method. The rest of my review will focus mostly on this latter part.

# Comments about AgentEvol and the methodology
- If I were the authors, I would merge Section 2 into Section 4.
- When looking at the final update equations, it seems that the AgentEvol method corresponds to the Reward Weighted Regression (RWR) algorithm. See
* Peters, J., & Schaal, S. (2007, June). Reinforcement learning by reward-weighted regression for operational space control. In Proceedings of the 24th international conference on Machine learning (pp. 745-750).

for the initial reference, and many other papers for more details. If this intuition is correct, then this raises numerous questions about the method: is the author’s derivation original, how does it compare to other derivations of RWR? RWR is quite old and more advanced algorithms such as AWR or AWAC have been published since then: how does AgentEvol compare to these more recent methods?
- the authors suddenly introduce a non-negative reward halfway in the methods: they should rather have a "problem statement" section upfront where they specify that their method only works for non-negative rewards.
- I think the authors should integrate the results of Fig. 4 and Table 5 into Table 3, even if this results in partial lines.
- in the experiments, the authors only perform M=4 iterations. Is the learning process very long?
- could the authors be more explicit on the methodology in the experimental section: do they separate for each environment a train set and a test set?
- what about general purpose agents such as GATO, OpenVLA, etc. ?
- in Table 3, AgentEvol does not seem to improve at all with respect to BC_base in WD and WT. Can the authors explain why?
- training a LLAMA-2-Chat-7B agent (or any other large size LLM) with PPO is not an easy task. Can the authors be more explicit about their methodology here? Do they use LORA tuning? See several papers about applying RL on LLM agents in text worlds, such as for instance:
* Carta, T., Romac, C., Wolf, T., Lamprier, S., Sigaud, O., & Oudeyer, P. Y. (2023, July). Grounding large language models in interactive environments with online reinforcement learning. In International Conference on Machine Learning (pp. 3676-3713). PMLR.
* Tan, W., Zhang, W., Liu, S., Zheng, L., Wang, X., & An, B. (2024). True knowledge comes from practice: Aligning llms with embodied environments via reinforcement learning. arXiv preprint arXiv:2401.14151.
Being a reinforcement learning expert, I'm aware that lot of details are missing in the results corresponding to Figure 4. I suspect this is also the case for all other studies. The authors should devote a full section (in the appendix?) to provide all the details that are missing about the methodology for each method, including theirs.
- In more details, about Figure 4 studies, I would be glad to see e.g. the learning curves, to figure out which methods are stable or not. I would also be glad to see which environments are used for training, and which are kept for testing, particularly in Baby-AI and ALFworld that I know better. In short, the results of Figure 4 may deserve a paper in themselves, the high level overview that we get here does not facilitate scientific evaluation of the work done.
- What do “interaction rounds” mean in Appendix E.1? I’m surprised by the very low numbers.
- Are the SOTA models (as said in the introduction) mentioned in Table 3 still SOTA?
- the abstract and introduction insist on the fact that a lot of works use BC, but the selection of baselines does not support this message: some baselines are zero-shot, some others use RL, this is not much consistent.

# Typos, minor errors:
- there is a typo in the title (envronments)!
- p.5 the authors mention “golden trajectories” without specifying what it means
- p.5 Then, We → we
- p.5 rigorously filtered → filter (the rest is at the present tense)
- p.6 , We → we
- p.6 “The former step” → The first step. Then “The latter step” → The second step
- p.6 The step of J(q, \pi…) → the step of doing what exactly? Please be explicit.

**Details Of Ethics Concerns:**

Nothing specific

---

> ### Author Response · Authors · 2024-11-22
> **Response to Reviewer jPpZ [1/6]**
>
> Thank you for your detailed reading and valuable feedback. We are also grateful for your recognition of the impact and value of our framework. Your suggestions are really helpful for improving our writing, formatting, and overall structure. We will revise the manuscript based on your comments and do our best to address your concerns.
>
> ## 1. Question about the structure of the paper
>
> Thank you for your professional suggestions. We believe they are crucial for improving our writing and the structure of the paper. We have made the following modifications:
>
> - We add extensive experiments and analyses regarding the AgentEvol method, which are now included in the revised Appendix F.
> - We shorten the detailed description of the AgentGym framework, removing a substantial amount of engineering-related content from Section 3 and moving it to Appendix D.
> - We merge Section 2 and Section 4 to enhance the overall coherence of our paper.
> - Additionally, we would like to kindly point out that the AgentGym framework, database (including AgentTraj, AgentTraj-L, AgentEval), and the AgentEvol algorithm are all key components, specifically designed to facilitate the self-improvement of LLM-based agents across multiple environments. This has been explained in the fourth point of the General Response.
>
> We hope these modifications meet your expectations.
>
> ## 2. Question about the evaluation of AgentEvol.
>
> Due to space limitations, our evaluation of the AgentEvol method wasn’t fully comprehensive. We've added several experiments **in the first three points of the General Response** to better demonstrate its effectiveness.

---

> ### Author Response · Authors · 2024-11-22
> **Response to Reviewer jPpZ [2/6]**
>
> ## 3. Question regarding the RWR algorithm.
>
> Thank you for your insightful question. We would like to kindly clarify the following points:
>
> - The key focus of AgentGym is to build generally-capable agents that can self-evolve across various environments. Therefore, we first considered the popular online RL algorithm, PPO. However, current PPO implementations in the LLM field are mainly suited for single-turn conversation. They perform poorly on multi-turn, long-term agent tasks and pose engineering challenges.
> - **AgentEvol is derived from the Dayan & Hinton's framework[1, 2], and it is an adaptation and application based on the idea of Dayan & Hinton's framework in the LLM-based agent field**,  designed to enable agents to achieve self-improvement and interactive training across multiple environments and tasks. RWR [3] is also derived  from the Dayan & Hinton's framework, but its practical implementation is different from our method. Specifically, in AgentEvol, we set $r=0$ for failed trajectories, and $r=1$ for successful ones (see p.8 Line 416 in the revised version). This means that although AgentEvol is weighted by rewards, only successful trajectories are included in the loss calculation. Failed trajectories are assigned a weight of 0 in the policy optimization process and are excluded from the loss. For the solidness and thoroughness of our experiments, we evaluate the standard RWR baseline in the General Response, point 2.
> - Additionally, we highly acknowledge that AWR and AWAC are more advanced advantage-weighted RL algorithms. However, in our task setting, rewards are at the trajectory level, meaning the **agent only receives a reward at the end of a complete interaction trajectory**. As such, we cannot calculate stepwise advantages to guide optimization. This is why we did not compare against methods like AWR. (Of course, we could adopt a more fine-grained sampling process, starting from intermediate steps and using Monte Carlo to estimate stepwise values. We aim to provide a more detailed discussion in our future work.)
>
> ## 4. Question about "non-negative reward".
>
> Thank you for pointing out this issue. We apologize for the sudden introduction of "non-negative reward". On p.5 Line 251 (revised version), we mentioned the range of the final reward *r*, but we admit it wasn’t clear enough and could be easily overlooked. Based on your suggestion, we have updated the manuscript accordingly.
>
> We would also like to clarify that, since we aim to optimize across multiple environments and tasks, we have to standardize rewards from different environments to fit within a $[0,1]$ range for stable training. This might differ from traditional RL/agent practices, leading to some misunderstanding.

---

> ### Author Response · Authors · 2024-11-22
> **Response to Reviewer jPpZ [3/6]**
>
> ## 5. Question regarding the main results in Table3.
>
> We would like to kindly clarify the following:
>
> - In Table 3 (the main experiment table), we aim to show the testing and training results based on the AgentGym framework. Therefore, we compare with both open-sourced and close-sourced SOTA models. **The baselines in Table 3, as well as our AgentEvol method, are all task-agnostic.** We believe this comparison is fairer.
> - Regarding Fig. 4, due to limited computational resources and implementation difficulties, we are unable to run PPO experiments across all 14 environments. Therefore, we cannot merge these results with the main table. Additionally, to the best of our knowledge, PPO algorithms executed on LLM agent tasks currently are all task-specific. Running PPO across multiple environments is very challenging.
> - Regarding Table 5, **scaling (model size, data size) is a key focus in the** **LLM** **field**, so we present it in a separate table for discussion. Following Reviewer ZWi2's suggestion, we add a more detailed analysis and discussion on scalability (see Q1 regarding Reviewer Zwi2).
>
> ## 6. Question regarding M=4 iterations.
>
> Thank you for your thoughtful question. The evolution process requires a substantial runtime. On eight A100-80GB GPUs, **a single iteration requires approximately 35 hours**. This includes about 3 hours for the learning step, 6 hours for evaluation across all environments, and 26 hours for the exploration step, which involves multiple interactions between the agent and the environment. A summary of exploration times across environments is shown below:
>
> |                       | WS   | ALF  | TC   | Sci  | Baby | WT   | MV   | TL   | MZ     | WD     | BD       | Total |
> | --------------------- | ---- | ---- | ---- | ---- | ---- | ---- | ---- | ---- | ------ | ------ | -------- | ----- |
> | Exploration Step Time | 3h   | 4h   | 2h   | 4h   | 1h   | 3h   | 3h   | 3h   | 10mins | 20mins | 2h30mins | 26h   |
>
> Although increasing the number of iterations $M$ yields performance gains, it gradually converges in later iterations. As described in Section 5.3, we perform an ablation study on the iteration number, and to further validate this, we conduct additional experiments for $M=6$. The performance results are detailed below:
>
> |       | ALF   | Baby  | TC    | Sci   |
> | ----- | ----- | ----- | ----- | ----- |
> | $M=4$ | 88.00 | 82.70 | 64.00 | 38.00 |
> | $M=5$ | 88.50 | 82.00 | 62.00 | 37.00 |
> | $M=6$ | 89.00 | 81.88 | 63.00 | 37.09 |
>
> The improvement from $M=4$ to $M=5$ and $M=6$ is minimal, with less than a 1% increase in most environments. Given the substantial runtime required for each additional iteration and the limited performance gains, we select $M=4$ as a trade-off between computational efficiency and performance.

---

> ### Author Response · Authors · 2024-11-22
> **Response to Reviewer jPpZ [4/6]**
>
> ## 7. Question about train sets and test sets.
>
> We apologize for any confusion caused by our writing. The right half of Table 2 on p.4 (revised version) shows the dataset sizes for each environment. **We do separate a train set and a test set for each environment.** Here, "Traj. Size" represents the size of the small training set (AgentTraj), "Traj-L Size" represents the size of the large training set (AgentTraj-L), and "Eval. Size" represents the size of the test set. If the training set size is 0 for an environment (e.g., WA, AM, and ST), it means that the environment is used solely for testing.
>
> ## 8. Question about general purpose agents.
>
> Thank you for your insightful question. It is very helpful for our future research and extensions. To the best of our knowledge:
>
> - **Gato** is a very powerful, multi-modal, multi-task, and multi-embodiment generalist agent. Meanwhile, AgentGym is specifically designed for complex, long-term, multi-turn sequential decision-making tasks. We are excited about the possibility of integrating such a strong agent into our framework to evaluate its capabilities.
> - **OpenVLA** is a well-known VLA model designed for real-world robotic tasks. Currently, AgentGym is still a language-based framework. In this context, instructions, actions, and observations are all language-based and textual. Additionally, robotic tasks differ slightly from the focus of our LLM-based agents (see Q1: "Why do we select these environments" in response to Reviewer VNyh). We will include these multi-modal agents in our future work.
>
> ## 9. Question regarding the lack of improvement in certain environments.
>
> Please refer to the first point of the General Response. We also kindly emphasize that AgentEvol is stronger than $BC_{base}$ in 8 environments and stronger than $BC_{large}$ in 7 environments.
>
> ## 10. Question about implementation details of PPO and other methods.
>
> We have provided an analysis and discussion in the second point of the General Response, **and included the implementation details of all methods in Appendix E**.

---

> ### Author Response · Authors · 2024-11-22
> **Response to Reviewer jPpZ [5/6]**
>
> ## 11. Question about “interaction rounds” mean in Appendix E.1.
>
> Thank you for your question. We apologize for the confusion caused by our description. "Interaction rounds" in Appendix E.1 (now Appendix F.1 in the revised version) refers to **the average number of interactions required for an LLM-based agent to complete a task in the environment**. This is the same with the metric provided in the last column of Table 2 (Statistics of AgentTraj-L).
>
> In each interaction round, the environment returns the current observation, and the agent takes actions based on this feedback. **The number of interaction rounds typically depends on the task** **goal** **in different environments and the efficiency of the agent.** For example, in the WebShop, the interactions usually include the following rounds:
>
> - Agent executes "Search[something]" on the webpage, and the environment returns search results.
> - Agent executes "Click[some button]" to select an item, and the environment returns the product detail page.
> - Agent selects the product attributes.
> - Agent selects the "Buy" button, and the environment returns the reward.
>
> Additionally, for the BIRD, which is a single-round SQL programming task, the agent only needs to perform one action on the database, so the number of interaction rounds for the BIRD task is set to 1.
>
> **This may differ significantly from traditional RL agent tasks**. In LLM-based agent tasks, due to the limitation of model's input length, an average task is typically solved in 10~20 rounds of text interactions. We apologize for the confusion this may have caused.
>
> ## 12. Question regarding SOTA models in Table3.
>
> The baselines we compare in Table 3 are still SOTA models. We would like to clarify the following:
>
> - Regarding close-sourced models, we test GPT-4, Claude 3.5, and Deepseek-chat. We expect to test the latest O1 models in the future when sufficient usage and resource are available.
> - Regarding open-sourced models, **we discuss the relevant works in the LLM-based agent field** (see Table 1 for details):
>   - AgentBench, AgentBoard, and AgentOhana [4, 5, 6]create excellent benchmarks to evaluate LLMs as agents, but they do not train specific agent models.
>   - Pangu-Agent: No publicly available model provided.
>   - AgentLM from AgentTuning [7] is an early milestone in the LLM-based agent field. It trains agent models of various sizes for multiple tasks and still shows strong performance. **We believe it is appropriate to select AgentLM as** **SOTA** **models**.
>   - Llama-2-chat is not listed as SOTA models. As one of the well-regarded models and our backbone model, we also test it.
>
> We expect to include more models in our work, e.g., Qwen, Gemma,  along with more of the latest closed-source models.
>
> #

---

> ### Author Response · Authors · 2024-11-22
> **Response to Reviewer jPpZ [6/6]**
>
> ## 13. Question about BC baselines.
>
> Thank you for your professional question. We would like to clarify the following:
>
> - Zero-shot and few-shot are evaluation approaches for agents, while BC and RL are training approaches. Therefore, zero-shot evaluation is decoupled from the specific training methods.
> - **In our experiments, we do include three BC baselines: $BC_{base}$, $BC_{large}$, and AgentLM**. All of them have been trained on a large number of agent trajectories. Models trained with BC already possess instruction-following capabilities and can distinguish between different tasks based on the system prompt and instruction. **Therefore, we evaluate these models, including AgentEvol, in a zero-shot setting.**
> - Additionally, many works still use BC, such as AutoWebGLM[8] and ToolLLM[9]. However, we do not include them in our baselines for comparison, as these models are task-specific or environment-specific. In our main experiment, our model demonstrates general capabilities across 11 environments.
>
> ## 14. Question regarding typos and minor errors.
>
> Thank you for your careful review. We have corrected all the typos in the manuscript.
>
> - Re: p.5 "golden trajectories". We apologize for the imprecise wording. What we intended to convey is a meaning similar to "golden label". In the BIRD-SQL environment, it includes a crowdsourced text-to-SQL dataset, where the questions and corresponding SQL answers are already provided. An example is as follows. We call this data "golden trajectories (labels)" because these high-quality data don't require any further annotation. We have corrected this term in the revised manuscript. Thank you for your careful review.
>
>   ```sql
>   SELECT T3.name
>   FROM student AS T1 INNER JOIN registration AS T2 ON T1.student_id = T2.student_id INNER JOIN course AS T3 ON T2.course_id = T3.course_id
>   WHERE T1.type = 'UG' ORDER BY T2.sat
>   DESC LIMIT 1
>   ```
>
> - Re: p.6 "The step of J(q, \pi...)". Our wording may have caused some confusion. In Line 313 and Line 315 on p.6, 'The former step' and 'The latter step' refer to the two parts before and after the minus sign in Eq(5) (i.e., ${\\mathbb{E}_q [\\log p(O=1|\\tau)]}$ and $\text{KL}[q(\tau)||\pi\_{\theta}(\tau)]$    ). We aim to provide a clearer and more intuitive explanation of these two parts, so that readers who are not familiar with RL can better understand some of AgentEvol's theoretical methods. We have followed your suggestion and updated the manuscript accordingly.
>
> **References**
>
> [1] Dayan P, Hinton G E. Using expectation-maximization for reinforcement learning[J]. Neural Computation, 1997, 9(2): 271-278.
>
> [2] Abdolmaleki A, Springenberg J T, Tassa Y, et al. Maximum a posteriori policy optimisation[J]. ICLR 2018.
>
> [3] Peters, Jan, and Stefan Schaal. "Reinforcement learning by reward-weighted regression for operational space control." *Proceedings of the 24th international conference on Machine learning*. 2007.
>
> [4] Liu, Xiao, et al. "Agentbench: Evaluating llms as agents." *arXiv preprint arXiv:2308.03688* (2023).
>
> [5] Ma, Chang, et al. "AgentBoard: An Analytical Evaluation Board of Multi-turn LLM Agents." *arXiv preprint arXiv:2401.13178* (2024).
>
> [6] Zhang, Jianguo, et al. "AgentOhana: Design Unified Data and Training Pipeline for Effective Agent Learning." *arXiv preprint arXiv:2402.15506* (2024).
>
> [7] Zeng, Aohan, et al. "Agenttuning: Enabling generalized agent abilities for llms." *arXiv preprint arXiv:2310.12823* (2023).
>
> [8] Lai, Hanyu, et al. "AutoWebGLM: A Large Language Model-based Web Navigating Agent." *Proceedings of the 30th* *ACM* *SIGKDD Conference on Knowledge Discovery and* *Data Mining*. 2024.
>
> [9] Qin, Yujia, et al. "Toolllm: Facilitating large language models to master 16000+ real-world apis." *arXiv* *preprint* *arXiv:2307.16789* (2023).

---

> > ### Comment · Area_Chair_kPX6 · 2024-11-24
> > **Please respond to rebuttal ASAP**
> >
> > Dear reviewer,
> > The process only works if we engage in discussion. Can you please respond to the rebuttal provided by the authors ASAP?

---

> > ### Comment · Reviewer_jPpZ · 2024-11-25
> > **Statistical significance ?**
> >
> > How many seeds did the authors use in the additional results they provide in their general response ?
> >
> > My feeling is that the authors are still trying to embrace too many things in a single paper, at the cost of low scientific accuracy.

---

> > > ### Author Response · Authors · 2024-11-27
> > > **Response to Reviewer jPpZ's Official Comment 2 [1/2]**
> > >
> > > Thank you very much for recognizing our work and increasing your score. We provide the following clarifications, hoping they will address your concerns.
> > >
> > > ## 1. Re: the question of random seeds of additional results
> > >
> > > Thank you for your feedback. In the following, we will carefully respond to your questions.
> > >
> > > - To enable LLM-based agents' self-improvement across multiple environments, our framework covers diverse environments and tasks. Due to the high computational demands based on LLM, we make significant efforts to manage and organize computational resources and provide additional results. Despite time and resource constraints, for experiments that do not require exploration (i.e.,  BC and DPO), we use **3 seeds** and report the average results to ensure stability. In contrast, for experiments involving exploration (e.g., AgentEvol and PPO), we fix the seeds due to the substantial computational overhead associated with LLMs' next-token prediction.  This aligns with common practices in the LLM field, especially with limited resources and large amout of experiments  [1, 2, 3].  As mentioned in our previous response (point 6), we provide detailed information on computational costs: AgentEvol requires about **35 hours per iteration on 8 A100-80GB GPUs**. Moreover, PPO in the WebShop environment takes approximately **3 hours 50 minutes per epoch** with the same setup for **10** epochs.
> > > - Considering that this is a common challenge faced by LLM researchers, we kindly request for your understanding. In response to your concern, we expect to conduct additional experiments with different seeds for exploration-based tasks in the next revised manuscript.

---

> ### Author Response · Authors · 2024-11-27
> **Response to Reviewer jPpZ's Official Comment 2 [2/2]**
>
> ## 2. Re: the suggestion of adjustments to organization of the paper contents.
>
> - To achieve our research goal of exploring the **self-improvement of LLM-based agents in diverse environments**, we do incorporate several components, including the AgentGym framework,  dataset, benchmark suite, and self-improvement methods across diverse environments. These components are crucial building blocks for exploring our research question.
> - Following your suggestion, we have moved certain descriptions and engineering details to the Appendix and merged Sections 2 and 4 to better organize the paper, making it more focused on the scientific contribution. We will continue to improve our manuscript following your advice.
> - In response to your new suggestion, we will make full use of computational resources and conduct additional experiments to further strengthen the solidness of our work.
>
> **References**
>
> [1] Zeng, Aohan, et al. "Agenttuning: Enabling generalized agent abilities for llms." *arXiv* *preprint* *arXiv:2310.12823* (2023).
>
> [2] Lai, Hanyu, et al. "AutoWebGLM: A Large Language Model-based Web Navigating Agent." *Proceedings of the 30th* *ACM* *SIGKDD Conference on Knowledge Discovery and* *Data Mining*. 2024.
>
> [3] Chen, Zehui, et al. "Agent-FLAN: Designing Data and Methods of Effective Agent Tuning for Large Language Models." *arXiv* *preprint* *arXiv:2403.12881* (2024).

---

### Official Review · Reviewer_PSRy · 2024-10-30

**Soundness:** 2
**Presentation:** 3
**Contribution:** 2
**Rating:** 5
**Confidence:** 4

**Summary:**

The authors propose AgentGym, an interactive framework with various environments, which includes a database with expanded instructions, high-quality trajectories AGENTTRAJ and AGENTTRAJ-L, and a benchmark suite AGENTEVAL. The authors also design AgentEvol, an exploration-learning self-evolution method. The authors conduct experiments

**Strengths:**

As in the summary, the paper is very comprehensive, including many aspects about LLM-based agents.

**Weaknesses:**

The paper include both an interactive framework AgentGym, which is composed of several components, and self-evolution method AgentEvol.
It is not clear what the focus is.

ReAct may not be the best reference approach to build LLM-based agents.
In industrial practice, it has been adopted for around 1.5 years. What are the successful LLM-based agents?
In academic research, there are recent papers for better reasoning / planning capacity, e.g.,
Quiet-STaR: Language Models Can Teach Themselves to Think Before Speaking, COLM 2024
AlphaZero-Like Tree-Search can Guide Large Language Model Decoding and Training, ICML 2024

There are actually papers criticizing the reasoning / planning capacity of LLMs, which is the foundation of agents, e.g.,

GSM-Symbolic- Understanding the Limitations of Mathematical Reasoning in Large Language Models https://arxiv.org/abs/2410.05229

LLMs Still Can't Plan; Can LRMs? A Preliminary Evaluation of OpenAI's o1 on PlanBench
https://arxiv.org/abs/2409.13373



Agents are about optimal sequential decision making.
Why only success rate?
Does the LLM-based agent community care about optimal solutions?


AGENTEVOL is one particular RL approach, which may fit some tasks but not others.
As a comprehensive framework, it is desirable to explore various RL approaches.

LLM-based agents are not able to handle the basic grid world problems.
How about including some problems the current RL can handle well in the benchmark?

Figure 1 appears cool (for something like a cartoon book). However, it does not provide more info than the caption, i.e., it is not quite necessary, for a top tier AI conference paper.

**Questions:**

See weakness

---

> ### Author Response · Authors · 2024-11-22
> **Response to Reviewer PSRy [1/4]**
>
> Thank you very much for your valuable suggestions and questions. We will improve our manuscript based on your feedback.
>
> ## 1. Question regarding the focus of our paper.
>
> Thank you for your valuable question. We believe it's important to clarify the motivation and focus of our paper. The AgentGym framework and the AgentEvol algorithm are both key components, designed to facilitate the self-evolution of LLM agents across multiple environments. We have explained this in the fourth point of the General Response. We hope this helps clarify your concerns.
>
> ## 2. Question about ReAct and approaches to building LLM-based agents.
>
> Thank you for your insightful question. We are glad to discuss this with you.
>
> - First, we acknowledge that ReAct may not be the best reference method for constructing a general-purpose, powerful LLM-based agent. Based on our understanding, ReAct is a reasoning format that guides and facilitates the interaction between the agent and environments.
> - In industrial practice, to the best of our knowledge, some representative and successful LLM-based agents include "Computer Use" proposed by Anthropic, as well as GitHub Copilot and Cursor. We do not have access to the details of how these successful applications were constructed. **However, we believe that the reasoning format is just one of the key factors behind the success of these agent applications, rather than the only factor.** A powerful LLM-based agent is one that has a decision-making core based on a large language model (LLM), which extends its input (e.g., various types of textual input) and actions (e.g., calling APIs, using tools, performing embodied actions, etc.). Such agents require insightful design across multiple aspects and dimensions [1,2,3,4].
> - In academic research, with the rise of O1, we have also noticed excellent works on reasoning, such as Quiet-STaR and AlphaZero-like Tree-Search. However, we would like to clarify that these works are decoupled from our AgentGym framework. AgentGym is a unified framework for training, testing, and sampling. **Therefore, whether your agent uses a ReAct approach, Quiet-STaR, or MCTS for reasoning, it can integrate closely with our framework**. We help test the final performance of the agent and can also use high-quality reasoning data sampled by the agent for subsequent training.
> - **Notably, the reason we employed the ReAct format in this work is largely to maintain consistency with other works in the LLM-based agent field (e.g., AgentBench, AgentTuning, AgentFlan, AgentOhana, etc.) [5,6,7,8].** This not only facilitates fair comparison but also enables the community to conduct in-depth research and exploration based on this work.
> - Finally, we will incorporate more reasoning/planning paradigms into our future work, enhancing the comprehensiveness of our framework.

---

> > ### Comment · Reviewer_PSRy · 2024-11-24
> > **Self-improvement based on a fixed, imperfect LLM is not a valid approach.**
> >
> > Thanks authors for the explanation.
> >
> > If you treat GitHub Copilot and Cursor as "agents", you may treat any AI/LLM applications as agents.
> >
> > There is a fundamental issue with ReAct: it treats an LLM as an oracle.
> > No LLM is perfect, so treating an LLM as an oracle is actually incorrect.
> >
> >
> > LLMs are not perfect, so that they can not guarantee reliable information.
> > RL follows a trial and error approach, by interacting with the environment, but with a true / perfect world model implicitly.
> > Self-improvement based on a fixed, imperfect LLM is not a valid approach.

---

> ### Author Response · Authors · 2024-11-22
> **Response to Reviewer PSRy [2/4]**
>
> ## 3. Question regarding the reasoning / planning capacity of LLMs.
>
> We fully agree with your point that the reasoning/planning capacity of LLMs is the foundation of LLM-based agents.
>
> - To the best of our knowledge: GSM-Symbolic is an advanced benchmark created based on GSM8K symbolic templates. LLMs' performance tends to be unstable when changing the format or values of the problem. One hypothesis is that LLMs may have learned fixed patterns from the training data through pattern recognition. PlanBench is a time-tested dataset for evaluating LLM planning abilities. Since GPT-3, many models have made little progress on this benchmark, and O1 has not yet met expectations.
> - These works have proposed and tested their hypotheses through extensive experiments. In light of the experimental results, we must admit that many current LLMs may not yet meet the standard in terms of reasoning abilities. However, these works also acknowledge that O1 outperforms other models. Which indicates the potential benefits of expanding the exploration space and enhancing test-time scaling.
> - At the same time, building such a foundation is a gradual process. Without a very strong LLM as the base model, the agents constructed on it may not perform well. For example, in our experiments with Llama-2-7B. However, through optimization with the AgentEvol method, the reasoning ability of agents on specific tasks can match or even surpass that of originally stronger models.
>
> **The key focus of AgentGym is to enhance reasoning/planning abilities through self-evolution across different environments**. The OOD experiments we have added in the third point of the General Response also indicate that improvements in reasoning abilities have a degree of generalization.
>
> ## 4. Question about metrics: optimal solutions or success rate?
>
> We kindly provide the following clarifications:
>
> - **In the LLM-based agent community, researchers focus more on task success/reward rate because they want the agent to successfully accomplish specific task  [5, 6, 9].** For each environment, we list the corresponding metric in p.5 Table 2. $Reward = 1$ is defined as *Success*.
> - However, we fully agree with your point. For an agent, focusing on optimal sequential decision-making is one of the key metrics. This means the agent can maximize long-term returns. Relying solely on success rate may overlook this aspect.
> - In our preliminary experiments, we observe that in simpler environments like Webshop, the agent can achieve high rewards. However, it may not reach the final goal, i.e., the success rate remains low. **Our experience is that in LLM-based agent tasks**, the optimal solution always achieves the highest reward (e.g. success), but higher rewards do not necessarily indicate an optimal solution. **This may differ from traditional RL settings.**
> - At the same time, using success rate as a metric also helps **mitigate reward hacking**, preventing the agent from performing irrelevant actions to gain high scores.

---

> > ### Comment · Reviewer_PSRy · 2024-11-24
> > **Misunderstandings with optimal solution and reward function.**
> >
> > The authors made several mistakes:
> >
> > - "In the LLM-based agent community, researchers focus more on task success/reward rate because they want the agent to successfully accomplish specific task."
> >
> > Accomplishing a task if far from optimizing a task. Consider a simple example: reaching a destination vs reaching a destination via shortest path.
> >
> > - "Our experience is that in LLM-based agent tasks, the optimal solution always achieves the highest reward (e.g. success), but higher rewards do not necessarily indicate an optimal solution. This may differ from traditional RL settings."
> >
> >
> > With a well defined reward function, an optimal solution is optimal.
> >
> > - "At the same time, using success rate as a metric also helps mitigate reward hacking, preventing the agent from performing irrelevant actions to gain high scores."
> >
> > Ditto.

---

> > > ### Author Response · Authors · 2024-11-25
> > > **Response to Reviewer PSRy's Official Comment 2 [1/2]**
> > >
> > > Thanks for your discussion. We would like to kindly clarify and point out that:
> > >
> > > ## 1. Clarifying our research scope again.
> > > Our research centers on **large language model-based agents for multi-round decision-making tasks (i.e., large language models for agent tasks)**, with our motivation rooted in this area rather than in the traditional RL setting.
> > >
> > > ## 2. Re: "Self-improvement based on a fixed, imperfect LLM is not a valid approach".
> > > - **Imperfect LLMs and world models are not the focus of our research, nor are they the focus of the LLM-based agent field.** Our goal is to enhance the capabilities of LLM-based agents, achieving higher accuracy and reward. This allows weaker and imperfect models to gradually improve.
> > > - **Regarding ReAct, it is a commonly used approach in the LLM-based agent field and considered a common practice**. As mentioned in our first response, one of the reasons we adopted the ReAct format in this work is to maintain consistency with other works in the LLM-based agent field (e.g., AgentBench, AgentTuning, AgentFlan, ArCHer, etc.) [1,2,3,4]. This is crucial for research continuity and ease of use within the community.
> > >
> > > ## 3. Re: "with mixed performance, AgentEvol is not quite a contribution".
> > > - Regarding mixed performance, we reiterate the results from General Response Point 1:
> > >   - **AgentEvol on multi-tasks vs. AgentEvol on single-task**: **The improvement of general capabilities is a critical challenge for the LLM-based agent and LLM communities** [5,6,7]. Experimental results show that AgentEvol on single tasks outperforms AgentEvol on multi-tasks. Achieving performance improvements across multiple tasks often requires compromising performance on individual tasks. This is a common and reasonable phenomenon in the LLM field and even the broader NLP domain [8,9,10,11,12,13]. **AgentEvol attains performance comparable to task-specific methods on each task**, which is already a significant achievement.
> > >   - **AgentEvol on multi-tasks vs. BC on multi-tasks**: In the case of multi-task mixed training, AgentEvol outperforms  $BC_{base}$, $BC_{large}$, and AgentLM on most tasks. **Specifically, it is not weaker than or surpasses $BC_{base}$ on all 11/11 tasks and $BC_{large}$ on 8/11 tasks.** Due to the limitations of the base model’s capabilities and prior knowledge, the quality of self-synthesized data by the model often struggles to surpass that of SOTA model annotations. Thus, we acknowledge that it is challenging to exceed SOTA models in every environment. However, achieving this level of performance on a relatively weaker base model with less advanced prior knowledge represents a critical milestone.
> > > - Additionally，**AgentGym is not just an evaluation framework: It is a unified framework designed for LLM-based agents, integrating training, sampling, and evaluation in-the-loop.** It offers the LLM-based agent community a pipeline for online multi-turn interactions between LLM-based agents and diverse, complex environments. We also provide a comprehensive user guide covering various aspects, such as BC training, self-improvement, preference alignment training, evaluation, and trajectory data collection.

---

> > > ### Author Response · Authors · 2024-11-25
> > > **Response to Reviewer PSRy's Official Comment 2 [2/2]**
> > >
> > > ## 4. Re: "Misunderstandings with optimal solution and reward function".
> > > - In LLM-based agent environments, most existing reward functions calculate rewards based on the ratio of **"completed goals / total goals"**, which has a clear gap with the metric you mentioned, "reaching a destination via shortest path." We believe this reflects a fundamental difference in the metrics prioritized by RL settings and LLM-based agent settings.
> > > - **We want to emphasize that researchers in the LLM community prioritize whether the model can complete the task.** If finding the optimal path is required, it becomes a more challenging problem and reduces the diversity of LLM sampling. **Therefore, such metrics are typically avoided in the LLM community, which instead focuses on relatively simple metrics like success accuracy and reward**. We have also cited many related works, such as AgentTuning, AgentBench, AgentFlan and etc., that use success/reward rate as their metric.
> > > - Furthermore, one of AgentGym's contributions is unifying the existing, diverse types of environments. Modifying the reward function for each environment is beyond the scope of our work.
> > >
> > > **References**
> > >
> > > [1] Liu, Xiao, et al. "Agentbench: Evaluating llms as agents." *arXiv preprint arXiv:2308.03688*(2023).
> > >
> > > [2] Zeng, Aohan, et al. "Agenttuning: Enabling generalized agent abilities for llms." ACL, 2024.
> > >
> > > [3] Chen, Zehui, et al. "Agent-FLAN: Designing Data and Methods of Effective Agent Tuning for Large Language Models." ACL, 2024.
> > >
> > > [4] Zhou, Yifei, et al. "Archer: Training language model agents via hierarchical multi-turn rl." ICML, 2024.
> > >
> > > [5]  Wooldridge, Michael, and Nicholas R. Jennings. "Intelligent agents: Theory and practice." *The knowledge engineering review* 10.2 (1995): 115-152.
> > >
> > > [6] Silver, David, et al. "Mastering the game of go without human knowledge." *nature* 550.7676 (2017): 354-359.
> > >
> > > [7] Reed, Scott, et al. "A generalist agent." *arXiv preprint arXiv:2205.06175* (2022).
> > >
> > > [8] Standley, Trevor, et al. "Which tasks should be learned together in multi-task learning?." International conference on machine learning. PMLR, 2020.
> > >
> > > [9] Zhang, Yu, and Qiang Yang. "A survey on multi-task learning." IEEE transactions on knowledge and data engineering 34.12 (2021): 5586-5609.
> > >
> > > [10] Wang, Zhen, et al. "Multitask Prompt Tuning Enables Parameter-Efficient Transfer Learning." The Eleventh International Conference on Learning Representations.
> > >
> > > [11] Dai, Damai, et al. "Deepseekmoe: Towards ultimate expert specialization in mixture-of-experts language models." *arXiv preprint arXiv:2401.06066*(2024).
> > >
> > > [12] Jiang, Albert Q., et al. "Mixtral of experts." *arXiv preprint arXiv:2401.04088*(2024).
> > >
> > > [13] Ye, Jiasheng, et al. "Data mixing laws: Optimizing data mixtures by predicting language modeling performance." *arXiv preprint arXiv:2403.16952* (2024).

---

> > > > ### Comment · Reviewer_PSRy · 2024-11-30
> > > > **Constructive suggestion**
> > > >
> > > > The following are a sample of papers which may help you reconsider the method and direction for LLM agents.
> > > >
> > > > Position: LLMs Can’t Plan, But Can Help Planning in LLM-Modulo Frameworks, ICML 2024
> > > >
> > > > On the Brittle Foundations of ReAct Prompting for Agentic Large Language Models, arXiv 2024
> > > >
> > > > Chain of Thoughtlessness? An Analysis of CoT in Planning
> > > >
> > > > Can Large Language Models Really Improve by Self-critiquing Their Own Plans?
> > > >
> > > > AI Agents That Matter, arXiv 2024
> > > >
> > > > When is Tree Search Useful for LLM Planning? It Depends on the Discriminator, ACL 2024
> > > >
> > > > TravelPlanner: A Benchmark for Real-World Planning with Language Agents, ICML 2024
> > > > “the current language agents are not yet capable of handling such complex planning tasks-even GPT-4 only achieves a success rate of 0.6%”
> > > >
> > > > There are also lots of papers discussing the limitations of LLMs, e.g.,
> > > >
> > > > Faith and Fate: Limits of Transformers on Compositionality, NeurIPS 2023
> > > >
> > > > Embers of Autoregression: Understanding Large Language Models Through the Problem They are Trained to Solve, PNAS 2024
> > > >
> > > > Can Large Language Models Infer Causation from Correlation? ICLR 2024

---

> > > > > ### Comment · Reviewer_PSRy · 2024-11-30
> > > > >
> > > > > AI researchers should not attempt to justify
> > > > > - reliance on unreliable information
> > > > > - unsound objective function, like success rate, but not an optimal one

---

> ### Author Response · Authors · 2024-11-22
> **Response to Reviewer PSRy [3/4]**
>
> ## 5. Question about exploring various RL approaches.
>
> We have added experiments and a detailed discussion to address your concern in the second point of the General Response. Additionally, in General response point 1, we analyze why AgentEvol performs better in certain environments (e.g., WebShop) and worse in others (e.g., Weather). We hope this analysis helps clarify your concerns.
>
> ## 6. Questions regarding grid world problems.
>
> Yes, we agree that basic grid world problems are essential for LLM-based agents, as these game environments provide an excellent testbed for verifying an agent's understanding of the environment and decision-making abilities.
>
> - When building the AgentGym framework, we did consider this. We introduced a similar environment, called "Maze". It is a maze problem described in text form. The agent can access information about its current position, goal position, and surrounding walls. We provide an example [https://anonymous.4open.science/r/AgentGym-ICLR-rebuttal-1B8D/Maze_example.png](https://anonymous.4open.science/r/AgentGym-ICLR-rebuttal-1B8D/Maze_example.png) to help you better understand this task.
> - If you have any more specific suggestions, we would be happy to include these environments in future versions of AgentGym to make our framework more complete.

---

> > ### Comment · Reviewer_PSRy · 2024-11-24
> > **with mixed performance, AgentEvol is not quite a contribution**
> >
> > "we analyze why AgentEvol performs better in certain environments (e.g., WebShop) and worse in others (e.g., Weather)"
> >
> > If the proposed self-evolution method AgentEvol has mixed performance, it is not quite a contribution, esp. for AgentGym, an evaluation framework. Going back to the first question, it is better off focusing on the framework, and make it convenient for users of the framework to compare many popular approaches.
> >
> > BTW, it is likely very hard to design a sound, general agent method. In RL, general methods like DQN and PPO are not the best for all problems.

---

> ### Author Response · Authors · 2024-11-22
> **Response to Reviewer PSRy [4/4]**
>
> ## 7. Question about Figure 1.
>
> Thank you for your valuable feedback. It has been a great encouragement to us. Indeed, the information provided by this figure is almost identical to the caption. However, we hope that Figure 1 is not just for visual appeal.
>
> - On one hand, we aim to help readers by offering a more intuitive understanding of the Self-evolution process through a comic-like illustration, especially those unfamiliar with LLM-based agent reasoning.Different parts of Figure 1 (such as the transition from classroom to wild environments) can help readers **better imagine how LLM-based agents learn and progress in various environments**.
> - On the other hand, the process shown in Figure 1 has been **validated in the subsequent experiments of the paper**. We use a similar process to train the agent: 1. Learning basic skills through Behavior Cloning, and 2. Continuous exploration and evolution in unseen environments and tasks.
> - Finally, since AgentGym framework is a little bit engineering, we also hope this figure provides some enjoyment for readers while going through the paper. Thank you again for your kind suggestion and understanding :-).
>
> Thank you once again for your valuable comments! If you find our clarifications satisfactory, we kindly ask for your consideration in adjusting your scores accordingly.
>
> **References**
>
> [1] Wooldridge M, Jennings N R. Intelligent agents: Theory and practice[J]. The knowledge engineering review, 1995, 10(2): 115-152.
>
> [2] Goodwin R. Formalizing properties of agents[J]. Journal of Logic and Computation, 1995, 5(6): 763-781.
>
> [3] Xi Z, Chen W, Guo X, et al. The rise and potential of large language model based agents: A survey[J]. arXiv preprint arXiv:2309.07864, 2023.
>
> [4] Wang L, Ma C, Feng X, et al. A survey on large language model based autonomous agents[J]. Frontiers of Computer Science, 2024, 18(6): 186345.
>
> [5] Liu, Xiao, et al. "Agentbench: Evaluating llms as agents." *arXiv preprint arXiv:2308.03688* (2023).
>
> [6] Zeng, Aohan, et al. "Agenttuning: Enabling generalized agent abilities for llms." *arXiv preprint arXiv:2310.12823* (2023).
>
> [7] Chen, Zehui, et al. "Agent-FLAN: Designing Data and Methods of Effective Agent Tuning for Large Language Models." *arXiv preprint arXiv:2403.12881* (2024).
>
> [8] Zhang, Jianguo, et al. "AgentOhana: Design Unified Data and Training Pipeline for Effective Agent Learning." *arXiv preprint arXiv:2402.15506* (2024).
>
> [9] Ma, Chang, et al. "AgentBoard: An Analytical Evaluation Board of Multi-turn LLM Agents." *arXiv* *preprint* *arXiv:2401.13178* (2024).

---

### Official Review · Reviewer_VNyh · 2024-11-01

**Soundness:** 3
**Presentation:** 2
**Contribution:** 2
**Rating:** 5
**Confidence:** 3

**Summary:**

This paper proposes an interactive framework that includes diverse scenarios and environments for LLM-based agents (AGENTGYM) and investigates self-evolution for LLM-based agents (AGENTEVOL) based on AGENTGYM.

**Strengths:**

1. The problem that this paper investigates, how to make LLMs explore and evolve themselves across diverse environments, is interesting.
2. The framework and the self-evolution method that this paper proposed make sense.

**Weaknesses:**

1. This paper would be better if the authors could provide better writing. In general, it would be better if the authors could discuss more about why they do that and mention some key technical features in the paper instead of putting a lot of technical details in the paper. For example, in section 3, the authors could discuss more about why you selected these environments and the AGENTGYM framework. The author could discuss less about the technical details and move them into the supplementary. If the author thinks some technical details are the key contributions of this paper,  the authors might make them as a separate section or subsection and discuss more details.
2. In the experiments section, the author could discuss more about the results they got. For example, in Table 3, why does their method work better than other LLMs in some environments (i.e., WebShop) while performing badly in other scenarios (i.e., Weather)? It would make this paper stronger if the authors could propose some hypothesis based on results instead of just describing what the table shows.

**Questions:**

Please check the weakness section.

---

> ### Author Response · Authors · 2024-11-22
> **Response to Reviewer VNyh [1/2]**
>
> Thank you very much for your valuable suggestions and questions. We hope our response addresses your concerns:
>
> ## 1. Question regarding the starting point of AgentGym framework and environments.
>
> Thank you for your valuable feedback. Your kind suggestions regarding the writing of our paper are very important to us. We will incorporate these changes into the new manuscript.
>
> To address your feedback, we have added a discussion on "The choice of environments and tasks in AgentGym" in Section 3 to clarify our starting point. Additionally, we have reduced some of the technical details and moved them to Appendix D.
>
> **The starting point of designing the AgentGym framework?** The key research question we explore in this paper is: How can we build generally-capable agents that can handle multi-turn decision-making tasks across various interactive environments? We have elaborated on this in the fifth point of the General Response; please refer to the discussion there.
>
> **Why do we select these environments?** We provide an explanation from two perspectives:
>
> - **Firstly, from the definition,** an LLM-based agent is an agent with a decision-making core based on a large language model (LLM), extending its input (e.g., different types of textual input) and action (e.g., calling APIs, using tools, performing embodied actions, etc.) [1, 2, 3, 4]. Therefore, the capabilities required for an LLM-based agent include [3, 4, 5]:
>   - **Input** **Side**: The agent needs to have the ability to receive different types of textual input or observations, such as plain text, HTML, code, etc. Therefore, the environments and tasks we use need to include these different inputs. Thus, we include some textual environments (e.g., ALF, TC), some web environments (e.g., WS, WA), and code environments (e.g., BD).
>   - **Decision-Making Side**: Centered around the LLM, we need to improve/evaluate the agent's basic knowledge, abilities in reasoning, planning, and understanding environmental information. Therefore, the tasks we use need to be challenging, requiring the LLM to complete them through reasoning and planning. For example, in the WS task, when faced with a web page containing many products, an LLM-based agent needs to guide its next actions through reasoning and information extraction. In the Sci task, when facing observations in different rooms, the LLM-based agent needs to read manuals, find raw materials, and synthesize items, which often requires basic common knowledge, strong planning and environmental information filtering and understanding abilities.
>   - **Action Side**: We need to improve/evaluate the agent's ability to output different forms of actions, such as plain text, tool and API calls, code, embodied actions, etc. Therefore, the environment needs to be influenced by different types of actions. For example, WT and MV tool-calling environments provide many APIs for the agent to call; the DB environment requires the output of SQL code; BabyAI and ALFWorld require some embodied actions, while WD requires plain text output.
>   - In fact, many environments require diverse capabilities, spanning the input, decision-making, and action sides.
>
> - **From another perspective,** in the field of NLP, many researchers believe that an LLM-based agent is not for simple chat or single-turn QA, but for **complex, long-term, multi-turn sequential decision-making tasks through interaction with the environment**. These tasks are also referred to as **agent tasks** by researchers in the LLM/NLP field [3, 4, 5, 6, 7]. Therefore, in this context, the environments and tasks we need to build in AgentGym must have these characteristics. Specifically, the tasks we include generally require multiple rounds of interaction with the environment to complete, and the context length is usually longer than that of LLM's Q&A tasks or reasoning tasks (because it needs to include instructions, historical information, and current observations). From this perspective, AgentGym is an environment framework specifically designed for LLM-based agents.

---

> ### Author Response · Authors · 2024-11-22
> **Response to Reviewer VNyh [2/2]**
>
> ## 2. Question about technical details in Sec 3.
>
> To provide a detailed overview of our framework, we indeed describe many details regarding framework architecture, instruction collection, and trajectory generation in several subsections of Section 3. This undoubtedly affects the focus of our paper and the reader's perception. Based on your suggestion, we have made revisions (See Q1).
>
> ## 3. Question about the varying performance in different environments.
>
> Thank you for your insightful question. We believe your suggestion is very important for our paper. **In the first point of the General Response**, we provide a more detailed discussion and attempt to analyze the underlying reasons for this phenomenon with experiments.
>
> Thank you once again for your valuable comments! If you find our clarifications satisfactory, we kindly ask for your consideration in adjusting your scores accordingly.
>
> **References**
>
> [1] Wooldridge M, Jennings N R. Intelligent agents: Theory and practice[J]. The knowledge engineering review, 1995, 10(2): 115-152.
>
> [2] Goodwin R. Formalizing properties of agents[J]. Journal of Logic and Computation, 1995, 5(6): 763-781.
>
> [3] Xi Z, Chen W, Guo X, et al. The rise and potential of large language model based agents: A survey[J]. arXiv preprint arXiv:2309.07864, 2023.
>
> [4] Wang L, Ma C, Feng X, et al. A survey on large language model based autonomous agents[J]. Frontiers of Computer Science, 2024, 18(6): 186345.
>
> [5] Yao S, Zhao J, Yu D, et al. React: Synergizing reasoning and acting in language models. ICLR 2023.
>
> [6] Sumers T R, Yao S, Narasimhan K, et al. Cognitive architectures for language agents. TMLR 2024.
>
> [7] Zhou Y, Zanette A, Pan J, et al. Archer: Training language model agents via hierarchical multi-turn rl. ICML 2024.

---

> > ### Comment · Area_Chair_kPX6 · 2024-11-24
> > **Please respond to rebuttal asap**
> >
> > Dear reviewer,
> > The process only works if we engage in discussion. Can you please respond to the rebuttal provided by the authors ASAP?

---

> > > ### Comment · Reviewer_VNyh · 2024-11-25
> > >
> > > I appreciate the authors’ detailed response to my concerns. However, I believe the main issue remains unresolved.
> > >
> > > In the reply, the authors identify their key research question as:
> > >
> > > > **How can we build generally-capable agents that can handle multi-turn decision-making tasks across various interactive environments?**
> > >
> > > While this is an ambitious and significant question, I find it potentially **too broad** to be effectively addressed in a single paper. This breadth could make the work challenging to follow and may leave readers with the impression that the paper attempts to tackle a grand problem without providing sufficient methodological or empirical support to justify its conclusions.
> > >
> > > To clarify my concerns, I would encourage the authors to reflect on the following points:
> > >
> > > ## 1. Validity of Environments and Tasks
> > > - Why were the selected environments and tasks chosen as the basis for evaluation?
> > > - Do these environments adequately demonstrate the agents’ ability to generalize across "various interactive environments"?
> > > If the chosen tasks are narrow or overly specific, it may undermine the claim of generalizability.
> > >
> > > ## 2. Comparison with Existing Methods
> > > - What **fundamentally differentiates** the proposed method, *AgentEvol*, from existing approaches?
> > > - While the concept of *self-evolution* is highlighted, it appears to focus on **collecting more data**.
> > >     - If the solution depends heavily on data collection, why do **current LLMs**, despite access to vast datasets, still struggle in these scenarios?
> > >     - Does fine-tuning a model on data from specific environments effectively address the broader problem of building generally-capable agents?
> > > This approach risks being overly task-specific and may not generalize as claimed.
> > >
> > > ---
> > >
> > > ### **Recommendation**
> > > In general, I would suggest **narrowing the scope** of the problem to a more focused and well-defined question. This would allow for a more rigorous exploration of the proposed methodology and its implications, ultimately making the paper more comprehensible and impactful.

---

> > > > ### Author Response · Authors · 2024-11-28
> > > > **Response to Reviewer VNyh Comment 2 [3/3]**
> > > >
> > > > **References**
> > > >
> > > > [1] Weng, Lilian. (Jun 2023). “LLM-powered Autonomous Agents”. Lil’Log. https://lilianweng.github.io/posts/2023-06-23-agent/.
> > > >
> > > > [2] Xi Z, Chen W, Guo X, et al. The rise and potential of large language model based agents: A survey[J]. *arXiv* *preprint* *arXiv:2309.07864, 2023.*
> > > >
> > > > [3] Wang L, Ma C, Feng X, et al. A survey on large language model based autonomous agents[J]. *Frontiers of* *Computer Science**, 2024, 18(6): 186345.*
> > > >
> > > > [4] Liu, Xiao, et al. "Agentbench: Evaluating llms as agents." *arXiv* *preprint* *arXiv:2308.03688* (2023).
> > > >
> > > > [5] Zeng, Aohan, et al. "Agenttuning: Enabling generalized agent abilities for llms." *ACL* *2024.*
> > > >
> > > > [6] Chen, Zehui, et al. "Agent-FLAN: Designing Data and Methods of Effective Agent Tuning for Large Language Models." *ACL* *2024.*
> > > >
> > > > [7] Zhang, Jianguo, et al. "AgentOhana: Design Unified Data and Training Pipeline for Effective Agent Learning." *arXiv* *preprint* *arXiv:2402.15506* (2024).
> > > >
> > > > [8] Zhou, Yifei, et al. "Archer: Training language model agents via hierarchical multi-turn rl." *ICML, 2024*.
> > > >
> > > > [9] Song, Yifan, et al. "Trial and error: Exploration-based trajectory optimization for llm agents." *arXiv* *preprint* *arXiv:2403.02502* (2024).
> > > >
> > > > [10] Li, Chengpeng, et al. "Mugglemath: Assessing the impact of query and response augmentation on math reasoning." *ACL* *2024*.
> > > >
> > > > [11] Yu, Longhui, et al. "Metamath: Bootstrap your own mathematical questions for large language models." *arXiv* *preprint* *arXiv:2309.12284* (2023).
> > > >
> > > > [12] Tarasov, Denis, and Kumar Shridhar. "Distilling LLMs’ Decomposition Abilities into Compact Language Models." *ICML 2024*.
> > > >
> > > > [13] Haluptzok, Patrick, Matthew Bowers, and Adam Tauman Kalai. "Language models can teach themselves to program better." *ICLR* *2023*.
> > > >
> > > > [14] Luo, Ziyang, et al. "Wizardcoder: Empowering code large language models with evol-instruct." *ICLR* *2024*.
> > > >
> > > > [15] Wei, Yuxiang, et al. "Magicoder: Empowering code generation with oss-instruct." *ICML 2024*.
> > > >
> > > > [16] Sun, Zhiqing, et al. "SALMON: Self-Alignment with Instructable Reward Models." *ICLR* *2024*.
> > > >
> > > > [17] Sun, Zhiqing, et al. "Principle-driven self-alignment of language models from scratch with minimal human supervision." *NeurIPS 2024*.
> > > >
> > > > [18] Schick, Timo, et al. "Toolformer: Language models can teach themselves to use tools." *NeurIPS 2023*.
> > > >
> > > > [19] Touvron, Hugo, et al. "Llama: Open and efficient foundation language models." *arXiv* *preprint arXiv:2302.13971* (2023).
> > > >
> > > > [20] Dubey, Abhimanyu, et al. "The llama 3 herd of models." *arXiv* *preprint* *arXiv:2407.21783* (2024).
> > > >
> > > > [21] Zhu, Qihao, et al. "DeepSeek-Coder-V2: Breaking the Barrier of Closed-Source Models in Code Intelligence." *arXiv* *preprint* *arXiv:2406.11931* (2024).

---

> > > > > ### Comment · Reviewer_VNyh · 2024-11-30
> > > > >
> > > > > Thank you for your detailed response. After carefully considering your rebuttal and the concerns raised by other reviewers, I think this paper would benefit from major revisions based on the feedback provided by all reviewers. Therefore, I decided to maintain my original score.

---

> ### Author Response · Authors · 2024-11-28
> **Response to Reviewer VNyh Comment 2 [1/3]**
>
> Thank you for your kind question and suggestion. Our clarifications are as follows:
>
> ## 1. Re: Clarification on our key research question
>
> Thank you very much for your feedback. It seems that our previous statements may have been somewhat too broad, leading to potential misunderstandings.
>
> - To clarify, we **narrow our research question to exploring how to develop LLM-based agents capable of self-evolution across** **multiple and diverse** **environments**. This is why we include the AgentGym framework, trajectory sets, benchmark suites, and AgentEvol in the paper. We do not intend to tackle the broader challenges of AGI or strong AI, but instead aim to advance the capabilities of intelligent agents within diverse environments.
> - Based on your feedback, we will narrow and clarify the scope of our study in the manuscript.
>
> ## 2. Re: "Validity of Environments and Tasks"
>
> > "Why were the selected environments and tasks chosen as the basis for evaluation?"
>
> - In our response to "Q1: regarding the starting point of the AgentGym framework and environments," we have clarified why we select these environments. We consider two main aspects: **1. The definition of an LLM-based agent**, including the input side, decision-making side, and action side [1,2,3]. **2. The capabilities that the LLM-based agent community is concerned with**, e.g. handling complex, long-term, multi-turn sequential decision-making tasks through interaction with the environment.
> - In addition, following your suggestion, we have also detailed our thoughts on this question and the related technical features in the revised Section 2.1 of the paper.
> - We hope our response addresses your concerns. Please let us know if you have any further questions.
>
> > "Do these environments adequately demonstrate the agents’ ability to generalize across various interactive environments? If the chosen tasks are narrow or overly specific, it may undermine the claim of generalizability."
>
> - We have carefully collected a diverse range of environments from the LLM-based agent field, including categories such as web navigating, house-holding, tool-using, programming, text games, and digital games, in order to avoid being "narrow or overly specific". Additionally, **compared to** **related** **works, we have included a greater number and variety of tasks** [4,5,6]. For more detailed information, please refer to Table 1 on p.3 (comparison with other agent frameworks). These environments are sufficient to demonstrate the multi-tasking capabilities of the agents. We expect to include more tasks and environments in the future, as this is a rapidly evolving field, and new challenges continually emerge.
>
> ## 3. Re: "Comparison with Existing Methods"
>
> Thank you for your insightful comment. To clarify the fundamental differentiation of AgentEvol from existing approaches, we highlight the following points:
>
> - AgentEvol enables LLM-based agents to perform **exploration and learning across multiple environments.**
> - Existing LLM-based agent methods typically rely on imitating expert trajectories, such as behavioral cloning. These methods are often **task-specific and lack exploration** [5, 6, 7]. Additionally, some approaches allow agents to improve through environmental feedback (i.e., self-improvement), but these methods are typically constrained to **specific environments**, limiting their performance across multiple and diverse environments [8, 9].

---

> ### Author Response · Authors · 2024-11-28
> **Response to Reviewer VNyh Comment 2 [2/3]**
>
> ## 4. Re: Question about the data used for an LLM-based agent
>
> > "While the concept of *self-evolution* is highlighted, it appears to focus on **collecting more data**."
>
> - Self-evolution can be described as self-improvement, which focuses on **enhancing the capabilities of LLM-based agents**. The goal is to train more capable agents by collecting more and higher-quality data through interaction with various environments. **This aligns with the technical approach of many works in the LLM field**, such as mathematical reasoning [10, 11, 12], code generation [13, 14, 15], alignment [16, 17], tool using [18].
>
> > If the solution depends heavily on data collection, why do **current** **LLMs**, despite access to vast datasets, still struggle in these scenarios?
>
> - Although current LLMs have access to large amounts of data, the data is not highly relevant or of high quality for LLM-based agent tasks, which causes them to struggle in these environments. For example, pre-training uses terabyte-scale corpora, such as Common Crawl, C4, Github, Wikipedia, StackExchange, etc. [19, 20, 21], which provides the model with basic language understanding and generation abilities.  However, the performance of these base models on agent tasks is moderate, as demonstrated by Llama2-Chat in our main results.
>
> ## 5. Re: Question about overly task-specific
>
> Thank you for your feedback. In the following, we will carefully respond to your question.
>
> - Our goal is to develop LLM-based agents capable of self-evolution across multiple and diverse environments (e.g., web shopping and household), instead of a broader problem of AGI or strong AI. We apologize for the misunderstanding and will clarify this point in the revised version of the paper.
> - To achieve this, LLMs provide a solid backbone with prior knowledge and general capabilities to some extent [1, 2, 3]. This foundation enables the agent to perform effectively across multi-task environments.
> - Moreover, we have made considerable efforts to collect **a diverse set of environments** within the framework. Our results demonstrate that AgentEvol shows **promising performance** across these environments. Furthermore, it shows **generalization capabilities** on unseen tasks and environments, as detailed in the third point of our General Response. This highlights **our approach is not overly task-specific**.
>
> ## 6. Re: Recommendation for narrowing the scope
>
> Thank you for your valuable suggestion. We acknowledge that the statements of our key research question may have caused some confusion. To clarify, we have narrowed our focus and now explicitly frame the research question as: "How to develop LLM-based agents capable of self-evolution across diverse environments."  We include AgentGym framework,  dataset, benchmark suite, and self-improvement methods across diverse environments for the research question.
>
> We will revise the paper to provide a clearer and more focused description of the research question. Thank you again for your helpful feedback.

---

### Official Review · Reviewer_ZWi2 · 2024-11-04

**Soundness:** 4
**Presentation:** 3
**Contribution:** 4
**Rating:** 8
**Confidence:** 4

**Summary:**

* Comprehensive framework designed to evaluate and evolve LLM based agents across diverse interactive environments
* Agents can explore and learn autonomously
* AGENTGYM includes a suite of 14 environments covering seven real-world scenarios, along with expanded instructions, a benchmark suite (AGENTEVAL), and high-quality trajectory datasets (AGENTTRAJ and AGENTTRAJ-L)
* Presents AGENTEVOL, an exploration-learning method derived from the RL as Inference framework, enabling agents to self-evolve across multiple environments
* Experimental results demonstrate that agents evolved using AGENTEVOL can achieve performance comparable to or exceeding state-of-the-art models

**Strengths:**

Comprehensive Framework: AGENTGYM provides a versatile and extensible platform for evaluating and training LLM-based agents across a wide range of tasks and environments. This breadth enhances the generalizability and practicality of the agents developed using this framework.

Innovative Methodology: The introduction of AGENTEVOL as an exploration-learning method allows agents to improve themselves based on environmental feedback, reducing reliance on human supervision and expert trajectories

Promising Experimental Results: The agents evolved using AGENTEVOL outperform baseline models and, in some cases, exceed the performance of state-of-the-art models like GPT-4-Turbo on several tasks

Addressing Scalability: The paper tackles the scalability issues inherent in SFT by allowing agents to explore and learn autonomously, which could lead to more efficient and scalable training processes

**Weaknesses:**

Scalability Concerns with Larger Models: The computational costs associated with running large LLMs in interactive environments are significant. The paper could benefit from a more detailed analysis of the scalability of AGENTEVOL when using larger models

Comparative Analysis: The comparisons with other exploration-based methods, such as PPO and LLM-Planner, are brief. A more in-depth analysis could strengthen the validation of AGENTEVOL's effectiveness

Overfitting Risk: The potential for agents to overfit to specific environments or tasks during the self-evolution process is not thoroughly addressed. Strategies to mitigate overfitting would enhance the robustness of the approach

**Questions:**

How does the method ensure safety and prevent harmful behaviors during self-modification? How do you put constraints on it during the evolution process?

---

> ### Author Response · Authors · 2024-11-22
> **Response to Reviewer ZWi2 [1/2]**
>
> Thank you very much for recognizing the novelty, contributions, and writing of our work. This has greatly encouraged us. We hope our response addresses your concerns:
>
> ## 1. Question regarding the scalability of AGENTEVOL.
>
> Our analysis is from two perspectives:
>
> - **Performance Improvement with Larger Models**
>   - In Section 4.3, Table 5, we conduct experiments with models of different scales (DeepSeek-Coder-1.3B and Llama-2-13B), and the results are visualized [https://anonymous.4open.science/r/AgentGym-ICLR-rebuttal-1B8D/Scalability_of_AgentEvol.png](https://anonymous.4open.science/r/AgentGym-ICLR-rebuttal-1B8D/Scalability_of_AgentEvol.png) to better illustrate the observed scaling trends. AgentEvol demonstrates significantly improved performance as the model size increases, although we do not perform fine-grained hyper-parameter searching. This indicates that AgentEvol can leverage larger models to achieve further performance gains.
>
> - **Analysis of Computational Costs**
>   - As shown in the right Figure (accessible [https://anonymous.4open.science/r/AgentGym-ICLR-rebuttal-1B8D/Scalability_of_AgentEvol.png](https://anonymous.4open.science/r/AgentGym-ICLR-rebuttal-1B8D/Scalability_of_AgentEvol.png)), we also analyze the inference time for models of varying scales. While the inference time does increase with model size, larger models exhibit higher interaction efficiency, resulting in only a marginal increase in overall inference time. This suggests that AgentEvol can scale to larger models while maintaining an acceptable computational costs.
>
>
> We appreciate your insightful comments, which have helped us refine our discussion on scalability.
>
> ## 2. Question about comparisons with other exploration-based methods.
>
> We apologize for the brief analysis of other baseline methods. We have added related experiments, training curves, case studies, and in-depth analysis **in the second point of the General Response**. We hope this meets your satisfaction.

---

> ### Author Response · Authors · 2024-11-22
> **Response to Reviewer ZWi2 [2/2]**
>
> ## 3. Question regarding overfitting risk.
>
> Thank you for your professional question. In our early experiments, we did observe overfitting. We apologize for not providing a detailed discussion in the paper. We have added the corresponding experimental results below, and we hope this enhances the rigor of our paper and addresses your concerns.
>
> In the AgentEvol framework, overfitting is mainly caused by two potential factors: the continuously iterating model and the growing training dataset.
>
> - **Data:** We show the ablation results of different data merging strategies in Figure 3 on p.9. The experiment results indicate that accumulating training data generated in each iteration can lead to performance degradation after several iterations. Therefore, **we choose to merge the generated data from each iteration with the initial** **training set**. This approach ensures the model's learning of basic skills while also allowing it to learn higher-quality, higher-reward trajectory data over iterations.
> - **Model:**  In our experimental setup, **we optimize the initial agent at each iteration to mitigate** **overfitting**. We add further experiments and analysis to **the third point of clarifications of the General Response**. The results are presented [https://anonymous.4open.science/r/AgentGym-ICLR-rebuttal-1B8D/Ablation_study_overfit.png](https://anonymous.4open.science/r/AgentGym-ICLR-rebuttal-1B8D/Ablation_study_overfit.png).
>
> In summary, we have provided a deeper discussion on overfitting risk from both the data and model perspectives. We hope these analyses clarify your concerns.
>
> ## 4. Question about safety issues during AgentEvol.
>
> We try to address your question from the following perspectives:
>
> - **Data Preprocessing:** When constructing the AgentTraj dataset, we conduct strict data preprocessing. During the instruction and trajectory collection process, we check all instructions and responses for harmful or sensitive tokens, preventing harmful behavior from the source of the training data.
> - **Data Postprocessing:** In designing the various environments in AgentGym, we add corresponding rules. Harmful responses are assigned lower rewards, and these responses are discarded during the iterations of AgentEvol.
> - **Agent Environments:** Currently, all environments in AgentGym are offline (except for the Tool environment, which requires real-world APIs). Therefore, the agent’s interactions take place in a sandbox, preventing any real-world harm.
> - **Preference Alignment Strategy:** For future work, we aim to extend AgentGym to real-world environments. For example, using OpenTable for restaurant bookings. In such settings, we will not only consider data preprocessing and postprocessing but also plan to introduce preference alignment techniques based on human feedback to reduce harmful behavior (e.g. RLHF, DPO) [1,2].
>
> Thank you once again for your valuable comments! We hope our responses address your concerns and enhance your confidence in our paper.
>
> **References**
>
> [1] Yang, Zonghan, et al. "Towards Unified Alignment Between Agents, Humans, and Environment." ICML 2024.
>
> [2] Yuan, Tongxin, et al. "R-judge: Benchmarking safety risk awareness for llm agents." *arXiv* *preprint* *arXiv:2401.10019* (2024).

---

> > ### Comment · Reviewer_ZWi2 · 2024-11-25
> > **Response to Rebuttal**
> >
> > I appreciate your detailed responses to each of my points. I think this is a strong paper but my score is already an 8 and I won't raise it higher than that.

---

> > > ### Author Response · Authors · 2024-11-27
> > > **Response to Reviewer ZWi2's Official Comment 2**
> > >
> > > Dear reviewer, we are deeply grateful for your recognition and kind words regarding our work, i.e., "strong paper". Your feedback is truly valuable, and we will make every effort to incorporate your suggestions into the next version of the manuscript. **If it is not too much trouble, we would kindly request that you consider enhancing your confidence score.** Thank you again for your time and consideration!

---

### Author Response · Authors · 2024-11-22
**General Response to All Reviewers [1/5]**

We are very grateful for the reviewers' recognition of our work, which is a great encouragement to us. We also appreciate the reviewers' valuable comments and suggestions. We will improve our paper based on your feedback to meet your expectations.

In general response, we want to clarify or address some questions that we believe are of common concern to multiple reviewers.

## 1. Re: Analysis of varying performance in different environments.

Thank you for your valuable question. In the response, we have added experiments, provided deeper analysis, and proposed two possible hypotheses. We hope these address your concerns.

- First, we clarify that: Our method is stronger than $BC_{base}$, but weaker than $BC_{large}$ in some tasks (SC, WD and WT). This is because $BC_{large}$ represents the upper performance bound of the BC method, utilizing LLM and human-annotated high-quality trajectory data on the complete instruction set. Despite this, our method is not weaker than $BC_{large}$ in most tasks.

- Next, we provide two perspectives to analyze the reasons behind the fluctuating performance of our method across different environments. **All experiment results are also presented in the revised Appendix F.4 .**

- **Perspective 1: Conflicts between different environments.** We add experiments about the performance of task-specific BC and **task-specific AgentEvol.** The experimental results are as follows. We find that BC on single tasks outperforms BC on multi-tasks, and AgentEvol on single tasks outperforms AgentEvol on multi-tasks. Additionally, AgentEvol on single tasks performs poorly on other tasks. This indicates that **there may be some conflicts between different environments and tasks**, which may lead to a certain degree of performance degradation and cause the varying performance in our method.

  | Method                    | WS       | ALF      | Baby     | WD       |
  | ------------------------- | -------- | -------- | -------- | -------- |
  | $BC_{base}$               | 66.5     | 77.5     | 69.3     | 12.0     |
  | $BC_{base}$(single task)  | 68.5     | 79.0     | 72.6     | 12.0     |
  | $BC_{large}$              | 73.5     | 83.0     | 74.2     | **36.0** |
  | $BC_{large}$(single task) | 74.0     | 84.0     | 76.9     | **36.0** |
  | AgentEvol                 | 77.0     | 88.0     | 82.9     | 12.0     |
  | AgentEvol on WS           | **78.0** | 3.0      | 2.8      | 4.0      |
  | AgentEvol on ALF          | 2.0      | **89.5** | 0.5      | 4.0      |
  | AgentEvol on Baby         | 1.5      | 1.0      | **84.1** | 4.0      |
  | AgentEvol on WD           | 0.0      | 0.0      | 6.80     | 24.0     |

- **Perspective 2: Lack of exploration capability.** We compare the size of trajectory data used to train the agent between AgentEvol and BC methods. The results are represented below.

  - It is important to note that the data for $BC_{large}$ comes from SOTA models and expert human annotators (AgentTraj-L). **AgentEvol's data consists of two parts: 1. AgentTraj, a smaller set used to train the base agent; 2. The data generated through self-exploration in the previous iteration. The subscript in the table indicates the difference in data size between the current iteration and $BC_{large}$.** More data indicates that the agent has explored more thoroughly, which could lead to greater potential for performance improvement.

    | Method       | iter | WS             | ALF           | WD           | WT          |
    | ------------ | ---- | -------------- | ------------- | ------------ | ----------- |
    | $BC_{large}$ |      | 3930           | 2420          | 955          | 311         |
    | AGENTEVOL    | 1    | 5661$_{+1731}$ | 2529$_{+109}$ | 585$_{-370}$ | 264$_{-47}$ |
    |              | 2    | 5982$_{+2052}$ | 2714$_{+134}$ | 585$_{-370}$ | 292$_{-19}$ |
    |              | 3    | 6061$_{+2131}$ | 2734$_{+314}$ | 579$_{-376}$ | 284$_{-27}$ |

  - **We find significant gaps in the agent's exploration ability across different environments**. In some environments (e.g. WS, ALF), the agent explores more successful trajectories over time, while in others, the performance even decreases. **Overall, AgentEvol outperformes $BC_{large}$ in 7 environments, and is weaker than $BC_{large}$ in 3 environments.**

---

> ### Author Response · Authors · 2024-11-22
> **General Response to All Reviewers [2/5]**
>
> ## 2. Re: Comparison between AgentEvol and other RL / exploration-based methods.
>
> Thank you for your insightful questions. We have added extra RL baselines, Reward Weighted Regression (RWR) algorithm, and provided deeper analysis with case studies and training reward curves. We hope these additional discussions will strengthen the validation of AgentEvol's effectiveness.
>
> - **Regarding RWR baseline**: Following Reviewer jPpZ’s suggestion, we discuss the differences in implementation details between AgentEvol and RWR [1]. We believe it is valuable to include this classic offline RL algorithm as a baseline. Additional experiments have been added below.
>
> - **Regarding DPO baseline**: We would like to kindly emphasize that DPO is also a well-known offline alternative to RL in the field of LLMs [2, 3]. **The experimental results are provided in p.10, Table 4**. We have also added the implementation details of DPO in Appendix E of the revised manuscript.
>
> - Following the suggestions of Reviewer ZWi2 and Reviewer PSRy, we have conducted a detailed comparison of AgentEvol with other RL methods and exploration-based methods. **All statistical results are provided below and also included in the revised Appendix F.5.**
>
>   - **Selection of baselines**: Our evaluation is comprehensive and sufficient, including Prompt-based, BC, offline-RL, and online-RL methods. Experimental results demonstrate that AgentEvol achieves superior performance across various tasks when compared with representative algorithms.
>
>   - **Training cost**: The training costs for $BC_{large}$, RWR, and AgentEvol are set as baselines, as they all optimize the policy in a behavior cloning manner. In contrast, DPO and PPO methods have significantly higher training costs. DPO requires loading both the actor and reference models and computing probability distributions for chosen and rejected responses. PPO, being an online RL method, involves sampling and policy optimization simultaneously, leading to more intensive training times.
>
>     | Method          | Type             | Avg. Training Cost | Accuracy (WS) | Accuracy (ALF) | Accuracy (Baby) |
>     | --------------- | ---------------- | ------------------ | ------------- | -------------- | --------------- |
>     | LLM-Planner [4] | prompt-based     | /                  | 18.9          | 68.5           | 82.5            |
>     | $BC_{large}$    | behavior cloning | 1 ×                | 73.5          | 83.0           | 74.2            |
>     | RWR [1]         | offline-RL       | 1 ×                | 68.0          | 76.5           | 82.1            |
>     | DPO [2]         | offline-RL       | 4.3 ×              | 75.0          | 86.5           | 78.3            |
>     | PPO [5]         | online-RL        | **15 ×**           | 68.0          | 83.5           | 69.8            |
>     | **AGENTEVOL**   | offline-RL       | 1 ×                | **76.5**      | **88.0**       | **82.7**        |
>
>   - **Learning stability**: The results are presented [https://anonymous.4open.science/r/AgentGym-ICLR-rebuttal-1B8D/Learning_stability.png](https://anonymous.4open.science/r/AgentGym-ICLR-rebuttal-1B8D/Learning_stability.png). For consistency, we set the smallest unit of the x-axis for training time as an epoch. **It is clear that algorithms optimized with BC objectives are more stable in performance improvements, leading to faster** **convergence**. While DPO shows significant improvement in the early stages, **overfitting** **occurs quickly as training progresses**. PPO, on the other hand, exhibits noticeable instability throughout the training process, with no clear learning trend during the same number of epochs.
>
>   - **Case Study**: We compare the performance of several RL baselines when executing the same task in the WebShop environment. The results are presented [https://anonymous.4open.science/r/AgentGym-ICLR-rebuttal-1B8D/Case_rl.pdf](https://anonymous.4open.science/r/AgentGym-ICLR-rebuttal-1B8D/Case_rl.pdf). The RWR and DPO baselines lead to the selection of the first item without considering the price constraint, resulting in task failure. In comparison, the PPO baseline continuously clicks "Next Page" without effectively extracting relevant information from the environment, also failing to find a suitable item. **After evolution, the agent demonstrates improved capabilities by accurately parsing product details, conducting effective multi-round interactions**, and successfully identifying a long-lasting, lead-free soy candle within the price range.

---

> ### Author Response · Authors · 2024-11-22
> **General Response to All Reviewers [3/5]**
>
> ## [Continuation of 2]
>
> - Additionally, we provide the Mean Training Reward curve for each environment following Reviewer jPpZ's advice. The results are presented [https://anonymous.4open.science/r/AgentGym-ICLR-rebuttal-1B8D/Mean_training_reward.png](https://anonymous.4open.science/r/AgentGym-ICLR-rebuttal-1B8D/Mean_training_reward.png). For WebShop and ALFWorld, we present the results for 10 epochs; for BabyAI, we present the results for 8 epochs. We also observe that PPO encounters instability and fluctuations in training rewards. **This could be due to the standard PPO algorithm, which only uses outcome-based rewards and struggles with optimizing sparse, long-term, and multi-turn trajectories, limiting the model’s exploration and learning**. In the early stages of training, rewards encounter a noticeable drop when the critic model is not fully optimized. This phenomenon is consistent with the performance of LLMs in the standard RLHF pipeline [5, 6, 7]. Full training details are included in Appendix E of the revised manuscript.

---

> ### Author Response · Authors · 2024-11-22
> **General Response to All Reviewers [4/5]**
>
> ## 3. Re: The lack of thorough evaluation of AgentEvol.
>
> Thank you for your valuable suggestions! We have added some additional experiments regarding AgentEvol
>
> 1. Adding experiments of **evaluation on OOD tasks and environments**.
>
>    - To explore performance on tasks or environments not seen during the evolution phase (i.e. OOD tasks), we conduct experiments. In these experiments, the task types and settings in ALF and Baby are unseen by the agent during both the BC and AgentEvol phases, and the entire environments of Tool-Academia(AM) and Tool-Sheet(ST) are entirely new for the agent. The results are presented below, showing that our method demonstrates stronger generalization ability on unseen tasks and environments compared to other methods.
>
>      | Method         | ALF-OOD  | Baby-OOD | AM       | ST       |
>      | -------------- | -------- | -------- | -------- | -------- |
>      | Llama2-Chat-7B | 0.0      | 2.2      | 0.0      | 0.0      |
>      | AgentLM-7B     | 57.7     | 4.4      | 10.0     | 14.3     |
>      | $BC_{base}$    | 60.8     | 6.2      | 20.0     | 24.3     |
>      | $BC_{large}$   | 64.9     | 6.1      | 20.0     | 25.2     |
>      | **AGENTEVOL**  | **67.5** | **6.2**  | **25.0** | **26.2** |
>
> 2. Adding experiments about the performance of task-specific BC and **task-specific AgentEvol.**
>
>    - We have provided a detailed discussion in the first point of the General Response. The results are presented in the first point of the General Response . Here, we mainly highlight the results of the BC and AgentEvol methods running in isolated environments.
>
> 3. Adding experiments regarding the choice of base model.
>
>    - Following Reviewer ZWi2's advice, we examine the overfitting risk during our AgentEvol process. Specifically, we conduct experiments comparing two strategies: fine-tuning the initial agent and fine-tuning the agent from the last iteration. As shown in Figure 6 in the Appendix F.2 (also accessible [https://anonymous.4open.science/r/AgentGym-ICLR-rebuttal-1B8D/Ablation_study_overfit.png](https://anonymous.4open.science/r/AgentGym-ICLR-rebuttal-1B8D/Ablation_study_overfit.png)), re-optimizing the initial agent mitigates overfitting and ensures more stable performance across diverse environments compared to continuous fine-tuning. **Therefore, we select the initial agent as the base model in each iteration of self-evolution.**

---

> ### Author Response · Authors · 2024-11-22
> **General Response to All Reviewers [5/5]**
>
> ## 4. Re: Question regarding the focus of our paper.
>
> To address the reviewers' concerns, we would like to clarify the starting point and motivation of our paper:
>
> - In this paper, the key issue we explore is: **How can we build generally-capable agents that can handle multi-turn decision-making tasks across various interactive environments?** Our goal is to train LLM-based agents that can explore and evolve themselves across diverse environments.
> - To achieve this, **we identify three key elements** (which we detail in the abstract and introduction). 1. diverse interactive environments for agent exploration, 2. a trajectory set to equip agents with basic capabilities and prior knowledge, and 3. an effective and scalable approach for agent improvement across environments. Specifically:
>   - **The AgentGym framework implements the first element**, while the AgentTraj dataset represents the second element.
>   - Based on these elements, **we continue to explore the potential of training self-evolving agents and demonstrate the effectiveness of AgentEvol through experiments.** This constitutes the third element.
> - Based on Reviewer VNyh's feedback, we explain why we select these environments in AgentGym. If you are interested, please refer to Q1 in response to Reviewer VNyh.
> - Following the suggestions of Reviewer VNyh and Reviewer jPpZ, we have revised the writing and format of our paper.
>
> **References**
>
> [1] Peters, Jan, and Stefan Schaal. "Reinforcement learning by reward-weighted regression for operational space control." *Proceedings of the 24th international conference on* *Machine learning*. 2007.
>
> [2] Rafailov, Rafael, et al. "Direct preference optimization: Your language model is secretly a reward model." *Advances in Neural Information Processing Systems* 36 (2024).
>
> [3] Song, Yifan, et al. "Trial and error: Exploration-based trajectory optimization for llm agents." *arXiv* *preprint* *arXiv:2403.02502* (2024).
>
> [4] Song, Chan Hee, et al. "Llm-planner: Few-shot grounded planning for embodied agents with large language models." *Proceedings of the IEEE/CVF* *International Conference on Computer Vision*. 2023.
>
> [5] Ouyang, Long, et al. "Training language models to follow instructions with human feedback." *Advances in neural information processing systems* 35 (2022): 27730-27744.
>
> [6] Zheng, Rui, et al. "Secrets of rlhf in large language models part i: Ppo." *arXiv* *preprint* *arXiv:2307.04964* (2023).
>
> [7] Trung, Luong, et al. "Reft: Reasoning with reinforced fine-tuning." *Proceedings of the 62nd Annual Meeting of* *the Association for Computational Linguistics* *(Volume 1: Long Papers)*. 2024.

---

### Meta-Review · Area_Chair_kPX6 · 2024-12-20

**Metareview:**

This paper introduces a benchmark for LLM based agents AgentGym and then an RL algorithm for learning performant agents called AgentEvol. The benchmark is useful across domains from games to web agents and embodied agents, tools and even code. The RL algorithm, derived from control as inference boils down to something pretty close to RWR, with a particular choice of reward.

Strengths:
There is a tremendous engineering effort behind this paper, the benchmark is very cool and very useful.
Getting an RL algorithm to work in this setting is a nontrivial effort!

Weaknesses
The paper is doing too much. Ideally the AgentGym can be proposed and analyzed by itself, without getting into control as inference and RL algorithm derivation and such. All reviewers noted this as well, it's just that currently stated the paper is a bit scattered across benchmarks and methods with not enough detail devoted to either. The paper will have a lot of impact, it just needs to be narrowed down to 1 clear impact, preferably separating the benchmark and the agentevol parts.

The difference from RWR needs to be made clear - the 0/1 rewards reason is not a particularly compelling one, you can set rewards to -inf and 0 and you get the same impact. I'd suggest skipping the whole derivation in that case.

Figure 1 is not very professional and informative for an academic conference, this should be restructured to be more informative.

^ this is the reason I am suggesting we go against acceptance of this paper. It is a high potential paper, just needs more focus to really reach it's impact.

**Additional Comments On Reviewer Discussion:**

The reviewers brought up two major points - one about focus and the other about the RL algorithm.  Both are very valid points and I did not see a very clear rebuttal from the authors that was satisfactory in responding to these. The paper needs to be narrowed in scope and needs to have a clear focus, rather than trying to do it all.

---

### Decision · Program_Chairs · 2025-01-22

Reject